# Phylogeography of *Aphyocypris normalis* Nichols and Pope, 1927 at Hainan Island and adjacent areas based on mitochondrial DNA data

**I-Shiung Chen[1], Nian-Hong Jang-Liaw[2]***

**1** Institute of Marine Biology, National Taiwan Ocean University, Keelung, Taiwan, **2** Conservation Genetics Laboratory, Conservation and Research Center, Taipei Zoo, Taipei, Taiwan

\* taco.tw@gmail.com

**Data Availability Statement:** All data are available at NCBI Sequence, accession numbers: KC209831–KC209957 and KC209958–KC209979.

## Abstract

We investigated the genetic structure of the freshwater fish *Aphyocypris normalis*, in 33 populations around Hainan Island and southern mainland China. Sequencing of the mitochondrial DNA (mtDNA) cytochrome *b* from 127 specimens yielded 47 haplotypes, from which we inferred a Bayesian tree. This revealed three major divergences: a principal clade of specimens with widespread geographic distribution, plus two clades with limited distribution. We estimated that these diverged between 1.05–0.16 Ma. Additionally, based on molecular data and comparing with the climate patterns of Hainan Island, eight phylogeographic ranges (populations) of *A. normalis* were constructed: the eastern plain (E), northeastern hills and plain (NE), northwestern hills and lowlands (NW), central mountains (C), southeastern hills and plain (SE), southern mountains and hills (S), southwestern mountains and lowlands (SW), and western lowlands (W). The patterns of geographical divergence in this species do not reflect the isolation caused by the Qiongzhou (Hainan) Strait, which would generally be experienced by terrestrial animals on isolated islands. The present results indicate that the major clades within *A. normalis* have diverged before the temporary land bridge existed across the strait during the Last Glacial Maximum.

## Introduction

Biogeography is tied closely to both ecology and phylogenetics and is a key topic in classic texts on phylogenetic systematics [1–3]. Comparing the phylogenies of co-distributed taxa provides insight into how factors such as movement capability and life history influence the genetic signature left by geographic events. Beyond descriptive biogeography, understanding the factors that affect the evolution of taxa in a region allows for the development of testable predictions relating to patterns of genetic diversity [4].

Hainan, a *c.* 33,920 km$^2$ tropical island off the shore of southern China, is widely recognized as one of the world's biodiversity hotspots [5–10]. This large island is located in Beibu Bay (Gulf of Tonkin) and is isolated from Guangdong's Leizhou Peninsula to the north by the

**Funding:** This research was funded by National Science Council, Taiwan (NSC100-2923-B019-001-MY3) for the first author.

**Competing interests:** The authors have declared that no competing interests exist.

Qiongzhou (Hainan) Strait (maximum depth 114 m; Fig 1). Generally, Hainan Island shows moderate topographic relief, with few mountains over 1,500 m above sea level (ASL) near its center, with Wuzhizhan Mountain being the highest point on the island (1,840 m ASL; see Fig 5A, 5B). Due to its tropical monsoon climate, Hainan has distinct dry (November to April) and rainy (May to October) seasons. Most of the rivers in Hainan originate in the central area of the island and flow radially in different directions: the three largest are the Nandu River in the northern part of the island, the Changhua River in the west, and the Wanquan River in the east. The eastern part of Hainan lies in the path of typhoons, from which derives 80–90% of the island's annual precipitation during the summer rainy season. Major flooding occurs due to typhoons and sudden heavy rains during the rainy season [11].

The extent and patterns of intraspecific diversification vary among species and by general ecology (e.g., aquatic vs. terrestrial; [12]). In general, freshwater aquatic animals on islands,

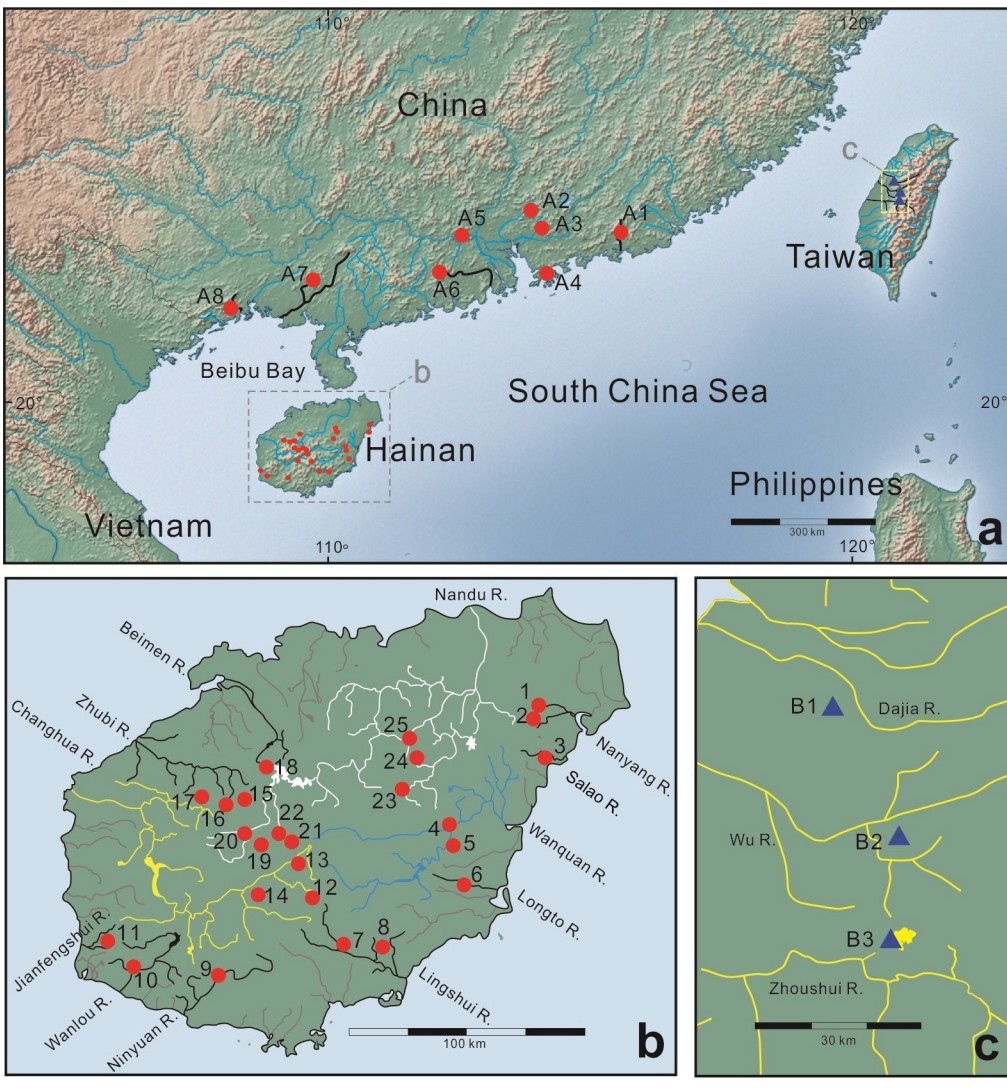

**Fig 1. Locations of sampling sites in this study.** Site numbers and river systems are listed in Table 1. a, southern mainland China sampling sites for *Aphyocypris normalis* (solid red circles); b, Hainan Island sampling sites for *A. normalis* (solid red circles). Three major river systems of Hainan Island are identified by different colors: white–the Nandu River; blue–the Wanquan River; yellow–the Changhua River; c, Taiwan sampling sites for *A. moltrechti* as the outgroup (solid blue triangles).

such as freshwater fishes, should share a similar evolutionary history to terrestrial species, and provide excellent opportunities for historical biogeographic research because of their limited dispersal capabilities. As a distinct biogeographical unit in China's zoogeography, Hainan Island hosts a unique assemblage of primary freshwater animals, mostly cyprinids [9, 13, 14]. At least 769 species of Cyprinidae have been reported in China [15, 16], of which at least 66 have been reported on Hainan Island [10, 17]. Their high biodiversity and wide distribution make Hainan cyprinids ideal subjects for phylogeographic studies.

*Aphyocypris normalis* Nichols and Pope, 1927 is a freshwater fish that ranges over southern mainland China and Hainan Island [18], as well as Vietnam on the Indochina Peninsula [19]. In mainland China and Hainan, we have observed that *A. normalis* is distributed in small brooks on hills and mountains up to elevations of *ca.* 640 m ASL. Due to its wide distribution in this area, analyses of the phylogenetic and demographic history of *A. normalis* across Hainan Island and adjacent areas could clarify the historical connections of the major river systems in these regions. The information of this fish on Indochina Peninsula is very limited, for example, the first formal record of *A. normalis* on this peninsula was in 2011 from northern and central Vietnam [19].

Recently, molecular data have been widely applied in biogeographic studies. Several phylogenetic and phylogeographic investigations using molecular approaches have been conducted on terrestrial species on Hainan Island, including: earthworms [20], a freshwater crab [21], freshwater fishes [22–30], a babbler [31], the peacock pheasant [32], deer [33], and ferns [34, 35] from Hainan. However, most of these reports treated Hainan Island merely as a single sampling source. Moreover, though both Huang et al. [25] and Li et al. [26] reported on the mitochondrial phylogeography of *A. normalis* in a similar geographic region, their sampling density was insufficient to clarify the phylogeography of this species on Hainan Island.

Herein, we present the first densely sampled phylogeographic study focused on the historical biogeography of a Hainan native species. Specifically, we examined the phylogeographic structuring of *A. normalis* in rivers across Hainan Island, plus in its adjacent range in mainland China, to assess genetic diversity through comprehensive geographic sampling. Moreover, we introduced a climate division hypothesis [36] to discuss the possible isolation mechanism of freshwater fauna in Hainan Island. The aims of our study were to i) construct a detailed phylogeny of *A. normalis* on Hainan Island based on data from the mitochondrial cytochrome *b* gene; ii) infer the distribution history of *A. normalis*; and iii) understand its evolutionary divergence in relation to Hainan Island's historic climatic patterns. On the basis of our results, we propose a hypothesis on the division of phylogeographic ranges for freshwater fishes of Hainan Island. In addition, short comments on the population diversity of a closely-related species, *Aphyocypris moltrechti* (Regan 1908) from Taiwan, and on the taxonomic status of this sibling cyprinid are provided.

## Materials and methods

### Sample collection

We utilized ethanol-fixed tissue samples from 127 specimens of *A. normalis*, collected at 25 sites in Hainan (83 specimens) and at 8 sites in southern mainland China (44 specimens). This sampling encompasses almost the entire geographic range of the species. The Hainan specimens were collected by the authors, plus by members of the first author's laboratory at National Taiwan Ocean University. As an outgroup, we utilized samples from 22 specimens of *A. moltrechti*, a freshwater fish native to Taiwan, that were collected at 3 localities (Fig 1 and Table 1). A total of 127 *A. normalis* specimens were successfully sequenced and the sequences were deposited in the GenBank database (accession codes: KC209831–KC209957). Eighty-

**Table 1. Sample locations and haplotype distributions of *Aphyocypris* samples used in this study.**

| | No. | River system | altitude (m) | Ns | Haplotypes (no. of individuals) | Clade(s) | Hd | s | π (%) | Voucher number* (no. of individuals) |
|---|---|---|---|---|---|---|---|---|---|---|
| *Aphyocypris normalis* | | | | | | | | | | |
| | 1 | Nanyang R. | 29 | 10 | h1(9), h2(1) | I | 0.2 | 4 | 0.073 | NTOUP201202-213(3), 214(7) |
| | 2 | Nanyang R. | 55 | 5 | h1(4), h3(1) | I | 0.4 | 1 | 0.037 | JL-N21(5) |
| | 3 | Salao R. | 35 | 2 | h1(2) | I | 0 | 0 | 0 | JL-N20(2) |
| | 4 | Wanquan R. | 122 | 10 | h4(10) | I | 0 | 0 | 0 | NTOUP201202-234 (10) |
| | 5 | Wanquan R. | 26 | 1 | h4(1) | I | - | - | - | JLF-38(1) |
| | 6 | Longto R. | 35 | 1 | h5(1) | I | - | - | - | NTOUP201202-256(1) |
| | 7 | Lingshui R. | 66 | 4 | h6(4) | I | 0 | 0 | 0 | NTOUP201006-398(1), JL-N22(3) |
| | 8 | Lingshui R. | 57 | 3 | h6(3) | I | 0 | 0 | 0 | NTOUP201202-406(2), 408(1) |
| | 9 | Ninyuan R. | 160 | 1 | h7(1) | III | - | - | - | NTOUP201006-400 (1) |
| | 10 | Wanlou R. | 41 | 5 | h8(5) | III | 0 | 0 | 0 | NTOUP201202-286(5) |
| | 11 | Jianfengshui R. | 52 | 1 | h9(1) | I | - | - | - | JL-N16(1) |
| | 12 | Changhua R. | 638 | 2 | h8(2) | III | 0 | 0 | 0 | JL-N15(2) |
| | 13 | Changhua R. | 257 | 1 | h8(1) | III | - | - | - | JL-N25(1) |
| | 14 | Changhua R. | 190 | 2 | h10(1), h11(1) | III | 1 | 2 | 0.183 | NTOUP201202-372(1), 377(1) |
| | 15 | Zhubi R. | 210 | 2 | h12(1), h13(1) | I (Ia, Ib) | 1 | 14 | 1.283 | NTOUP201202-332(2) |
| | 16 | Zhubi R. | 255 | 2 | h14(1), h15(1) | I (Ia) | 1 | 1 | 0.092 | NTOUP201004-070(2) |
| | 17 | Zhubi R. | 150 | 3 | h14(1), h16(1), h17(1) | I (Ia) | 1 | 3 | 0.183 | NTOUP201202-344(3) |
| | 18 | Beimen R. | 180 | 5 | h18(4), h19(1) | I (Ia) | 0.4 | 1 | 0.037 | NTOUP201202-326(5) |
| | 19 | Nandu R. | 257 | 2 | h20(1), h21(1) | I (Ib) | 1 | 3 | 0.275 | NTOUP201202-342(2) |
| | 20 | Nandu R. | 290 | 4 | h21(3), h22(1) | I (Ib) | 0.5 | 1 | 0.046 | NTOUP201006-413(1), JL-N20(3) |
| | 21 | Nandu R. | 293 | 4 | h23(1), h24(3) | I (Ib) | 0.5 | 1 | 0.046 | JL-N14 |
| | 22 | Nandu R. | 284 | 1 | h25(1) | I (Ib) | - | - | - | NTOUP201006-406(1) |
| | 23 | Nandu R. | 150 | 6 | h4(1), h26(3), h27(1), h28(1) | I, I (Ib) | 0.8 | 10 | 0.342 | NTOUP201006-388(1), JL-N19(5) |
| | 24 | Nandu R. | 170 | 5 | h29(5) | I | 0 | 0 | 0 | NTOUP201004-013(5) |
| | 25 | Nandu R. | 104 | 1 | h6(1) | I | - | - | - | NTOUP201004-011(1) |
| | A1 | Lou R. | 76 | 2 | h30(2) | II | 0 | 0 | 0 | JL-N10(2) |
| | A2 | Dong R. (Pearl R.) | 490 | 1 | h31(1) | II | - | - | - | NMNSF1744(1) |
| | A3 | Dong R. (Pearl R.) | 66 | 10 | h32(3), h33(2), h34(2), h35(1), h36(2) | I (Ic) | 0.867 | 7 | 0.295 | NMNSF1926(10) |
| | A4 | Pearl R. | 35 | 6 | h37(2), h38(1), h39(3) | I | 0.733 | 3 | 0.159 | JL-N28(3), N29(3) |
| | A5 | Xi R. (Pearl R.) | 47 | 7 | h4(1), h38(1), h40(4), h41(1) | I | 0.714 | 3 | 0.096 | NMNSF1734(7) |
| | A6 | Tan R. | 138 | 6 | h42(5), h43(1) | I | 0.333 | 5 | 0.153 | JL-N11(6) |
| | A7 | Nanliu R. | 100 | 10 | h44(7), h45(3) | I | 0.467 | 2 | 0.086 | JL-N13(10) |
| | A8 | Fangcheng R. | 54 | 2 | h46(1), h47(1) | I | 1 | 3 | 0.275 | JL-N09(2) |
| Total | | | | 127 | | | 0.96 | 88 | 0.689 | |
| *Aphyocypris moltrechti* | | | | | | | | | | |
| | B1 | Dajia R. | 463 | 2 | M1(2) | - | 0 | 0 | 0 | NMNSF262(1), 273(1) |
| | B2 | Wu R. | 560 | 10 | M2(10) | - | 0 | 0 | 0 | NMNSF2004(6), 2005(3), 2013(1) |
| | B3 | Zhoushui R. | 809 | 10 | M3(7), M4(3) | - | 0.467 | 5 | 0.214 | JLF-26(10)** |
| Total | | | | 22 | | | 0.697 | 9 | 0.384 | |

Sampling site numbers, river systems, altitudes of sampling sites, sample sizes (Ns), haplotypes, haplotype diversities (Hd), number of segregating sites (*s*), and nucleotide diversities (π) for specimens used in this study.

* NMNSF: Pisces collection of National Museum of Natural Science, Taichung (https://fishdb.sinica.edu.tw/eng/specimenlist.php?m=NMNS); NTOUP: Pisces collection of National Taiwan Ocean University, Keelung (https://digifish.biodiv.tw/); JL/JLF: uncatalogued specimens organized by the corresponding author; such as larvae, incomplete/broken samples which were excluded from the NMNS and NTOU collections, but useful for laboratory analyses.

** Specimens that have been treated as *Aphyocypris amnis*.

three of these specimens were collected from 25 localities within 12 river systems on Hainan Island, and 44 specimens were collected from 8 localities in five river systems in southern mainland China. We also sequenced 22 *A. moltrechti* specimens (accession codes: KC209958– KC209979) as outgroups (Table 1 and Fig 1). All of the samples used in this study are specimen-vouchered, coming from collections established in 2009–2012 as part of projects supported by the National Science Council, Taiwan (now Ministry of Science and Technology, Taiwan), or from the National Museum of Natural Science, Taiwan (Table 1).

## DNA amplification and sequencing

Genomic DNA was isolated using the Tissue and Cell Genomic DNA Purification Kit (Hopegen Biotechnology Development Enterprises, Taichung, Taiwan). The extraction of genomicDNA was performed according to the manufacturer's instructions with repeat membrane binding, salt washing, and centrifugation. Polymerase chain reaction (PCR) amplification was performed using the primer pair CypbF1 5′–TGACTTGAAGAACCACCGTTGTA–3′ and CypbR1 5′– CGATCTTCGGATTACAAGACCGATG –3′ [37] to target the 1,091 bp mtDNA cytochrome *b* (*cytb*) gene. Following an initial denaturation step at 95˚C for 2 min, the PCR comprised 35 cycles of denaturation (94˚C, 15 s), annealing (46˚C, 15 s), and extension (72˚C, 30 s), followed by a final extension for 10 min at 72˚C, on an MJ MINI PTC-1148C Personal Thermal Cycler (Bio-Rad, Mississauga, Canada) and using a PCR Master Mix Kit (Hopegen Biotechnology Development Enterprises, Taichung, Taiwan). Resulting PCR products were visualized through electrophoresis and purified using the HiYield Gel/PCR DNA Fragments Extraction Kit (RBC Bioscience, Taipei, Taiwan) prior to sequencing. Purified products were Sanger-sequenced using an ABI PRISM 3130xl Genetic Analyzer (Applied Biosystems, Foster City, CA, USA) and aligned using MUSCLE [38] and Molecular Evolutionary Genetics Analysis software (MEGA X) [39]. The output was later trimmedvisually.

## Phylogenetic analysis

Haplotype phylogenetic reconstructions, based on the resulting *cytb* sequences, were performed using neighbor-joining (NJ), maximum likelihood (ML), and Bayesian inference (BI) methods. The first two analyses were performed on unique haplotypes in PAUP* version 4 beta [40] with 1000 bootstrap replicates, and with HKY+I as the best-fitting model of sequence evolution, as determined by the AIC criterion in jModeltest0.1.1 [41]. Bootstrap branch support values were inferred from 1000 replicates using the NJ option for NJ and a heuristic search for ML. Bayesian analyses were performed in MrBayes 3.1.2 [42] with 2,000,000 generations, again with the HKY+I model as estimated in jModeltest per the BIC criterion. Replicates were used for nodal evaluation and sampling trees every 100 generations. Approximately 25% of sampling trees were discarded (as burn-in). A consensus tree was calculated using the remaining 15,000 trees (with log-likelihoods converged to stable values). Two separate runs with four Markov chains were performed. Trees were routed with the outgroup species, *A. moltrechti*. The genetic distances among *A. normalis* populations and the outgroup species were calculated using MEGA X with the Kimura 2-parameter model (K2P) [43]. Standard error estimates were obtained from bootstrapping over 1000 replicates.

Haplotype networks were inferred using TCS v1.21 [44]. A parsimony network was constructed using the statistical parsimony algorithm [45, 46] and haplotypes were grouped hierarchically into a set of nested clades [47].

A phylogenetic tree based on the novel *cytb* sequences from this study, plus sequences obtained from GenBank having a high degree of similarity to these at the nucleotide level (best BLAST hit more than 87% in similarity), was also inferred. This incorporates *cytb* sequences of

*A. arcus* (GenBank accession number AP011398) that were approximately 91–92% similar to *A. normalis* [48], *Mylopharyngodon piceus* (DQ026435; 88–89%) [49], *A. chinensis* (AB218688; 88%) [50, 51], *A. kikuchii* (JX184925; 88%) [51–53], *Squaliobarbus curriculus* (AF051877; 88%), *Hypophthalmichthys nobilis* (AF051866; 88%), *Oxygaster anomalura* (HQ009868; 88%) [54], *Xenocypris fangi* (AF036205; 87%) [55], *Ctenopharyngodon idella* (JN673556; 87%), and *Ischikauia steenackeri* (AB239601; 87%) [50]. This tree was constructed by using the Bayesian inference (BI) method with MrBayes 3.1.2 [42], using the HKY+I+G model estimated with jModeltest under the BIC criterion, and with 2,000,000 generations. A *cytb* segment of *Rasbora steineri* (JX843769) [56] was used as an outgroup in this tree.

## Divergence times among major clades

The divergence time of the most recent common ancestor (MRCA) of *A. normalis* specimens was estimated by a Bayesian analysis using BEAST ver. 1.7.4 [57]. We assumed the constant population size setting as a coalescent tree prior, which is applicable for trees reporting the relationships between individuals within the same species. Posterior distributions were obtained via Markov Chain Monte Carlo analysis with $2 \times 10^7$ steps (length of chain) sampling every 1000 steps under a fixed pairwise evolutionary rate with a strict clock model. We used previously reported molecular substitution rates estimated for cyprinid *cytb*, ranging from roughly 0.6% to 1.3% per million years (Myr; e.g., 0.76%, [58, 59]; 0.60–1.24%, [60]; 0.58–1.1%, [61]; 0.76–0.92%, [62]; 1.05%, [26]). We adopted 1.0% Myr as the mean value of a normally distributed clock rate covered by above range of *cytb* clock rate following Ketmaier et al. [63] and Zhang et al. [64]. The parameters of the substitution model GTR were estimated independently for each haplotype. Results were viewed using Tracer v 1.5 [65]. The effective sample sizes (ESSs) for all parameters exceeded 200. A low ESS (<200) means that the parameter contained a lot of correlated samples and thus may not represent the posterior distribution well.

## Population structure

The relative contributions of genetic variation within and between populations in *A. normalis* were tested via hierarchical analysis of molecular variance (AMOVA) [66] in Arlequin 3.5 [67], with significance assessed by using 1023 random permutations of the dataset. In this analysis, we focused on four *a priori* hypotheses we modeled that reflect different ways of partitioning populations to test the amount of variation explained between geographical units: isolated by the strait (*i.e.* Hainan Island vs. mainland; hypothesis 1a), drainage by river system, both on Hainan Island and on the mainland (hypothesis 1b), and two hypotheses of phylogeography within Hainan Island, as hypothesis 2a and hypothesis 2b explained below. For the hypothesis 1a, all sampled populations were combined into two groups representing the localities of their origin to evaluate movement across the strait. Alternatively, to evaluate movement between river systems (1b), we combined populations into 17 groups based on drainage basins or close geographic proximity (*i.e.*, the following populations were combined: sites 1–2, 4–5, 7–8, 12–14, 15–17, 19–25, and A2–A5; see Fig 1). If genetic structuring is driven by isolation among drainages due to drainage basin boundaries, then we expect to see high levels of structuring among drainages. For the hypotheses 2a and 2b, we evaluated movement between rivers on Hainan Island and checked the robustness of dividing the region into eight biogeographical areas based on climate [36] (hypothesis 2b; also see Discussion), compared with 12 groups based on drainage basins on Hainan Island (hypothesis 2a; S1 Fig shows the four hypotheses mentioned above). Cases where high percentages of variation occurred between groups suggest that an important historic barrier to gene flow exists, which is consistent with the relevant hypothesis [68, 69]. The $F_{ST}$ values were computed using the information of haplotype using Arlequin 3.5 as well.

In addition, we applied SAMOVA 2.0 to define groups of populations that are geographically homogeneous and maximally differentiated from each other, and to evaluate their hierarchical differentiation [70]. A total of 100 initial independent processes were tested, followed by 10,000 steps of the simulated annealing process, which maximizes the proportion of total genetic variance among groups. We set the population/sampling localities of a single river system as a grouping unit (17 counts), following hypothesis 1b in AMOVA analysis. SAMOVA analyses were run under hypotheses of two to 16 groups (K) without geographic restrictions to examine the grouping pattern and to explore if a larger number of groups existed. The $F_{CT}$ index (proportion of total genetic variance due to differences between groups of populations) was used to select the best grouping, that is, the most suitable K value. We selected the number of groupings that maximizes this component, meaning that under that hypothesis the groups of populations are maximally differentiated from each other [70].

## Results

### Sequence diversity and haplotype distribution

All sequences obtained from 127 *Aphyocypris normalis* and 22 *A. moltrechti* specimens were 1,091 bp in length. No insertions, deletions, or stop codons were found in these sequences. Among the sequences of *A. normalis*, 88 polymorphic sites (8.07%) were identified and 57 sites (5.22%) were parsimony-informative. The estimated Transition/Transversion ratio is 11.50. In total, 47 haplotypes were detected and are reported in Table 1, as well as their frequencies as represented by the number of individuals. Forty-one of them were private haplotypes. Within-population genetic variations of nucleotide diversity ($\pi$) of sampling locality populations, excluding those localities (sites 5, 6, 9, 11, 13, 25, and A1) where only one specimen was collected, ranged from 0−1.28%; the mean of all individuals was 0.69%. The haplotype diversity (Hd) within each locality ranged from 0.00−1.00 (Table 1). Most of the southeastern Hainan Island localities/rivers (sites 1−11) showed low haplotype diversities (0−0.4) and low nucleotide diversities (0−0.073%). In contrast, localities in southern mainland China showed higher haplotype (0−1) and nucleotide diversities (0−0.295%). Nucleotide diversity was highest at site 15 (1.283%) on the Zhubi River, which was represented by only two specimens.

On Hainan Island, 23 individuals sampled from the Nandu River exhibited 12 haplotypes, ten of which were exclusive to the river. These haplotypes could be separated into two groups. One belonged to Clade I (h4, h6, h26, h28, and h29), in which they were single or two nucleotide substitutions to each other. Another belonged to Subclade Ib (h20, h21, h22, h23, h24, h25, and h27), in which they are single or several substitutions (up to 11) to each other. The five individuals from the Changhua River carried three haplotypes (two exclusive) whereas 11 individuals from the Wanquan River carried only one, which was shared with the Nandu River and the Xi River (the Pearl River system) specimens in mainland China. Moreover, the 24 individuals from the Pearl River system in mainland China carried 12 haplotypes, 11 being exclusive to that stream; copious haplotypes were detected in a single population. Only four haplotypes (h1, h4, h6, h8) were present in more than three populations, with haplotype 4 being the most frequent by far. Haplotype 4 is also the only haplotype shared between Hainan Island and mainland China (sites 4, 5, 23, A5; Fig 1).

### Phylogenetic relationships

Bayesian, NJ, and ML haplotype trees (Fig 2, Bayesian tree with BI, NJ, and ML supporting values) showed three distinct clades among the *A. normalis cytb* haplotypes. Clades I and II, defined by 43 haplotypes from 116 specimens, represented almost all populations including those on Hainan Island and in southern mainland China, except those from the Ninyuan

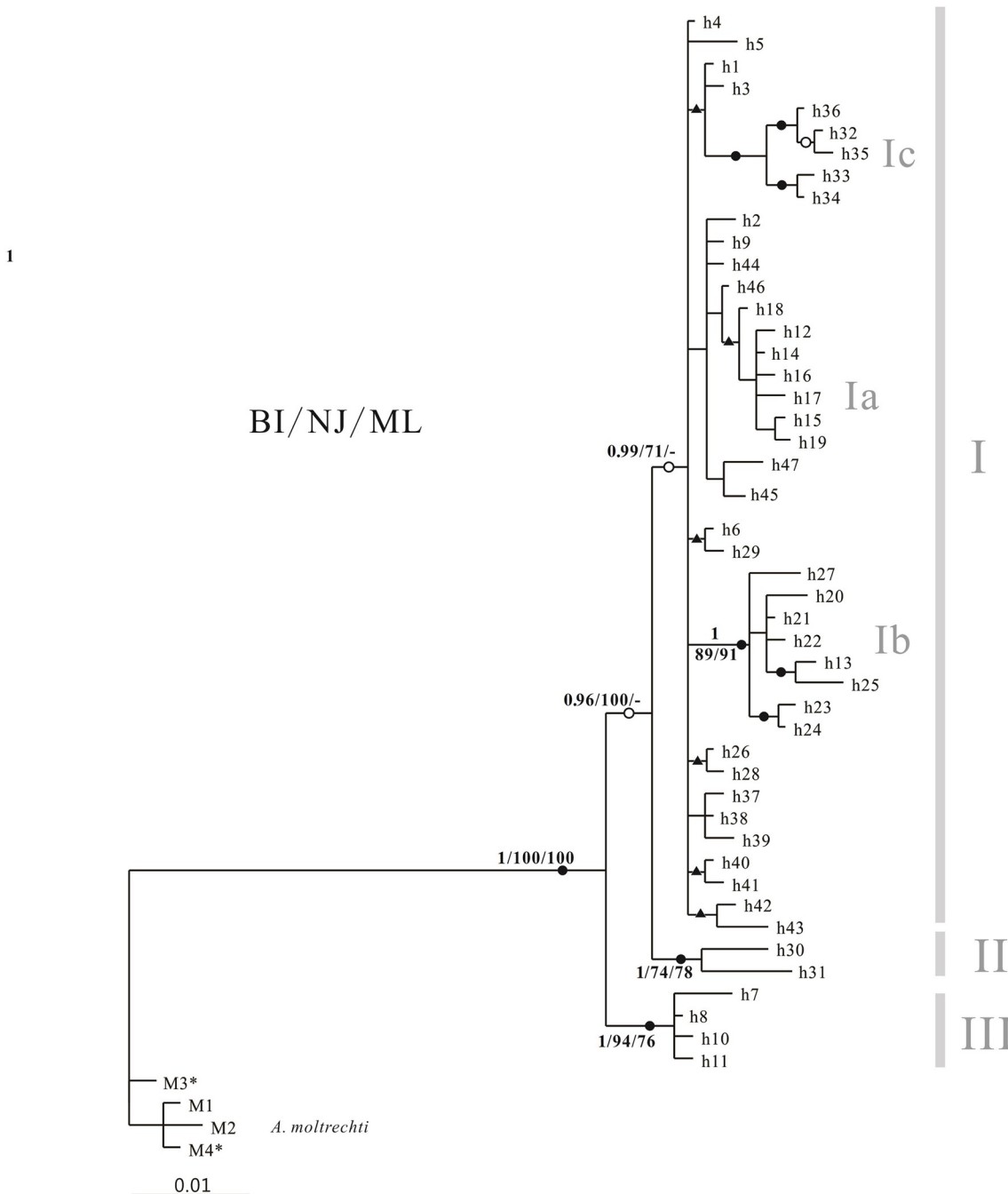

**Fig 2. Bayesian tree of 1091-bp-long haplotypes inferred from partial *cytb* gene.** Branch lengths are proportional to the scale given in nucleotide substitutions per site. Numbers at internal nodes are Bayesian inferences (only value >0.90 are shown) and bootstrap values (>70%) for NJ and ML analyses. Only the values for major clades or sub-clades are shown. Solid circles at internal nodes indicate they were supported by all three analyses; open circles indicate they were supported by Bayesian and NJ analyses; and solid triangles indicate they were supported by Bayesian analysis only. Abbreviations beside clades or sub-clades correspond to the clades listed in Table 1. * indicates the haplotypes from the Zhoushui River, which were assigned to *Aphyocypris amnis* by Liao et al. [71].

River (site 9), the Wanlou River (site 10), and the Changhua River (sites 12–14), which were occupied by Clade III with four haplotypes from 11 individuals. Clade II was represented by two haplotypes (h30, h31) from three individuals collected from the Lou River (site A1), which

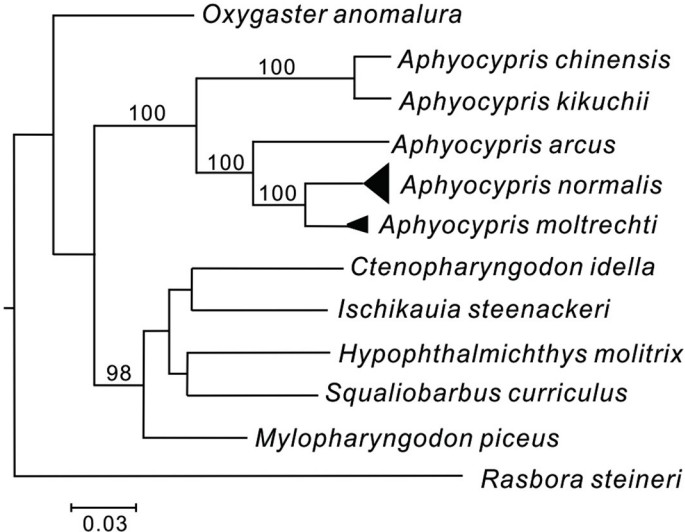

**Fig 3. Phylogenetic analysis of *Aphyocypris normalis* and its closely related species derived by the Bayesian method under the HKY+I+G model.** Numbers at internal nodes are Bayesian posterior probabilities and only value >90% are shown.

grouped with Clade I first and then Clade III. The Bayesian tree exhibited no discrepancies with the NJ and ML trees (S2 Fig) among these three major clades. Additionally, there are more subclades within Clade I: (1) Subclade Ia, represented by seven haplotypes (h12, h14, h15, h16, h17, h18, h19) from 11 individuals collected from the Zhubi River (sites 15–17) and the Beimen River (site 18) in Hainan; (2) Subclade Ib, represented by eight haplotypes (h13, h20, h21, h22, h23, h24, h25, h27) from 13 individuals collected from the Zhubi River (site 15) and the Nandu River (sites 19–23) in Hainan; and (3) Subclade Ic, represented by five haplotypes (h32–h36) from 10 individuals collected from the mainland's Dong River (site A3). These clades/subclades are generally supported by high bootstrap/supporting values inferred from Bayesian, NJ, and ML analyses (Fig 2).

Fig 3 shows the phylogenetic relationships between *A. normalis* and closely related cyprinid species. The sequences of *A. normalis* and *A. moltrechti* used in this study were tightly grouped, and the two together grouped with *Aphyocypris arcus* and other two species of *Aphyocypris* (*A. kikuchii* and *A. chinensis*). Like *A. normalis* and *A. moltrechti*, *A. kikuchii* and *A chinensis* also formed a close monophyletic group. All these species grouped together monophyletically at a high confidence level.

The average sequence difference among all *A. normalis cytb* haplotypes was 0.94 ± 0.13% (mean ± SD; range, 0.09–2.35%). Within the three major clades, the average haplotype differences were 0.75 ± 0.12% (Clade I), 1.2 ± 0.31% (Clade II), and 0.32 ± 0.12% (Clade III). The distances between major clades (Clade I, II and III) ranged from 1.53–1.82%, and those between subclades of Clade I ranged from 1.18–1.55% (Table 2).

## Haplotype network

In our haplotype network (Fig 4), two major clusters were identified (separated by the red broken line in Fig 4); only the smaller cluster of haplotypes h7, h8, h10, h11 was agreed with the

**Table 2. Estimates of evolutionary divergence (K2P) between sequence pairs between clades/sub-clades (below) and standard errors (above).**

|  | 1 | 2 | 3 | 4 | 5 | 6 | d×10² within clade/sub-clade |
|---|---|---|---|---|---|---|---|
| 1. I |  | - | - | - | 0.0003 | 0.0003 | 0.75 |
| 2. Ia | - |  | 0.0029 | 0.0032 | 0.0032 | 0.0038 | 0.19 |
| 3. Ib | - | 0.0118 |  | 0.0035 | 0.0034 | 0.0039 | 0.51 |
| 4. Ic | - | 0.0121 | 0.0155 |  | 0.0037 | 0.0040 | 0.35 |
| 5. II | 0.0153 | 0.0149 | 0.0183 | 0.0189 |  | 0.0036 | 1.20 |
| 6. III | 0.0164 | 0.0166 | 0.0200 | 0.0204 | 0.0182 |  | 0.32 |

Bayesian haplotype tree and ML as the Clade III (see Fig 2). Another two major clades (Clades I and II) identified by Bayesian, NJ, and ML methods were not monophyletic in the network result. Haplotypes of these two clades comprised the super-cluster of this network. In the super-cluster, the Clade I of phylogenetic tree contained at least 12 mutational differences from haplotypes of Clade II (h30 and h31). Haplotype 4 (h4) was the major haplotype (represented by 13 individuals, mainly from the Wanquan River), occupying the central position of all haplotypes, and occurred at three sampling locations on Hainan (sites 4, 5, 23) and a fourth

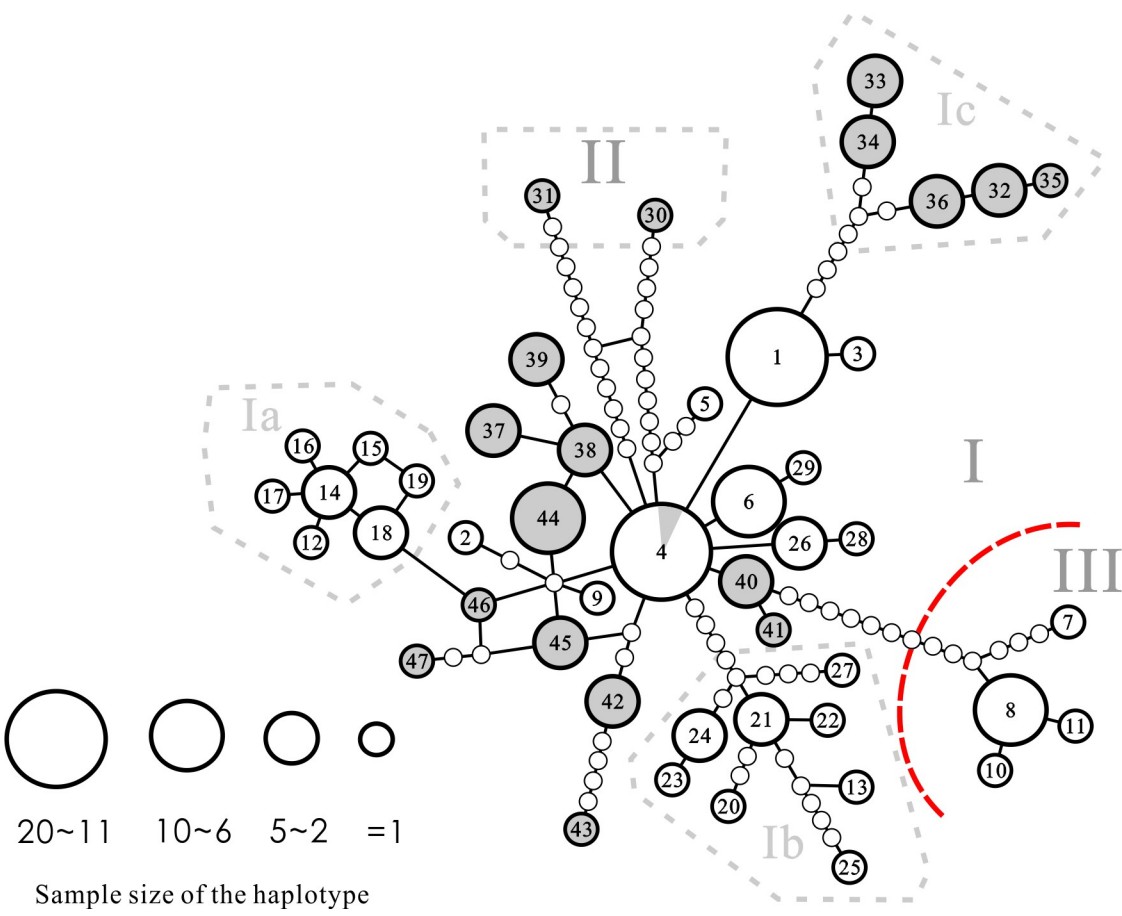

**Fig 4. Statistical parsimony haplotype network for haplotypes constructed by TCS 1.21.** Numbers inside circles are haplotype codes. Circle size reflects the number of individuals having the corresponding haplotype. Small solid dots signify possible missing haplotypes. A line connecting two haplotypes represents one nucleotide substitution. White circles: haplotypes collected from Hainan Island; grey circles: haplotypes collected from southern mainland China. The red broken line separates the two major clusters in this analysis.

**Table 3. Estimates of time since the most recent common ancestor (tMRCA; time scale in millions of years; standard errors are presented in parentheses) and 95% HPD estimates for clades/sub-clades for the main internal nodes of *Aphyocypris normalis* phylogeny (Fig 2) obtained using the *cytb* sequences based on a constant population model.**

|  | tMRCA(I+II) | tMRCA(Ia) | tMRCA(Ib) | tMRCA(Ic) | tMRCA(II) | tMRCA(I) | tMRCA(III) | tMRCA(ALL) |
|---|---|---|---|---|---|---|---|---|
| Mean (±SE) (Ma) | 0.832(0.0001) | 0.164(0.0001) | 0.330(0.0001) | 0.236(0.0001) | 0.514(0.0001) | 0.634(0.0011) | 0.309(0.0001) | 1.049 (0.0002) |
| 95% HPD lower (Ma) | 0.587 | 0.077 | 0.199 | 0.103 | 0.277 | 0.458 | 0.118 | 0.740 |
| 95% HPD upper (Ma) | 1.079 | 0.264 | 0.469 | 0.378 | 0.742 | 0.830 | 0.521 | 1.367 |

location on the mainland (site A5). Furthermore, another haplotype (h1) represented by 15 individuals detected in two small rivers of eastern Hainan (sites 1−3) contained only one mutational difference from h4. The haplotype h1 was located between h4 and five haplotypes of Subclade Ic, which represented by 10 individuals from a single site on the mainland (site A3). In the smaller cluster of the whole network (Clade III), the variation among the haplotypes was low with no more than six mutational differences. Three of the four Clade III haplotypes clustered closely with only one base variation between each other. The fourth haplotype (h7, represented by a single specimen from the Ninyuan River at site 9) contained five or six mutational differences relative to the other Clade III haplotypes.

## Divergence times among major clades

Divergence times among clades/subclades of *A. normalis* were estimated at 0.031−1.049 million years ago (Ma; Table 3). MRCA divergence time for each of the three major clades was estimated at 0.634 (95% HPD: 0.458−0.830; Clade I) Ma, 0.514 (0.277−0.742; Clade II) Ma, and 0.309 (0.118−0.521; Clade III) Ma. The divergence time for Clades I and II was estimated to be 0.832 (0.587−1.079) Ma. These results suggest that major-clade divergence events occurred in the Pleistocene.

## Population structure

For all of the *a priori* phylogeographic hypotheses that we examined, AMOVA recovered highly significant population structure ($F_{ST}$) values which implied genetic variation between tested groups. These models are summarized below. First, the Hainan Island vs. mainland hypothesis (Hypothesis 1a in Table 4) revealed that only 1.1% of the total genetic variation was

**Table 4. Results from the analysis of molecular variance (AMOVA) used to evaluate two *a priori* biogeographical hypotheses.**

| Hypothesis | Comparison (number of groups) | Source of variation (percentage) | | | F statistics | | |
|---|---|---|---|---|---|---|---|
| | | Among groups | Within groups, among populations | Within populations | $F_{CT}$ | Fsc | $F_{ST}$ |
| 1a | Hainan Island vs. mainland (2) | 1.1 | 78.35 | 20.54 | 0.01104 | 0.79229* | 0.79459* |
| 1b | Drainage by all river systems (17) | 32.77 | 47.01 | 20.21 | 0.32773* | 0.69930* | 0.79785* |
| 2a | Drainage by river systems in Hainan (12) | 51.14 | 36.34 | 12.52 | 0.51140* | 0.74371* | 0.87478* |
| 2b | Hypothetical biogeographic regions (8) | 70.15 | 17.51 | 12.34 | 0.70153* | 0.58652* | 0.87659* |

Significant structuring is shown as $F_{CT}$ values, where a value of '1' indicates complete isolation and '0' indicates no population difference. Here, the term 'groups' refers to the populations being compared under each hypothesis. The source of variation from among groups, within groups among populations, and within populations is given as a percentage for each comparison. $F_{SC}$ values correspond to within groups among populations; $F_{ST}$ provides the degree of genetic differentiation among the populations, that is, the proportion of the total genetic divergence that separate the populations. Values of Fst ranging from 0 to 1 were rated as Fst < 0.05 (insignificant or little genetic differentiation), 0.05 to 0.15 (moderate differentiation), 0.15 to 0.25 (moderate—large genetic differentiation), and Fst > 0.25 (immense/ pronounced levels of genetic differentiation) [72, 73].

*P < 0.01.

partitioned among the two regions, indicating a high level of genetic movement across the strait between Hainan Island and the mainland. We predicted that populations from both sides of Qiongzhou Strait and Beibu Bay would be genetically distinct, but this was not the case. Besides, the $F_{CT}$ values of this model was low (0.1104). Our subsequent hypothesis of this experiments (Hypothesis 1b) was designed to test the isolation power that independent river systems without freshwater connections would show lower levels of gene flow, and we found that this was more possible than hypothesis 1a, but also might not be true. The variations among groups and among populations within groups in Hypothesis 1b were similar as 32.77% and 47.01%, respectively. Second, we tested the isolation mechanism of *A. normalis* occurred on Hainan Island. In Hypothesis 2a, the variations among groups and among populations within groups for all river systems on Hainan Island are 51.14% and 36.34%, respectively. The results of Hypothesis 2a, as well as Hypothesis 1b, did not strongly support genetic isolation between river systems. The last hypothesis (2b), dividing the region into eight biogeographical areas based on climate revealed that most of the variation was between groups (70.15%; Table 4), which indicates that the hypothetical biogeographic region division is a strong barrier to gene flow.

In the SAMOVA analyses, the highest $F_{CT}$ value was observed at K = 3 (S1 Table). Under this K, 78.76% of the variation could be explained by variation among groups (Table 5). The identified groups were the Lou River population in mainland China (site A1; see Table 1 and Fig 1), populations of Changhua, Ninyuan and Wanlou Rivers in Hainan (sites 9, 10, and 12–14), and the remained populations both in mainland and Hainan Island (S3 Fig). These groups coincide with the results of the AMOVA that the strait between Hainan Island and the mainland does not form a barrier impeding the genetic interaction between them. The second highest $F_{CT}$ value in the SAMOVA analyses was observed at K = 4, which showed very close to the highest one (0.78714 vs. 0.78762; see S1 Table). In this four-groups hypothesis, the specimen collected from Ninyuan River (site 9) was separated from the populations of Changhua and Wanlou Rivers in advance, in which indicated that the specimen from Ninyuan River is unique to other specimens in this study.

In general, SAMOVA analyses detected a group (populations of Changhua, Ninyuan and Wanlou Rivers in Hainan) within Hainan populations found in AMOVA hypothesis 2b (S1C Fig), which showed a unique phylogenetic group different from all other *A. normalis* populations tested in this study (S3 Fig). This group was also identified as Clade III in the phylogenetic analysis mentioned above.

## Discussion

### Genetic diversity and structuring in *Aphyocypris normalis*

Our novel sequence data revealed three major genetically differentiated clades among the *A. normalis* specimens analyzed in this study, as supported by NJ, ML, and Bayesian analyses.

**Table 5. Sources of variation calculated with SAMOVA under K = 3.**

| Source of variation | d.f. | Sum of squares | Variance components | Percentage of variation | Fixation indices |
|---|---|---|---|---|---|
| Among groups | 2 | 437.383 | 17.62423 | 78.76 | $F_{SC}$ = 0.38251 |
| Within groups, among populations | 14 | 232.451 | 1.81779 | 8.12 | $Fst$ = 0.86886 |
| Within populations | 110 | 322.788 | 2.93444 | 13.11 | $Fct$ = 0.78762 |
| Total | 126 | 992.623 | 22.37646 | | |

*d.f.*, degrees of freedom

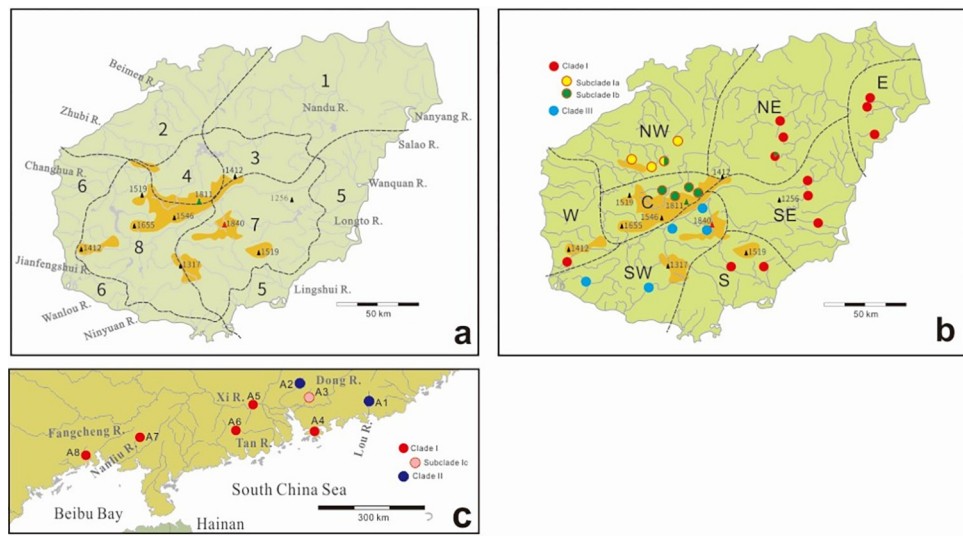

**Fig 5.** a, Climate ranges of Hainan Island, based on He and Zhang [36]; b, eight phylogeographic ranges inferred from this study's *cytb* dataset analyses, modified by climate characteristics; c, haplotype distribution in the continental area. Sampling site locations and haplotypes at each collection site are also marked. The yellow areas in Hainan maps (a, b) indicate mountainous regions more than 750 m ASL. Red triangle: Wuzhishan Mountain. Green triangle: Yinggeling Mountain.

Clades I and II occupy most of this species' habitat units, whereas Clade III occurs only in southwestern Hainan Island. The Clades I+II and Clade III groups occupy different ranges; however, no obvious geographic barriers exist between them.

It is generally accepted that geographical isolation is a very efficient isolation mechanism for freshwater animals; e.g., straits between islands and the mainland [74–77]. Nonetheless, concurring with the results of Huang et al. [25] and Li et al. [26], the geographic isolation of the Qiongzhou strait between Hainan Island and mainland China did not create efficient isolation barriers for *A. normalis* populations. Furthermore, Clade I was composed of specimens collected from 15 currently isolated rivers on Hainan Island and the mainland. Moreover, haplotypes h4, h6, and h8 all were found in separate rivers on Hainan Island and even across the strait (h4). Li et al. [26] reported that the Yinggeling and Wuzhishan Mountains (see Fig 5A, 5B) provides isolation mechanisms on *A. normalis* in Hainan and separated them into northeastern group and southwestern group. With more sampling sites in our work, we further proposed that Yinggeling Mountain formed a more efficiency isolation barrier than Wuzhishan Mountain, which might separate Subclade Ib and Subclade III. The pattern of lower genetic diversity in the populations of Hainan Islands is common with respect to the continent. Most of the Hainan populations showed low haplotype and nucleotide diversities, and populations in southern mainland China showed higher haplotype and nucleotide diversities in comparison, except the populations in Zhubi River (sites 15–17). The highest nucleotide diversity was detected at site 15 (1.283%) on the Zhubi River, as well as the haplotype diversity, however, which were represented by only two specimens. Limited sampling size may mislead the accuracy of analysis results, nevertheless, all the analyzed data presented in this study were based on valid specimens and are the only information of these populations at present.

Our comprehensive phylogenetic analyses, AMOVA and SAMOVA testing conclude that the phylogeography of *A. normalis* on Hainan Island was not shaped solely by the geographical barriers. Instead, effective isolation mechanisms could have been formed by events other than geographic barriers, such as the elevation of different habitats after climatic events and rainfall,

leading to the divergence of breeding strategies (e.g., [78]), or other water characteristics of lotic systems. It is therefore possible that the *A. normalis* population diversity of Hainan was shaped not only by geographic barriers or by habitat separation in different river systems, but also resulted from climatic events, such as floods, which provided opportunities for interaction between populations. He and Zhang [36] divided Hainan Island into eight climate regions primarily according to rainfall distribution: (1) NE seashore plain, more cool, windy but more wet; (2) NW seashore plain, more cool, windy but more dry; (3) NE hill district and basin, more cold and wet in winter; (4) NW mountain, hill district and basin, occasionally fierce cold winter; (5) SE seashore plain, more windy, more warm and wet; (6) SW seashore plain, more warm and fierce dry; (7) SE mountain and hill district, valley, and basin climate, more wet and warm; and (8) SW mountain and hill district basin, more warm and dry (Fig 5A). Their divisions fit the results of phylogeography of *A. normalis* on Hainan, as inferred from the *cytb* analyses of this study. Considering the climate divisions, topology of river systems, and the results of our *cytb* analyses, we propose eight phylogeographic ranges of *A. normalis* (Fig 5B): (1) eastern plain (E), including several small rivers, (2) northeastern hills and plains (NE), generally composed of hills and lowlands of the Nandu River system, (3) northwestern hills and lowlands (NW), including the northwestern lowland and catchment area of the Beimen and the Zhubi Rivers, (4) central mountain area (C), comprised of the catchment area upstream of the Nandu River of west–central Hainan Island, (5) southeastern hills and plains (SE), including the Wanquan and the Longto Rivers, (6) southern mountains and hills (S), mainly being the catchment area of the Lingshui River, (7) southwestern mountains and lowlands (SW), including the Ninyuan and the Wanlou Rivers and the watershed upstream of the Changhua River, and (8) western lowlands (W), including the Jianfengshui River and several lowland rivers on the west side of Hainan Island. The main difference between our phylogeographic ranges and He and Zhang's climate regions is that their division did not make allowances for the continuity of the river systems. However, the connections made by rivers are important interaction routes for freshwater animals. In this consideration, most river systems are catalogued separately except for areas upstream areas of the Nandu River (range C) and the Changhua River (range SW).

It is noteworthy that populations in upstream sections of the Nandu River formed a unique and highly supported phylogenetic subclade (Ib) in the central mountain area, the highest portion of the island. However, an individual identified with the haplotype characteristic of range C was found at a distant site within the upstream Nandu River system (range NE; also site 23 in Fig 1B). This implies that upstream Nandu River populations of *A. normalis* in ranges C and NE once interacted and later were separated by topographical events such as river capture.

Besides, in the upper reach of the Nandu River, a huge reservoir with a water surface area of approximately 144 km$^2$ and a backwater length of approximately 51 km was formed since 1970. The Songtao Reservoir is the largest body of water in Hainan, with a total reservoir capacity of $3.345 \times 10^9$ m$^3$ and covers 0.17 percent of this island [79]. In this study, *A. normalis* populations of sites 19–25 were separated by Songtao Reservoir into two parts: sites 19–22 in the upstream of Songtao Reservoir, and sites 23–25 in the downstream of it. Noteworthy, the genetic data were in accordance with this distribution pattern. The populations of sites 19–22 kept the haplotypes h21, h22, h23, h24 and h25, which belonged to Subclade Ib. On the other hand, the populations of sites 23–25 kept haplotypes h4, h6, h26, h27, h28 and h29, in which all belonged to Clade I but h27 (subclade Ib). A huge lentic system such as Songtao Reservoir can be an efficiency barrier for small-sized non- pelagic freshwater fish like *A. normalis*.

Subclade Ib is currently limited mainly to the west–central mountain area and shows clear differentiation between nearby populations in ranges NW and SW. Upstream Changhua River (range SW, haplotypes h8, h10 and h11) populations show a close relationship to individuals

collected from small rivers (i.e., the Ninyuan and Wanlou Rivers with haplotypes h7 and h8, respectively) in the southwestern seashore lowlands that run south–westward into the Southern China Sea. The specimens collected from these three rivers are grouped as a distinct monophyletic Clade III group (see Fig 2). There is no connection between the Changhua River and those small rivers presently, so the existence of Clade III provides powerful evidence that upstream reaches of the Changhua River once ran southwestward into the catchment area of the Ninyuan and Wanlou Rivers. A river capture event must have subsequently occurred to reroute the upstream portions of the Changhua River to flow northward as the Changhua River of today. Furthermore, the populations (those of Changhua, Ninyuan and Wanlou Rivers) in range SW are also grouped together by SAMOVA analysis (S3 Fig). They are possibly different from other populations and show a unique phylogeographic range on Hainan Island. However, because our inferences were based on mtDNA only, nuclear genome data should be examined for further information on population relationships and history.

It is widely accepted that the ichthyological fauna of Hainan Island show close relationships with that of the Pearl River System [17, 80, 81]. The isolation effect of the Qiongzhou Strait is not apparent based on current *cytb* sequence data. This study's two eastern–most collection sites on the mainland (A1 and A2) form Clade II, a sister clade of Clade I, and site A3 of the Pearl River also shows the distinct Subclade Ic. These populations were probably isolated by distance or other alternatives such as isolation by environment (the A2 and A3; for they are located in the large Pearl River System, same as A4, A5 and A6) or by vicariance (A1) from their ancestor population. The other populations of southern mainland China were all identified as members of Clade I and showed no obvious differentiation from parts of Hainan Island populations.

## Distribution processes of *Aphyocypris normalis* in Hainan and southern China with a comment on *Aphyocypris arcus* in Hainan

Hainan Island was formed by shifts in the location of regional land masses due to plate tectonics and subsequently separated by rising sea levels, and this island was connected to southern mainland China and northern Vietnam again by land bridge formation during the Last Glacial Maximum (LGM) around 19–26.5 kya (thousand years ago) while the sea level was around 80–100 m below the present day [82, 83]. With the advent of the Holocene warm period, Hainan again became isolated once more [84], with the Qiongzhou Strait having a minimum width of 20–40 km. These geographic events provided opportunities for animals to migrate across the strait and become isolated.

According to our divergence time estimation of the major clades of *A. normalis*, the order of phylogenetic isolation events began with (1) Clade III separated from all other populations due to separation within ancient river systems running southwestward at about 1.05 Ma, (2) Clade II separated from Clade I due to isolation by vicariance or by distance at about 0.83 Ma, followed by the formation of three subclades of Clade I, i.e., (3) Subclade Ia separated from other Clade I when its range was limited to the mountainous area of central Hainan at about 0.33 Ma, (4) Subclade Ic in site A3 separated from remaining Clade I due to isolation by distance (A3 is located in the large Pearl River System, same as A2, A4, A5 and A6 in which no efficient geographic barrier is shown between them) at about 0.24 Ma, and (5) populations of Subclade Ia separated from the remaining Clade I populations by being limited to rivers that run northwestward in Hainan. The common ancestor of *A. normalis* was presumably distributed widely in the lowlands of southern mainland China and Hainan Island. The major clades within *A. normalis* have diverged before the temporary land bridge that existed across the strait during the LGM. Moreover, with the stream capture and flood events [10, 25, 30], the river

systems within Hainan Island did not provide efficient isolation mechanisms similar to the strait, as there are shared haplotypes among geographically distant/ isolated river systems (e.g., haplotype h4 in the Wanquan River, sites 4 and 5; the Nandu River, site 23; and the Xi River, site A5. See Table 1). This research demonstrates that apparent vicariance events do not necessarily explain interregional population differentiation even in freshwater fish.

A unique phylogenetic group (Clade III) composed of populations of southwest Hainan was detected in this study (S2 Fig). It is possible that this group shares a close phylogenetic relationship with the populations in the Indochina Peninsula if *A. normalis* occurred there [19]. It would be helpful to analyze specimens collected from the Indochina Peninsula in future, both to clarify the migratory history of this species and to better understand the natural history of terrestrial fauna near the South China Sea.

*Aphyocypris normalis* and *A. arcus* are very similar, both in morphology [85, 86] and phylogeny [48]. The taxonomic confusion between these two species has been the subject of ichthyologists' debate for decades. Du et al. [87] described the differences between *A. normalis* and *A. arcus* by comparing the morphology and anatomy of specimens collected from both southern mainland China and Hainan Island. They provided a morphological key for identifying these two species and claimed that they have sympatric distributions on Hainan Island. However, we did not detect any haplotype belonging to *A. arcus* from our Hainan Island specimens. We suggest a high possibility that *A. arcus* does not naturally occur on Hainan Island based on the DNA analysis result in this study. More comprehensive comparative studies of *A. arcus* both on DNA and morphology are therefore necessary in the future.

## Comment on the genetic diversity of *Aphyocypris moltrechti* in Taiwan

We used *Aphyocypris moltrechti* as the comparison outgroup in this study. We used 22 museum specimens from three rivers in central Taiwan, including two from the Dajia River (site B1 in Fig 1, collected in 2003), ten from the Wu River (B2, 2008), and ten from the Zhoushui River (B3, 2011), which latter were assigned to a newly described species, *Aphyocypris amnis* Liao, Kullander and Lin [71]. Only four haplotypes were identified from 22 sequences that were 1,091 bp long and only one haplotype was found in the Dajia and Wu Rivers' population (Table 1). The K2P distances among haplotypes of *Aphyocypris moltrechti* (only the specimens from the Dajia R. and Wu R.) and *A. amnis* (the specimens from the Zhoushui R.) were estimated to be 0.0018–0.0065 (2–7 bp variation) and 0.0035 on average within all samples of these two groups (n = 22). For the specimens collected from the Zhoushui River, two haplotypes were identified that had 5 bp of variation between them (K2P distances = 0.0046) and 0.00215 within the population (n = 10). These results disclosed small nucleotide diversity among *A. moltrechti* specimens used in this study and are similar to another report that focused on population structure using complete *cytb* sequences (1,140 bp) and microsatellite DNA data [88], which revealed extremely low levels of intra-population polymorphism for the four populations in three rivers (same as this study) of *A. moltrechti* on nucleotide diversity of mtDNA analyses, though they concluded that the Zhoushui River population (assigned as a new species *A. amnis* later in the same year 2011) possesses unique haplotypes and should be handled as a management unit different from other populations. Comparing with the *cytb* genetic diversities among *A. moltrechti* populations in this study, the variation between *A. moltrechti* and *A. amnis* is comparatively small. The validity of *A. amnis* should be reconsidered carefully due to its comparatively low genetic divergence in comparison to *A. moltrechti* populations, as well as their ambiguous morphologic difference. Since the mtDNA is of maternal origin, the conclusion that comes with using only mitochondrial data may differ from phylogenies at the species level or higher. We suggest that more appropriate

nuclear markers, or reduced representation genome-wide markers should be used to evaluate the validity of *A. amnis* in the future.

## Supporting information

**S1 Table. $F_{CT}$ values for different numbers of population groups (K) inferred by the SAMOVA.** The yellow column indicates the K value with highest $F_{CT}$ value (K = 3). (XLSX)

**S1 Fig.** The four tested hypotheses via hierarchical analysis of molecular variance (AMOVA) in this study: **a**, isolated by the strait (hypothesis 1a); **b**, drainage by river system, both on Hainan Island and on the mainland (hypothesis 1b); **c**, eight biogeographical areas on Hainan Island modified from climate pattern [36] (hypothesis 2b); **d**, twelve groups model based on drainage basins on Hainan Island (hypothesis 2a). (TIF)

**S2 Fig. Phylogenetic analysis of *Aphyocypris normalis* and its sister species derived by NJ and ML methods.** Numbers at internal nodes are bootstrap values and only value >70% are shown. (TIF)

**S3 Fig. The grouping of *Aphyocypris normalis* based on the results of SAMOVA analysis (K = 3 with highest $F_{CT}$ value; see S1 Table).** (TIF)

## Acknowledgments

We would like to express our sincerest thanks to Ming-Fong Yeh, Kuang-Te Chen, and Shih-Pin Huang for their assistance to access and prepare the specimen collections for this study; thanks to Chung- Hao Juan for his help in doing the SAMOVA analysis. We are grateful to Una Chou and Graham L. Banes for proof- reading the manuscript. The fish samples used in this study were collected and vouchered with support from the National Science Council, Taiwan, as part of projects NSC98-2631-H019-001, NSC99-2631-H019-001 and NSC 100-2631-H-019-001. Finally, we wish to express our deeply appreciation to the anonymous reviewers for their careful reading of our manuscript and their many insightful comments and suggestions.

## Author Contributions

**Data curation:** Nian-Hong Jang-Liaw.

**Funding acquisition:** I-Shiung Chen.

**Investigation:** I-Shiung Chen, Nian-Hong Jang-Liaw.

**Methodology:** I-Shiung Chen, Nian-Hong Jang-Liaw.

**Project administration:** I-Shiung Chen.

**Resources:** I-Shiung Chen, Nian-Hong Jang-Liaw.

**Writing – original draft:** Nian-Hong Jang-Liaw.

**Writing – review & editing:** I-Shiung Chen, Nian-Hong Jang-Liaw.

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
