## [Decision Letter · Decision Letter 0]

12 Jan 2021

PONE-D-20-37420

Phylogeography of Aphyocypris normalis Nichols and Pope, 1927 at Hainan Island and adjacent areas based on mitochondrial DNA data

PLOS ONE

Dear Dr. Jang-Liaw,

Thank you for submitting your manuscript to PLOS ONE. After careful consideration, we feel that it has merit but does not fully meet PLOS ONE’s publication criteria as it currently stands. Therefore, we invite you to submit a revised version of the manuscript that addresses the points raised during the review process.

We look forward to receiving your revised manuscript.

Kind regards,

Tzen-Yuh Chiang

Academic Editor

PLOS ONE

Journal Requirements:

2. We note that Figure 1 and 5 in your submission contain map images which may be copyrighted. All PLOS content is published under the Creative Commons Attribution License (CC BY 4.0), which means that the manuscript, images, and Supporting Information files will be freely available online, and any third party is permitted to access, download, copy, distribute, and use these materials in any way, even commercially, with proper attribution. For these reasons, we cannot publish previously copyrighted maps or satellite images created using proprietary data, such as Google software (Google Maps, Street View, and Earth). For more information, see our copyright guidelines: http://journals.plos.org/plosone/s/licenses-and-copyright.

(1) You may seek permission from the original copyright holder of Figures 1 and 5 to publish the content specifically under the CC BY 4.0 license. 

Reviewers' comments:

Reviewer's Responses to Questions

**Comments to the Author**

1. Is the manuscript technically sound, and do the data support the conclusions?

Reviewer #1: Partly

2. Has the statistical analysis been performed appropriately and rigorously? 

Reviewer #1: No

3. Have the authors made all data underlying the findings in their manuscript fully available?

Reviewer #1: Yes

4. Is the manuscript presented in an intelligible fashion and written in standard English?

Reviewer #1: No

5. Review Comments to the Author

Reviewer #1: This manuscript by IS Chen and NH Jang-Liaw reports the mtDNA phylogeography of a cyprinid fish, Aphyocypris normalis, commonly distributed in the southern part of China including Hainan Island, based on an extensive geographical sampling. This seems to be the first report of “detailed” phylogeography and population structuring of freshwater fish within Hainan Island, and will contribute to our understanding of geographical formation of fish fauna under the climate (sea level) oscillation in Asian regions. On the basis of the potential importance of this work, it is worth to be published.

However, unfortunately I should say that the present form of the manuscript is still immature and includes several points to be improved much. I would be happy if my comments will help the authors improve their manuscript for its publication.

Major points:

(1) The scope, originality, and emphasis of this paper should be clarified and limited.

First of all, the authors must recognize and adequately value the closely relevant work, Huang et al. (2019: ZooKeys) [ref.21], which studied on the same species and geographic region and the similar topics with longer mtDNA sequences, but with lower sampling density. The previous study has already reached the same conclusions as parts of the present study. They should not be fully repeated in this paper. Huang et al. (2019) has already revealed the wide range distribution of a part of haplotypes across the Qiongzhou (Hainan) Strait and the interspecific relationships. I think the detailed population structure should be the main, original results in the present study.

Also, extra parts apart from the main purposes are found in Discussion; i.e., parts about the nomenclature of genera (L393–), Aphyocypris arcus (L522), and Aphyocypris moltrechti in Taiwan. I understand the authors’ mind, but they are not the discussion from this work.

(2) Some recent references relevant to this work should be referred to and cited.

I believe some recent references as exemplified below should be cited, and some data should be updated.

Kang et al. (2014) Mapping China’s freshwater fishes: diversity and biogeography. Fish Fish 15: 209–230. https://doi.org/10.1111/faf.12011

Xing et al. (2016) Freshwater fishes of China: species richness, endemism, threatened species and conservation. Diversity Distrib 22:358–370. https://doi.org/10.1111/ddi.12399

Xiong et al. (2018) Factors influencing tropical Island freshwater fishes: species, status, threats and conservation in Hainan Island. Knowl Manag Aquat Ecosyst 419:6. https://doi.org/10.1051/kmae/2017054

He et al. (2020) Diversity, pattern and ecological drivers of freshwater fish in China and adjacent areas. Rev Fish Biol Fisheries 30:387–404. https://doi.org/10.1007/s11160-020-09600-4

(3) The examination of population divisions using AMOVA should be more clearly explained in Methods (L182–) and Results (L302–). Quite complicated. Different (contrasted) models should be shown clearly in text, or, as Tables or Figure.

Also, why do not you consider the use of SAMOVA?

(4) Nested clade analysis is not actually used here and also should not be used.

There is a section titled “Nested clade analysis and variations among clades” in Results (L351–). However, the authors do not actually conduct this analysis. The nested clade analysis (NCPA) is not only a nesting method of a haplotype network but a series of procedures to identify past demographic events by calculating some summary statistics and inferring histories of clades based on a set of criteria (see, e.g., Templeton 1998: Mol Ecol 7:381–397). This method is presently not recommended to be used. See Panchal and Beaumont (2010: Syst Biol 59:415–432) and the cited references on debates on NCPA. The section is inadequate in double meaning.

(5) Limitation of the conclusions due to use of mtDNA (effectively one “gene”) should be adequately mentioned in Discussion.

Medium problems:

(6) L36, “The relationships of A. normalis between Hainan and main land China populations are also discussed in this report.”: Readers want to know the contents of the discussion.

(7) L177, “the Kimura 2-parameter model (K2P)”: Why do not the authors use the best model selected by jModeltest? Should validate the use of the model.

(8) L220, “under a fixed clock pairwise evolutionary rate of 1.0 %/Myr following Ketmaier et al. [54] and Zhang et al. [55] with a strict clock model.”: Explain why the fixed clock model and the evolutionary rate were adopted. The results of divergence time are basically determined by the latter value. A range of estimates of the cytb evolutionary rate has been reported in literature.

(9) L240–: The diversity indices should be compared only among populations with not very few specimens. I do not think that the values for those with one, two, or three specimens are meaningful.

(10) L253, “A high number of haplotypes is characteristic of one and only populations.”: I cannot understand this sentence.

(11) L291–: Move the explanation of methods to “Methods”.

Also, what is “Noncoding” (L294) here? The data consist of only cytb coding region, don’t it?

(12) L298–302, Is this part replicated with that from L240? I cannot understand the relationship between these parts.

(13) L324–, “suggesting that the differences in climates among biogeographic regions constitute important barriers to gene flow”: Not fully logical. Even if the boundary is based on climates, it does not mean that the gene flow was prevented due to the differences in climates.

(14) L382–, “0.6344±0.0011 Ma (mean ±s.e.)...”: Should show 95% HPD instead of standard error for the estimate. Estimation errors of the divergence time are represented by the former.

(15) L395–: Readers cannot find any information what species once belonged to those synonymous genera solely in this paper. Also, as mentioned above (1), this part is not the conclusion from this study (Huang et al. 2019 have already shown the tree), but just taxonomic comments based on the phylogenetic tree. Nevertheless, the paragraph includes a nomenclatural statement. It should be valuable but be done elsewhere.

(16) L423–: The comparison of the results of AMOVAs only indicates that genetic diversity that cannot be explained by drainage differences is still large. This does not mean that the drainage differences do not contribute to geographical isolation. In fact, an extent of geographic cohesion of haplotypes are found in Figure 5. The expression in L434 (not only by...) seems fine.

(17) L429–: Is there no possibility that the distribution of haplotypes was partly caused (disturbed) by artificial transportation?

(18) L437–: Figure 5 (especially 5b) should be placed in Results because of easy recognition of the distribution of haplotypes (clades).

(19) L510, “except for the west central mountains southwestern mountains, and lowlands of Hainan Island.”: From where does this inference come?

(20) L513, “the river systems within Hainan Island did not create efficient isolation barriers similar to the strait”: This should be explained only by the short period after the last isolation. The drainage and strait undoubtedly work as barriers for strictly freshwater fish.

(21) L516–, “this research demonstrates that vicariant events have not yet generated interregional population differentiation in P. normalis.”: Same as above.

(23) L533–561: As mentioned in (1), this part is out of range of this paper. It should be worth to discuss but be done elsewhere.

Minor/Trivial points:

(22) L27, “based on molecular data and modifications based on climate patterns”: Difficult to understand here in Abstract.

(23) L32, “(W).The”, and other places (L276...): Insert a space between two sentences.

(24) L36, “main land”  “mainland”

(25) L40, “It is”: What is “It”?

(26) L43, “how factors such as population structure ...”: Is the “population structure” a factor? I do not think so.

(27) L125, “were”  “are”

(28) L173, “20,000”  “15,000”?

(29) L212, legend of Fig. 3, “HKY+I+C”  “HKY+I+G”?

(30) L219, “cytb”: Italicized.

(31) L267, “the Lou River”: Where is it?

(32) L290–: A too long paragraph. Change the paragraph at least at L302 (For...).

(33) L295, “Among”  “Within”? Also, the correspondence between clades and average haplotype differences values should be clarified.

(34) L299, “among sampling locality populations”: What is this?

(35) L320, “the last model”: What is this?

(36) L343–, “and other two species of Aphyocypris. Like A. normalis and A. moltrechti, the two Aphyocypris species”: Show the species names of the two species.

(37) L352 and other places: TCS is the name of a computer software, not a name of method or algorithm.

(38) L381, “MCVRA”: What is this?

(39) L385, “in the middle to late Pleistocene”: The range (at least of means) does not seem to include the late Pleistocene.

(40) L419, “is”  “were”

(41) L462, “Subclade...”: Change the paragraph.

(42) L487, “no obvious differentiation from Hainan Island populations.”  “... from parts of Hainan Island populations”?

(43) L498, “These geographic events”: Not very precise. Sea level rising in the Holocene did not provide the opportunity.

(44) L504, L506, “Subclade Ia separated from other Clade I populations”, “Subclade Ic separated from remaining Clade I populations”: A clade cannot separate from a population.

(45) L511, “Clades”  “The major clades”?

(46) Fig. 1: In the panel a, “b” and “c” in white are less visible.

(47) Fig. 5: Indicate names of major river systems.

(48) Table 2: “evolutionary divergence”: Should show which index was used.

--- end of comments

6. PLOS authors have the option to publish the peer review history of their article (what does this mean?). If published, this will include your full peer review and any attached files.

Reviewer #1: No

---

## [Author Response · Author response to Decision Letter 0]

25 Feb 2021

We appreciate the reviewers’ comments and suggestions. The manuscript has been revised accordingly. The following are our answers to the comments: 

Comments of the Editor

2. We note that Figure 1 and 5 in your submission contain map images which may be copyrighted. All PLOS content is published under the Creative Commons Attribution License (CC BY 4.0), which means that the manuscript, images, and Supporting Information files will be freely available online, and any third party is permitted to access, download, copy, distribute, and use these materials in any way, even commercially, with proper attribution. For these reasons, we cannot publish previously copyrighted maps or satellite images created using proprietary data, such as Google software (Google Maps, Street View, and Earth). For more information, see our copyright guidelines: http://journals.plos.org/plosone/s/ licenses-and-copyright.

OUR REPLY—All the figures in this manuscript were made by the corresponding author. The Figure 1a was made under the assistant of a web-based soft SimpleMappr first, then added other information such as collecting sites using CorelDraw. As the copyright announcement of SimpleMappr in https://www. simplemappr.net/# tabs=6 , “All versions of SimpleMappr map data found on this website are in the Public Domain. You may use the maps in any manner, including modifying the content and design, electronic dissemination, and offset printing. The primary author, David P. Shorthouse has waived all copyright, related or neighboring rights, and financial claim to the maps and invites you to use them for personal, educational, and commercial purposes. No permission is needed to use SimpleMappr….”, the corresponding author is sure that Figure 1a is not copyrighted image to be published in PLOS ONE, as well as Figure 1b, c and 5. They were made using CorelDraw without any copyrighted materials on the final printed version of them. 

Comments to the Author

(1) The scope, originality, and emphasis of this paper should be clarified and limited. First of all, the authors must recognize and adequately value the closely relevant work, Huang et al. (2019: ZooKeys) [ref.21], which studied on the same species and geographic region and the similar topics with longer mtDNA sequences, but with lower sampling density. The previous study has already reached the same conclusions as parts of the present study. They should not be fully repeated in this paper. Huang et al. (2019) has already revealed the wide range distribution of a part of haplotypes across the Qiongzhou (Hainan) Strait and the interspecific relationships. I think the detailed population structure should be the main, original results in the present study. Also, extra parts apart from the main purposes are found in Discussion; i.e., parts about the nomenclature of genera (L393–), Aphyocypris arcus (L522), and Aphyocypris moltrechti in Taiwan. I understand the authors’ mind, but they are not the discussion from this work. 

OUR REPLY—Our work is very similar to Huang et al. (2019), both in topics and main results. However, the sampling scales and locality size are very different between these works. The river systems in Hainan are so complex which are difficult to study with single sampling site for each river system. Anyway, the published work should be respect and we impressed their contribution by point out their achievement in introduction part and citing them in this work. 

In this manuscript we talk about some taxonomic confusion beside the phylogeography of Aphyocypris normalis, which were proposed in Liao et al. (2011). We agree that another comprehensive study(ies) on both morphological and genetic views are necessary to clarify the mess occurred among these cyprinid genera, maybe not in this work. We removed the first section of the DISCUSSION to simplify the scope of this work. 

For the discussion on Aphyocypris moltrechti in Taiwan, we wish to keep this section in this study. The validity of A. amnis in Taiwan bothers ichthyologists and fieldworkers a lot. We pointed out the small variation between A. moltrechti and A. amnis in population level by using them as outgroup of this study; We think it is important and a useable contribution to ichthyologic fauna research of SE Asia, beside the main results of biogeography of this study. We sincerely wish to keep this section to reminder the ones who will concern this issue that the validity of A. amnis need to be discussed more carefully in the future. 

(2) Some recent references relevant to this work should be referred to and cited.I believe some recent references as exemplified below should be cited, and some data should be updated.

Kang et al. (2014) Mapping China’s freshwater fishes: diversity and biogeography. Fish Fish 15: 209–230. https://doi.org/10.1111/faf.12011

Xing et al. (2016) Freshwater fishes of China: species richness, endemism, threatened species and conservation. Diversity Distrib 22:358–370. https://doi.org/10.1111/ddi.12399

Xiong et al. (2018) Factors influencing tropical Island freshwater fishes: species, status, threats and conservation in Hainan Island. Knowl Manag Aquat Ecosyst 419:6. https://doi.org/10.1051/kmae/2017054

He et al. (2020) Diversity, pattern and ecological drivers of freshwater fish in China and adjacent areas. Rev Fish Biol Fisheries 30:387–404. https://doi.org/10.1007/s11160-020-09600-4

 OUR REPLY—Many thanks for your useful information. We update our manuscript with the recent references provided by the you. 

(3) The examination of population divisions using AMOVA should be more clearly explained in Methods (L182–) and Results (L302–). Quite complicated. Different (contrasted) models should be shown clearly in text, or, as Tables or Figure.

Also, why do not you consider the use of SAMOVA? 

 OUR REPLY—We modified the description of method (L185 to 206) and result (line 321 to 345) of AMOVA to clarify them more. Besides, we tried to use SAMOVA following reviewer’s suggestion, too. The results of SAMOVA were informative, it agreed that the Qiongzhou Strait might not be an effective isolation mechanism for genetic interaction of both side as AMOVA results and that from Huang et al. (2019). However, the user cannot group the populations (to calculate K value in different scenarios following the hypotheses we proposed) by themselves as we know. Which makes SAMOVA not suitable to compare the phylogeography of Aphyocypris normalis and the climate pattern (like hypothesis 4 of the AMOVA test; in which we need to group the analyzed units by ourselves) of Hainan in this study. We decided to keep both AMOVA and SAMOVA results in the revised manuscript. 

(4) Nested clade analysis is not actually used here and also should not be used.

There is a section titled “Nested clade analysis and variations among clades” in Results (L351–). However, the authors do not actually conduct this analysis. The nested clade analysis (NCPA) is not only a nesting method of a haplotype network but a series of procedures to identify past demographic events by calculating some summary statistics and inferring histories of clades based on a set of criteria (see, e.g., Templeton 1998: Mol Ecol 7:381–397). This method is presently not recommended to be used. See Panchal and Beaumont (2010: Syst Biol 59:415–432) and the cited references on debates on NCPA. The section is inadequate in double meaning. 

 OUR REPLY—We agree with your point. In the earlier version of this manuscript, we applied ANeCA v1.2 (Posada, 2008) to carry out NCPA to construct the dispersal pattern scenario based on a set of criteria provided by the latest inference key available at http://darwin. uvigo.es/download/geodisKey_06Jan11.pdf). We felt something wrong on the results and gave up the inferred demographic history part but keep the haplotype network result as “NCA” in the manuscript. However, as you mentioned before, “NCA” is somewhat confused with “NCPA”; now we use the term “haplotype network” to replace “NCA” to avoid confusion.

(5) Limitation of the conclusions due to use of mtDNA (effectively one “gene”) should be adequately mentioned in Discussion.

OUR REPLY—We mentioned it in the last sentences of conclusion of this revised manuscript. 

(6) L36, “The relationships of A. normalis between Hainan and main land China populations are also discussed in this report.”: Readers want to know the contents of the discussion.

OUR REPLY—We present it in “Genetic diversity and structuring in Aphyocypris normalis” section of DISCUSSION.

(7) L177, “the Kimura 2-parameter model (K2P)”: Why do not the authors use the best model selected by jModeltest? Should validate the use of the model.

 OUR REPLY—The best fitting model of sequence evolution was HKY+I in this work. Unfortunately, the MEGA 5 used in this study does not include HKY model, as well as the latest version MEGAX. The Kimura two-parameter (K2P) model (Kimura 1980) is probably the most widely used of all models of nucleotide substitution for estimating genetic distances and phylogenetic relationships. We know the reason why the K2P model is popular in evolutionary studies is not because the K2P model is the most precise model, but probably either because many authors have used it, or because it is the default of various platforms for phylogenetic analyses. However, we finally decided to apply the K2P model for its high availability in most phylogeny analysis tools.

(8) L220, “under a fixed clock pairwise evolutionary rate of 1.0 %/Myr following Ketmaier et al. [54] and Zhang et al. [55] with a strict clock model.”: Explain why the fixed clock model and the evolutionary rate were adopted. The results of divergence time are basically determined by the latter value. A range of estimates of the cytb evolutionary rate has been reported in literature.

OUR REPLY—Our data is a simple single mtDNA gene dataset. In general, the strict clock is used in analyses of sequence data sampled at the intraspeciﬁc level, for which there is an expectation of low rate variation among populations. This is because many of the factors associated with lineage effects, such as differences in generation time and DNA repair mechanisms, are unlikely to show measurable variation among conspeciﬁc individuals. We decided to use the fixed clock model with the fixed evolutionary rate following Ketmaier et al. [54] and Zhang et al. [55] to simplify our analysis from too complex to discuss.

(9) L240–: The diversity indices should be compared only among populations with not very few specimens. I do not think that the values for those with one, two, or three specimens are meaningful.

OUR REPLY—We understand the view you pointed out. Practically, the sample collecting in field is not always easy, especially target on some special species. We wish to use every specimen as possible as we could for do not let them be scarified in vain. Table 1 displays the detailed information of the used specimen, which reveals the collection details by separate collection sites, even those with small sampling size. It is always important to clarify the sampling information for phylogeography studies we believe. We present the genetic distances between clades/subclades in Table 2. 

(10) L253, “A high number of haplotypes is characteristic of one and only populations.”: I cannot understand this sentence.

 OUR REPLY—We modified this sentence to “Many haplotypes were detected in one single population.”

(11) L291–: Move the explanation of methods to “Methods”.

Also, what is “Noncoding” (L294) here? The data consist of only cytb coding region, don’t it?

 OUR REPLY—We removed them to “Data analysis” section of Material and Methods and deleted the mistake statement of “Noncoding…”.

(12) L298–302, Is this part replicated with that from L240? I cannot understand the relationship between these parts.

 OUR REPLY—We moved the sentences L298-302 behind L270.

(13) L324–, “suggesting that the differences in climates among biogeographic regions constitute important barriers to gene flow”: Not fully logical. Even if the boundary is based on climates, it does not mean that the gene flow was prevented due to the differences in climates.

 OUR REPLY—We modified it to “…the differences in climates among biogeographic regions constitute effective isolation mechanisms to gene flow”

(14) L382–, “0.6344±0.0011 Ma (mean ±s.e.)...”: Should show 95% HPD instead of standard error for the estimate. Estimation errors of the divergence time are represented by the former. 

 OUR REPLY—We modified the description following your suggestion.

(15) L395–: Readers cannot find any information what species once belonged to those synonymous genera solely in this paper. Also, as mentioned above (1), this part is not the conclusion from this study (Huang et al. 2019 have already shown the tree), but just taxonomic comments based on the phylogenetic tree. Nevertheless, the paragraph includes a nomenclatural statement. It should be valuable but be done elsewhere.

 OUR REPLY—We removed this section following your comment. 

(16) L423–: The comparison of the results of AMOVAs only indicates that genetic diversity that cannot be explained by drainage differences is still large. This does not mean that the drainage differences do not contribute to geographical isolation. In fact, an extent of geographic cohesion of haplotypes are found in Figure 5. The expression in L434 (not only by...) seems fine. 

 OUR REPLY—We modified this section following your comment. 

(17) L429–: Is there no possibility that the distribution of haplotypes was partly caused (disturbed) by artificial transportation? 

 OUR REPLY—We did not consider the possibility for artificial introduced scenario in A. normalis. According to our observation in the field, this fish prefers habitats in small brooks in mountain area, instead of large, smooth waters, which keeps them far away from fish farms in low land accompanied with aquaculture species (to be caught and released accidently). And they are small in size, poor in color, and not a rare species in Hainan Island considered as a non-concerned species. There seems no reason to introduce them by local people. 

(18) L437–: Figure 5 (especially 5b) should be placed in Results because of easy recognition of the distribution of haplotypes (clades).

OUR REPLY—We moved Figure 5 to L372 (and re-name it to Fig. 3 in the revised manuscript) following your suggestion. 

(19) L510, “except for the west central mountains southwestern mountains, and lowlands of Hainan Island.”: From where does this inference come?

OUR REPLY—In the first beginning we supposed the Clade I might be the ancestor of other clades; the distribution area of Clade I is supported to be same as that of the common ancestor of A. normalis. It is over explained and we removed this sentence.

(20) L513, “the river systems within Hainan Island did not create efficient isolation barriers similar to the strait”: This should be explained only by the short period after the last isolation. The drainage and strait undoubtedly work as barriers for strictly freshwater fish. 

 OUR REPLY—In general the drainage works as barriers for strictly freshwater fish; however, river capture and flood events can provide opportunities for freshwater fish to move across such barriers. Such events occurred in Hainan frequently. We met a serious flood in in Hainan 2011 caused heavy rainfalls. The water level raised to 4 m higher in a small mountainous brook we visited. Over one million people suffered by the flood in lowland towns then. We witnessed the power of flood and saw how easy it broke the barriers in Hainan. 

We modified the description in L528-531. 

(21) L516–, “this research demonstrates that vicariant events have not yet generated interregional population differentiation in P. normalis.”: Same as above.

OUR REPLY—We had modified the description to make it more reasonable. 

(22) L27, “based on molecular data and modifications based on climate patterns”: Difficult to understand here in Abstract.

 OUR REPLY—We had modified the description to make it more reasonable.

(23) L32, “(W).The”, and other places (L276...): Insert a space between two sentences.

OUR REPLY—We modified it following your suggestion.

(24) L36, “main land” 

OUR REPLY—We corrected it as “mainland”.

(25) L40, “It is”: What is “It”?

OUR REPLY—It referred to “Biogeographic studies”; we modified these words as “Biogeography”.

(26) L43, “how factors such as population structure ...”: Is the “population structure” a factor? I do not think so

OUR REPLY—.We had modified the description to make it reasonable. 

(27) L125, “were”  “are”

(28) L173, “20,000”  “15,000”?

(29) L212, legend of Fig. 3, “HKY+I+C”  “HKY+I+G”?

(30) L219, “cytb”: Italicized.

OUR REPLY—We modified them following your suggestions.

(31) L267, “the Lou River”: Where is it?

OUR REPLY—It is Site A1. We add “(Site A1)” in the sentence to make it clear more. 

(32) L290–: A too long paragraph. Change the paragraph at least at L302 (For...).

OUR REPLY—We modified it following your suggestion.

(33) L295, “Among”  “Within”? Also, the correspondence between clades and average haplotype differences values should be clarified.

OUR REPLY—We modified them following your suggestion. In table 2 we provide a more clear description in the table legend. 

(34) L299, “among sampling locality populations”: What is this?

 OUR REPLY—It should be “of sampling locality populations”. We had corrected it in the revised manuscript.

(35) L320, “the last model”: What is this?

OUR REPLY—It is the last hypothesis described in the section: “eight biogeographical areas based on climate”

(36) L343–, “and other two species of Aphyocypris. Like A. normalis and A. moltrechti, the two Aphyocypris species”: Show the species names of the two species.

OUR REPLY—We modified it following your suggestion.

(37) L352 and other places: TCS is the name of a computer software, not a name of method or algorithm.

OUR REPLY—We replace the term “TCS” by “haplotype network”.

(38) L381, “MCVRA”: What is this?

OUR REPLY—It was misspelling of “MRCA”- the most recent common ancestor.

(39) L385, “in the middle to late Pleistocene”: The range (at least of means) does not seem to include the late Pleistocene.

OUR REPLY—We simplified is as “in the Pleistocene”.

(40) L419, “is”  “were”

(41) L462, “Subclade...”: Change the paragraph.

OUR REPLY—We modified them following your suggestions.

(42) L487, “no obvious differentiation from Hainan Island populations.” 

OUR REPLY—We modified it to “... from parts of Hainan Island populations”?

(43) L498, “These geographic events”: Not very precise. Sea level rising in the Holocene did not provide the opportunity.

 OUR REPLY—We add “, and being isolated” to figure out the effects of sea level rising event. 

(44) L504, L506, “Subclade Ia separated from other Clade I populations”, “Subclade Ic separated from remaining Clade I populations”: A clade cannot separate from a population.

OUR REPLY—We modified the sentences following your suggestion.

(45) L511, “Clades”  “The major clades”?

(46) Fig. 1: In the panel a, “b” and “c” in white are less visible.

(47) Fig. 5: Indicate names of major river systems.

(48) Table 2: “evolutionary divergence”: Should show which index was used. 

OUR REPLY—We modified them following your suggestions.

---

## [Decision Letter · Decision Letter 1]

7 Apr 2021

PONE-D-20-37420R1

Phylogeography of Aphyocypris normalis Nichols and Pope, 1927 at Hainan Island and adjacent areas based on mitochondrial DNA data

PLOS ONE

Dear Dr. Jang-Liaw,

Thank you for submitting your manuscript to PLOS ONE. After careful consideration, we feel that it has merit but does not fully meet PLOS ONE’s publication criteria as it currently stands. Therefore, we invite you to submit a revised version of the manuscript that addresses the points raised during the review process.

We look forward to receiving your revised manuscript.

Kind regards,

Tzen-Yuh Chiang

Academic Editor

PLOS ONE

Reviewers' comments:

Reviewer's Responses to Questions

**Comments to the Author**

1. If the authors have adequately addressed your comments raised in a previous round of review and you feel that this manuscript is now acceptable for publication, you may indicate that here to bypass the “Comments to the Author” section, enter your conflict of interest statement in the “Confidential to Editor” section, and submit your "Accept" recommendation.

Reviewer #1: (No Response)

2. Is the manuscript technically sound, and do the data support the conclusions?

Reviewer #1: Partly

3. Has the statistical analysis been performed appropriately and rigorously? 

Reviewer #1: Yes

4. Have the authors made all data underlying the findings in their manuscript fully available?

Reviewer #1: Yes

5. Is the manuscript presented in an intelligible fashion and written in standard English?

Reviewer #1: No

6. Review Comments to the Author

Reviewer #1: This manuscript by IS Chen and NH Jang-Liaw is the revised version of PONE-D-20-37420, which reports the mtDNA phylogeography of a cyprinid fish, Aphyocypris normalis, commonly distributed in the southern part of China including Hainan Island, based on an extensive geographical sampling. Responding to several critical opinions on the original version, the manuscript has been largely revised and at least partly improved.

However, unfortunately I should again point out several flaws in the present form of the manuscript, which should be further improved. Also, the manuscript is still immature in terms of structure, explanation, and sentences. I would be happy if my comments will help the authors improve their manuscript for its early publication.

Major points:

(1) Divergence time estimation:

L245–: Because the divergence time estimation is directly affected by the evolutionary rate, the validity of 1.0%/Myr should be explained, or some excuse is required for the resultant estimates. Also, in the Discussion part (L517–), more careful discussion is required for the divergence time.

Furthermore, I’m not sure whether the authors notice it or not, Huang et al. (2019) present about twice values for the corresponding estimates (e.g., tMRCA of A. normalis = 2.33 mya, 1.92–2.74). Huang et al. (2019) mentioned as “A molecular clock was calibrated using a divergence rate of 2% per million years (Yang et al. 2016)” and the basis of Yang et al. (2016) is “A molecular clock was calibrated using the divergence rate of 2% per million years, as previously postulated by Brown et al. [44] and Bermingham and Martin [45]”, i.e., very famous, classic papers’ values. Namely, Huang et al. (2019) should use the evolutionary rate of 1% (not 2% of the “pairwise” value, which should be half for one lineage), as same in this study. The discrepancy (twice values of the estimates) is quite strange. I simply guess that the Huang et al. (2019) misused the pairwise divergence value (2%/Myr) as the evolutionary rate directly, if the present study is correct. Similar mistakes can be found in not few papers. This can be mentioned to emphasize the value of the present revised analysis.

(2) L344–, “suggesting that the differences in climates among biogeographic regions constitute effective isolation mechanisms to gene flow”:

This “suggestion” is not satisfactory. Correlation does not necessarily mean causality. Also, there are many differences between the boundaries in Fig. 3a and 3b.

Also in the Discussion part (L457–), I cannot understand the degree of congruence between climate division by He and Zhang [45] and phylogeographic division by this study, although I agree that geographic and climatic settings (topographical features, frequency of floods, etc.) have affected the population structure by disturbing the separation of basins that is an important basis of isolation. I believe that the latter claim has general value and should be emphasized in this paper. However, the evidence for the claim in this manuscript is still not sufficiently convincing.

(3) “isolation by distance” is not well evidenced.

The authors contrast an isolation by distance scenario with a vicariance scenario (L501, L520–523, L531–). But they do not provide the evidence distinguishing these scenarios. Peripheral populations can be isolated also by “small” vicariance factors.

(4) The structure of the “Discussion” part is confusing.

Obviously, the paragraphs from L508–L536 should be included the preceding section (“Genetic diversity and structuring in Aphyocypris normalis”), since the paragraph is the discussion deeply relating to those just above. Only the paragraph from L537 may be separated (but see the below comment).

(5) Discussion of A. arcus (L537–546) should be shortened or removed. In the latter case, the results should be also removed from the main text and figures.

The phylogenetic relationships of Aphyocypris spp. have already shown in Huang et al. (2019) with the same type (mtDNA, but longer) of data. The only new knowledge is that the authors could not collect the A. arcus in Hainan Island. Nobody can conclude that A. arcus is not distributed in Hainan from this result of the sampling. A possible way to do it is to reexamine the specimens used in Du et al. [76] with comparative specimens and to show their morphological key is not effective to identify different species. In this paper, at least the expression of the suggestion should be weakened more.

(6) Discussion of A. moltrechti and A. amnis (L548–580) should be much shortened or removed, because no new information is given in this study for this matter.

I agree that any taxonomic problems should be resolved as soon as possible. However, the present data do not provide any new information to resolve this problem. This part is just the authors’ claim.

That is, (1) the low genetic diversity in the Zhoushui R. population and its small divergence from other populations have already been reported in Chiang et al. [77] using mtDNA and microsatellite data; (2) Liao et al. [62] provide the morphological diagnostic characters for A. amnis, which seem not to be distinct and hence controversial; (3) generally, taxonomic ranking (intraspecific variation, ESU, subspecies, or species) of allopatric populations with small morphological/genetic differences could be often subjective; and (4) there are biological species with no genetic differences from relatives in neutral markers. A possible way to conclude this problem is to publish a nomenclatural article to synonymize the A. amnis with integrating morphological and genetic data.

However, some taxonomists may consider that if a population can be distinguishable from others in any characteristics or combination of characteristics, it can be treated as a “species” and the species rank is more beneficial for conservation action than just a local population.

I do not dare to discuss this matter with the authors, but I only like to say that the part should be shortened or removed because no new information is given in this study and the discussion is not satisfactory to lead to resolution.

Medium points:

(7) L36, “The relationships of A. normalis between Hainan and mainland China populations are also discussed in this report.”

Again, I should say that the readers of Abstract want to know what is discussed.

(8) L161–: The section of “Data analysis” is better to be divided. The next section “Divergence times among major clades” is also “data analysis”; so they are unbalanced. The section of “Data analysis” includes two topics; i.e., “Phylogenetic analysis” (including two datasets) and “Population structure” (AMOVA and SAMOVA).

Also in the “Results” part, the results of AMOVA/SAMOVA (L321–377) should not put in “Phylogenetic analysis” section because they are not phylogenetic analysis. A separate section should be established.

(9) L185–: In the explanation of AMOVA/SAMOVA, the authors should first explain all the models include two hierarchies of groups (i.e., groups of population samples and population samples). Some (schematic or realistic) figure for each model (even as supplementary) will be effective. Consider presenting it. It is not easy to understand the models only by text.

(10) around L246: The tree prior should be explained here. According to Table 5, the constant population model seems to be used. I strongly recommend that the analysis with other models, especially the Birth–Death model (see Ritchie et al. 2017: Syst Biol, DOI:10.1093/sysbio/syw095), should be also added and mentioned for showing the robustness of the results. This is because coalescent processes for the geographical samples of animals with low dispersal ability (like freshwater fish) should differ from those in a constant population.

(11) L247, “for each partition”: There is no explanation on the partition of sequence data. Should be explained.

(12) L295, “NJ and ML trees (S1 Figure)”: The information of the branch length of the ML tree should be kept and shown. Also, bootstrap values should be shown also in S1 Figure.

(13) L321–L345: Why does not the authors refer to the number of hypotheses here? What are the “two hypotheses” (L336) and “the last hypothesis” (L339)? The explanation here is difficult to understand. L321–323: This should not be a main topic (i.e., relation to hypotheses is unclear). Gene flow (L323) between which? It will much better to start to explain important issues.

(14) L359–: The result of SAMOVA should be shown as a figure (even if in supplementary).

(15) L396–: The term “Clade” is not suitable if it is defined in the haplotype network, because some are not monophyletic. The “Clade” should be those defined in Figure 2, shouldn’t it? Explain it first. Relating to this, not only Clade II but also Clade I is NOT monophyletic (L404).

(16) L483, “Subclade Ia is currently limited mainly to the west-central mountain area and shows clear differentiation between nearby populations in ranges NW and SW. “

I cannot understand this part. The Subclade Ia is distributed in NW. Probably it’s Subclade Ib in C, according to Fig. 3b, isn’t it?

(17) L496–, L519–: No figure is provided for haplotype (clade) distribution in the continental area (only explained in text). I wonder if such a figure also should be put in Fig. 3(c?).

Minor/Trivial:

(18) General: A blank line should be inserted between paragraphs and between figure legend/table footnote and text, if no indent is used. It is difficult for reviewers to identify the paragraphs and to distinguish the text from explanation of figures/tables in this manuscript.

(19) General: The name of a river is usually accompanied by “the” (definite article). There are inconsistencies in many places.

(20) L91, “with much lower sampling density”: Compared what? That does not yet appear.  e.g., “with too low sampling density to clarify....”

(21) L96: “within”  “with”?

(22) L98: “a freshwater fish”  “this freshwater fish”, “this species”.... Should be restricted.

(23) L102: “proposed”  “propose”

(24) L106: “were”  “are”

(25) L109: “Sample used”  “Samples used”

(26) L175, “15,000,000 trees”: This should be “15,000 trees”. (i.e., 2,000,000/100*0.75=15,000)

(27) L177–, “The genetic distances ... were calculated using MEGA5 with the Kimura 2-parameter model (K2P)”: In the response letter, the authors explaine the reason why the best model was not used as it (HKY+I) cannot be available in MEGA. There should not be large differences between the K2P and HKY because they are similar models (ti, tv, 2-parameter model). However, I wonder why the authors do not use other software that can use HKY+I to calculate the distances (e.g., PAUP*).

(28) L181, “which was carried out using ANeCA v1.2”: Should be deleted. TCS, which was actually used in this study, is an independent program. If ANeCA can automatically nest the network, the explanation should be changed. However, the result of the nesting is only partly shown in the Results (Fig. 5). Readers cannot see full results of the nested clades.

(29) L185, “The relative contribution...”: Change the paragraph. AMOVA/SAMOVA are quite different approaches from tree analyses.

(30) L193, “as (hypothesis 3) and (hypothesis 4)”: Should remove parentheses.

(31) L205: “suggests”  “suggest”

(32) L206: “the hypothesis”  “the relevant <or corresponding=""> hypothesis”

(33) L206, “SAMOVA 2.0 [48] to confirm that the grouped populations are geographically homogeneous and maximally differentiated from each other or not,...”: This explanation should be insufficient (inadequate) and reconsidered.

(34) L211, “We set the population of a single river system as a grouping unit (17 counts),”: What is the “population”? “population samples”?

(35) L213: “exam”  “examine”?

(36) L214–, “However,... pattern).”: I think this sentence should be removed. Why “However”?

(37) L216–222: Redundant. This part can be shortened.

(38) L228: “are”  “is”

(39) L269–, “Estimates of genetic variability differed accordingly.”: Not enough words, and not necessary. Should be removed

(40) L270–274, “Within-population ....(Table 1).”: Put this general description above the specific description ( L264).

(41) L286: The section title “Phylogenetic analysis” seems not suitable in the Results. For example, “Phylogenetic relationships” seems better.

(42) L290: “including”  “including those in”

(43) L318, “Distances”: Which distances? Should be clearly shown.

(44) L383–: “Like A. normalis and A. moltrechti, the two Aphyocypris species also formed...”  “Aphyocypris kikuchii and A chinensis also formed...”

(45) L392, “Haplotype Network and variations among clades”: “Network”  “network”. “variations among clades”  What variation?

(46) L394, “within haplotypes”: Impossible.

(47) L396, “MP tree”: No such analysis in this paper.

(48) L399, “a fourth”: “one of four sampling locations”? But there are eight localities on the mainland. Ununderstandable.

(49) L400, “However”: Ununderstandable.

(50) L402: “, and included”  “, which included” (h1 cannot include five haplotypes.)

(51) L405, L406, “the two haplotypes”, “these two haplotypes”: Clearly show them.

(52) L414, “Numbers inside circles are haplotypes.”: Should not be shown in bold.

(53) L426: “Pleistocene”  “the Pleistocene”

(54) L428: “tMRCA” and “tmarca” should be consistent.

(55) L439, “These two groups”: There are three groups (clades). What are the groups here?

(56) L455: “resulting”  “resulted”?

(57) L476, “highly supported phylogenetic subclade”: Show the subclade in parentheses.

(58) L484: “Upstream Changhua River (range SW) populations show...”  “Sharing Clade III haplotypes, the upstream Changhua River (range SW) populations show...”. (to add an explanation of the “close relationship”)

--- end of comments</or>

7. PLOS authors have the option to publish the peer review history of their article (what does this mean?). If published, this will include your full peer review and any attached files.

Reviewer #1: No

---

## [Author Response · Author response to Decision Letter 1]

24 Jun 2021

We appreciate the reviewer’s comments and suggestions. The manuscript has been revised accordingly. The following are our answers to the comments: 

Major points:

(1) Divergence time estimation:

L245–: Because the divergence time estimation is directly affected by the evolutionary rate, the validity of 1.0%/Myr should be explained, or some excuse is required for the resultant estimates. Also, in the Discussion part (L517–), more careful discussion is required for the divergence time.

Furthermore, I’m not sure whether the authors notice it or not, Huang et al. (2019) present about twice values for the corresponding estimates (e.g., tMRCA of A. normalis = 2.33 mya, 1.92–2.74). Huang et al. (2019) mentioned as “A molecular clock was calibrated using a divergence rate of 2% per million years (Yang et al. 2016)” and the basis of Yang et al. (2016) is “A molecular clock was calibrated using the divergence rate of 2% per million years, as previously postulated by Brown et al. [44] and Bermingham and Martin [45]”, i.e., very famous, classic papers’ values. Namely, Huang et al. (2019) should use the evolutionary rate of 1% (not 2% of the “pairwise” value, which should be half for one lineage), as same in this study. The discrepancy (twice values of the estimates) is quite strange. I simply guess that the Huang et al. (2019) misused the pairwise divergence value (2%/Myr) as the evolutionary rate directly, if the present study is correct. Similar mistakes can be found in not few papers. This can be mentioned to emphasize the value of the present revised analysis.

OUR REPLY—We had found that the difference of tMRC data between Huang et al.s’ result and ours. However, it was difficult to compare all the divergence time estimations in detail between these two works for they release very few information on it (only short description on divergence events order in the paragraph before “Discussion”). Yes, it seems a misusing as you mentioned. Your comment is valuable and we have added it in our discussion (L489-496). It is possible that the misusing is common in the results of other works like Glyptothorax (Chen et al. 2007), however, they did not provide clear, detailed information on their methods and results in the published papers as well. We cannot make comments on other papers on this issue.

(2) L344–, “suggesting that the differences in climates among biogeographic regions constitute effective isolation mechanisms to gene flow”:

This “suggestion” is not satisfactory. Correlation does not necessarily mean causality. Also, there are many differences between the boundaries in Fig. 3a and 3b.

Also in the Discussion part (L457–), I cannot understand the degree of congruence between climate division by He and Zhang [45] and phylogeographic division by this study, although I agree that geographic and climatic settings (topographical features, frequency of floods, etc.) have affected the population structure by disturbing the separation of basins that is an important basis of isolation. I believe that the latter claim has general value and should be emphasized in this paper. However, the evidence for the claim in this manuscript is still not sufficiently convincing.

OUR REPLY—In general, we will consider the vicariance of different rivers will be the main isolation mechanism for freshwater fauna. It does not work in Hainan; at least in the phylogeographic results of A. normalis in this study. We tried to find a possible hypothesis and finally found He and Zhang’ work (1985). We set and modified their climate division to identify our phylogeographic division hypothesis and test it by ANOVA. We compared the “Drainage by river systems” hypothesis (Scenario 3) and climate-modified biogeographic regions hypothesis (Scenario 4) in Hainan and found latter is a more reasonable scenario based on the AMOVA test (Table 3). 

(3) “isolation by distance” is not well evidenced.

The authors contrast an isolation by distance scenario with a vicariance scenario (L501, L520–523, L531–). But they do not provide the evidence distinguishing these scenarios. Peripheral populations can be isolated also by “small” vicariance factors.

OUR REPLY—We identified the “by distance” and “by vicariance” of scenario following the possible connection between rivers (such as belong to large Pearl River System will be considered as “by distance”). As you mentioned, it is well accepted that peripheral populations can be isolated also by “small” vicariance factors. However, in a land/island suffered by serious flood events frequently, like Hainan Island or Pearl River system in southern China, the vicariance factors occurred by close-by rivers should be considered as a weak isolation mechanism in the long differentiation history of freshwater fishes there. 

(4) The structure of the “Discussion” part is confusing.

Obviously, the paragraphs from L508–L536 should be included the preceding section (“Genetic diversity and structuring in Aphyocypris normalis”), since the paragraph is the discussion deeply relating to those just above. Only the paragraph from L537 may be separated (but see the below comment).

OUR REPLY—We had modified the title of this section to response your comment. Others please see our response to comment 5 below.

(5) Discussion of A. arcus (L537–546) should be shortened or removed. In the latter case, the results should be also removed from the main text and figures.

The phylogenetic relationships of Aphyocypris spp. have already shown in Huang et al. (2019) with the same type (mtDNA, but longer) of data. The only new knowledge is that the authors could not collect the A. arcus in Hainan Island. Nobody can conclude that A. arcus is not distributed in Hainan from this result of the sampling. A possible way to do it is to reexamine the specimens used in Du et al. [76] with comparative specimens and to show their morphological key is not effective to identify different species. In this paper, at least the expression of the suggestion should be weakened more.

OUR REPLY—Huang et al.s’ work (2019) did not touch the issue of population of A. arcus on Hainan Island. We (both of the authors and their team members) had excused more than 6 freshwater fish specimen collection trips in Hainan and over dozen collecting trips in southern mainland China. All the specimens used in this study were collected by our team members in the habitats, not purchasing from local markets (this is the reason why our sample size was small in some localities). For the corresponding author, he had a long trip in KuangXi Province for collecting freshwater fish specimen in 2002 with Pisces Division of Chinese Academy of Sciences, Beijing. He met several populations of A. arcus in Red River system in KuangXi and kept several specimens of A. arcus in NMNS, Taichung, Taiwan to public for later comparative studies like this work. We do not dare to say we are good enough to be experts on ichthyological fauna of Southern China, but we are sure that our collection efforts in Hainan were comprehensive enough to cover at least one possible A. arcus population. We found no any A. arcus in our specimen after the DNA check on all Aphyocypris species collected from Hainan. There will be arguments always that nobody can claimed that they had visit ALL possible A. arcus habitats in Hainan and have a conclusion on the absent of the species in the island based on their specimen collection results. However, we just provided our observation and comments on A. arcus in Hainan based on our data and experience. This will be a valid contribution on freshwater fish studies in this area we believe. 

We have weakened our comment by adding a sentence in the end of the section following your suggestion.

(6) Discussion of A. moltrechti and A. amnis (L548–580) should be much shortened or removed, because no new information is given in this study for this matter.

I agree that any taxonomic problems should be resolved as soon as possible. However, the present data do not provide any new information to resolve this problem. This part is just the authors’ claim.

That is, (1) the low genetic diversity in the Zhoushui R. population and its small divergence from other populations have already been reported in Chiang et al. [77] using mtDNA and microsatellite data; (2) Liao et al. [62] provide the morphological diagnostic characters for A. amnis, which seem not to be distinct and hence controversial; (3) generally, taxonomic ranking (intraspecific variation, ESU, subspecies, or species) of allopatric populations with small morphological/genetic differences could be often subjective; and (4) there are biological species with no genetic differences from relatives in neutral markers. A possible way to conclude this problem is to publish a nomenclatural article to synonymize the A. amnis with integrating morphological and genetic data.

However, some taxonomists may consider that if a population can be distinguishable from others in any characteristics or combination of characteristics, it can be treated as a “species” and the species rank is more beneficial for conservation action than just a local population.

I do not dare to discuss this matter with the authors, but I only like to say that the part should be shortened or removed because no new information is given in this study and the discussion is not satisfactory to lead to resolution.

OUR REPLY—We agree with your totally. All the points you mentioned had been released at the same year that the new species A. amnis was reported. However, Liao et al. (2011) did not face up to the released information, and consciously ignored them. The identification key provided by Liao et al. (2011) could not effectively distinguish the new species from A. moltrechti, which caused difficulties for the follow-up fish researchers in identification. We agree that the promotion of a treated population into the species rank is more beneficial for conservation action than just a local population, but it should be based on a taxonomic truth, using the correct and nature taxa, then carry out its conservation strategy. From a point of view on DNA, the A. amnis/A. moltrechti populations from three rivers do show genetic differentiation between them (although very tiny comparing to other cyprinids like A. normalis). However, the need to promoted the taxonomic rank for the population in the Zhoushui R. to protected it should be re-considered carefully. The differentiation of freshwater fish in different water systems is very common, and it is impossible for us to divide all the populations that have mutations and rising their conservation level, especially those with such a small degree of variation. If taxonomies are not established based on the “true” speciation view, the precious funding, resources for conservation will be diluted and waste, and cannot be used on the species/populations that need to be protected emergency. 

It is the responsibility of local ichthyologists to establish/correct ichthyological taxonomy in their land. We feel embarrassed that we have not been able to make a complete statement on this issue in a single paper. Herein we integrate some previously known but ignored information with few new data we provide to emphasis to this view and put it forward again. We do not expect to have a complete statement to reply to this issue, but we hope to remind Taiwanese ichthyologists that this issue is controversial, and we hope that there will be more complete reports on this issue in the future.

Medium points:

(7) L36, “The relationships of A. normalis between Hainan and mainland China populations are also discussed in this report.”

Again, I should say that the readers of Abstract want to know what is discussed.

OUR REPLY—We have modified the abstract following your comment.

(8) L161–: The section of “Data analysis” is better to be divided. The next section “Divergence times among major clades” is also “data analysis”; so they are unbalanced. The section of “Data analysis” includes two topics; i.e., “Phylogenetic analysis” (including two datasets) and “Population structure” (AMOVA and SAMOVA).

Also in the “Results” part, the results of AMOVA/SAMOVA (L321–377) should not put in “Phylogenetic analysis” section because they are not phylogenetic analysis. A separate section should be established.

OUR REPLY—We have modified this manuscript following your comments.

(9) L185–: In the explanation of AMOVA/SAMOVA, the authors should first explain all the models include two hierarchies of groups (i.e., groups of population samples and population samples). Some (schematic or realistic) figure for each model (even as supplementary) will be effective. Consider presenting it. It is not easy to understand the models only by text.

OUR REPLY—It is the major contribution of this work to describe the isolation scenarios revealed by the AMOVA/SAMOVA analyses. We have modified the manuscript to clarify the scenario reference and showed the result of SAMOVA with a new figure S2 following your comments (13) and (14) respectively. Your excellent comments help us to improve this manuscript a lot.

(10) around L246: The tree prior should be explained here. According to Table 5, the constant population model seems to be used. I strongly recommend that the analysis with other models, especially the Birth–Death model (see Ritchie et al. 2017: Syst Biol, DOI:10.1093/sysbio/syw095), should be also added and mentioned for showing the robustness of the results. This is because coalescent processes for the geographical samples of animals with low dispersal ability (like freshwater fish) should differ from those in a constant population.

OUR REPLY—In the first beginning of this study, we found the isolation mechanisms usual to freshwater animals (isolated by strait, separated between rivers, etc.) were not suitable in this study for we detected the closer relationships among populations between mainland China and Hainan Island, as well as between river systems. We decided to use a simple, neutral model to detect the MRCA in A. normalis in this study and finally we choose the constant population model. We have modified this section under your suggestion.

(11) L247, “for each partition”: There is no explanation on the partition of sequence data. Should be explained.

OUR REPLY—The “partition” is the DNA sequence data. We have modified this sentence to clarify it. 

(12) L295, “NJ and ML trees (S1 Figure)”: The information of the branch length of the ML tree should be kept and shown. Also, bootstrap values should be shown also in S1 Figure.

OUR REPLY—We have modified the S1 Figure following your suggestion.

(13) L321–L345: Why does not the authors refer to the number of hypotheses here? What are the “two hypotheses” (L336) and “the last hypothesis” (L339)? The explanation here is difficult to understand. L321–323: This should not be a main topic (i.e., relation to hypotheses is unclear). Gene flow (L323) between which? It will much better to start to explain important issues.

OUR REPLY—We have modified the Table 3 to refer the “Hypothesis scenarios” in this study to response your comment.

(14) L359–: The result of SAMOVA should be shown as a figure (even if in supplementary).

OUR REPLY—We have made a figure (Figure S2) to response your comment.

(15) L396–: The term “Clade” is not suitable if it is defined in the haplotype network, because some are not monophyletic. The “Clade” should be those defined in Figure 2, shouldn’t it? Explain it first. Relating to this, not only Clade II but also Clade I is NOT monophyletic (L404).

OUR REPLY—We have modified the Network descriptions following your comment.

(16) L483, “Subclade Ia is currently limited mainly to the west-central mountain area and shows clear differentiation between nearby populations in ranges NW and SW. “

I cannot understand this part. The Subclade Ia is distributed in NW. Probably it’s Subclade Ib in C, according to Fig. 3b, isn’t it?

OUR REPLY—Yes it should be Subclade Ib in C. We have corrected the mistake.

(17) L496–, L519–: No figure is provided for haplotype (clade) distribution in the continental area (only explained in text). I wonder if such a figure also should be put in Fig. 3(c?).

OUR REPLY—We have made a figure (Figure 3c) to response your comment.

Minor/Trivial:

(18) General: A blank line should be inserted between paragraphs and between figure legend/table footnote and text, if no indent is used. It is difficult for reviewers to identify the paragraphs and to distinguish the text from explanation of figures/tables in this manuscript.

OUR REPLY—We have marked all legends of tables and figures in blue color to response your comment.

(19) General: The name of a river is usually accompanied by “the” (definite article). There are inconsistencies in many places.

OUR REPLY—We have modified the manuscript following your comment.

(20) L91, “with much lower sampling density”: Compared what? That does not yet appear.  e.g., “with too low sampling density to clarify....”

OUR REPLY—We have modified the manuscript following your comment.

(21) L96: “within”  “with”?

OUR REPLY—We have corrected it following your comment.

(22) L98: “a freshwater fish”  “this freshwater fish”, “this species”.... Should be restricted.

OUR REPLY—We have modified the manuscript following your comment.

(23) L102: “proposed”  “propose”

OUR REPLY—We have corrected it following your comment.

(24) L106: “were”  “are”

OUR REPLY—We have corrected it following your comment.

(25) L109: “Sample used”  “Samples used”

OUR REPLY—We have corrected it following your comment.

(26) L175, “15,000,000 trees”: This should be “15,000 trees”. (i.e., 2,000,000/100*0.75=15,000)

OUR REPLY—We have corrected it following your comment.

(27) L177–, “The genetic distances ... were calculated using MEGA5 with the Kimura 2-parameter model (K2P)”: In the response letter, the authors explaine the reason why the best model was not used as it (HKY+I) cannot be available in MEGA. There should not be large differences between the K2P and HKY because they are similar models (ti, tv, 2-parameter model). However, I wonder why the authors do not use other software that can use HKY+I to calculate the distances (e.g., PAUP*).

OUR REPLY—It is a pity that PAUP had no any update since 2000 though it was very popular for many phylogenetic researchers. More and more studies turn to use other programs such as MrBayes and MEGA. In my opinion, MEGA is a “growing” software and the last update (MEGA X) was released in 2018. It is popular to researchers currently for several basic sequence statistics available in MEGA, and the most of all, easy to use with friendly operational Interface. I (the corresponding author) decided to use MEGA in this work and said goodbye to PAUP after this work.

(28) L181, “which was carried out using ANeCA v1.2”: Should be deleted. TCS, which was actually used in this study, is an independent program. If ANeCA can automatically nest the network, the explanation should be changed. However, the result of the nesting is only partly shown in the Results (Fig. 5). Readers cannot see full results of the nested clades.

OUR REPLY—We had removed these words following your suggestion.

(29) L185, “The relative contribution...”: Change the paragraph. AMOVA/SAMOVA are quite different approaches from tree analyses.

OUR REPLY—We have modified the manuscript following your comment.

(30) L193, “as (hypothesis 3) and (hypothesis 4)”: Should remove parentheses.

OUR REPLY—We have modified the manuscript following your comment.

(31) L205: “suggests”  “suggest”

OUR REPLY—We have corrected it following your comment.

(32) L206: “the hypothesis”  “the relevant hypothesis”

OUR REPLY—We have modified the manuscript following your comment.

(33) L206, “SAMOVA 2.0 [48] to confirm that the grouped populations are geographically homogeneous and maximally differentiated from each other or not,...”: This explanation should be insufficient (inadequate) and reconsidered.

OUR REPLY—We have modified this sentence following your suggestion.

(34) L211, “We set the population of a single river system as a grouping unit (17 counts),”: What is the “population”? “population samples”?

OUR REPLY—Here the word “population” is similar to “sampling localities”. We add the latter one to clarify what we want to say. 

(35) L213: “exam”  “examine”?

OUR REPLY—We have corrected it following your comment.

(36) L214–, “However,... pattern).”: I think this sentence should be removed. Why “However”?

OUR REPLY—We have removed this sentence following your suggestion.

(37) L216–222: Redundant. This part can be shortened.

OUR REPLY—We have modified this section following your comment.

(38) L228: “are”  “is”

OUR REPLY—We have corrected it following your comment.

(39) L269–, “Estimates of genetic variability differed accordingly.”: Not enough words, and not necessary. Should be removed

OUR REPLY—We have removed the incomplete sentence following your suggestion.

(40) L270–274, “Within-population ....(Table 1).”: Put this general description above the specific description ( L264).

OUR REPLY—We have re-arranged this section following your suggestion. 

(41) L286: The section title “Phylogenetic analysis” seems not suitable in the Results. For example, “Phylogenetic relationships” seems better.

OUR REPLY—We have modified the section title following your suggestion.

(42) L290: “including”  “including those in”

OUR REPLY—We have modified this sentence following your comment.

(43) L318, “Distances”: Which distances? Should be clearly shown.

OUR REPLY—We have modified this sentence following your comment.

(44) L383–: “Like A. normalis and A. moltrechti, the two Aphyocypris species also formed...”  “Aphyocypris kikuchii and A chinensis also formed...”

OUR REPLY—We have modified this sentence following your comment.

(45) L392, “Haplotype Network and variations among clades”: “Network”  “network”. “variations among clades”  What variation?

OUR REPLY—We have simplified the section title as “Haplotype network”.

(46) L394, “within haplotypes”: Impossible.

OUR REPLY—We have changed this word with “between”.

(47) L396, “MP tree”: No such analysis in this paper.

OUR REPLY—It was a misuse of “ML tree”. We have corrected it in the manuscript.

(48) L399, “a fourth”: “one of four sampling locations”? But there are eight localities on the mainland. Ununderstandable.

OUR REPLY—It meant that there are 4 locations which host the haplotype h4. Three of the locations are in Hainan (sites 4, 5, 23), and the last (fourth) one is in mainland (site A5). It shows that the distribution of h4 is over the strait. 

(49) L400, “However”: Ununderstandable.

OUR REPLY—We have replaced the “However” by “Furthermore” in the sentence.

(50) L402: “, and included”  “, which included” (h1 cannot include five haplotypes.)

OUR REPLY—We have modified this sentence following your comment.

(51) L405, L406, “the two haplotypes”, “these two haplotypes”: Clearly show them.

OUR REPLY—We have modified these sentences to response your suggestions.

(52) L414, “Numbers inside circles are haplotypes.”: Should not be shown in bold.

OUR REPLY—We have modified these words following your comment.

(53) L426: “Pleistocene”  “the Pleistocene”

OUR REPLY—We have corrected it following your comment.

(54) L428: “tMRCA” and “tmarca” should be consistent.

OUR REPLY—We have corrected them following your comment.

(55) L439, “These two groups”: There are three groups (clades). What are the groups here?

OUR REPLY—We have modified this section following your comment.

(56) L455: “resulting”  “resulted”?

OUR REPLY—We have corrected it following your comment.

(57) L476, “highly supported phylogenetic subclade”: Show the subclade in parentheses.

OUR REPLY—We have modified this section following your comment.

(58) L484: “Upstream Changhua River (range SW) populations show...”  “Sharing Clade III haplotypes, the upstream Changhua River (range SW) populations show...”. (to add an explanation of the “close relationship”)

OUR REPLY—We have modified this section to explain why they shared close relationship.

--- end of comments

---

## [Decision Letter · Decision Letter 2]

29 Jul 2021

PONE-D-20-37420R2

Phylogeography of Aphyocypris normalis Nichols and Pope, 1927 at Hainan Island and adjacent areas based on mitochondrial DNA data

PLOS ONE

Dear Dr. Jang-Liaw,

Thank you for submitting your manuscript to PLOS ONE. After careful consideration, we feel that it has merit but does not fully meet PLOS ONE’s publication criteria as it currently stands. Therefore, we invite you to submit a revised version of the manuscript that addresses the points raised during the review process.

We look forward to receiving your revised manuscript.

Kind regards,

Tzen-Yuh Chiang

Academic Editor

PLOS ONE

Reviewers' comments:

Reviewer's Responses to Questions

**Comments to the Author**

1. If the authors have adequately addressed your comments raised in a previous round of review and you feel that this manuscript is now acceptable for publication, you may indicate that here to bypass the “Comments to the Author” section, enter your conflict of interest statement in the “Confidential to Editor” section, and submit your "Accept" recommendation.

Reviewer #1: (No Response)

2. Is the manuscript technically sound, and do the data support the conclusions?

Reviewer #1: Partly

3. Has the statistical analysis been performed appropriately and rigorously? 

Reviewer #1: Yes

4. Have the authors made all data underlying the findings in their manuscript fully available?

Reviewer #1: Yes

5. Is the manuscript presented in an intelligible fashion and written in standard English?

Reviewer #1: No

6. Review Comments to the Author

Reviewer #1: This manuscript by IS Chen and NH Jang-Liaw is the second revised version of PONE-D-20-37420, which reports the mtDNA phylogeography of a cyprinid fish, Aphyocypris normalis, commonly distributed mainly in the southern part of China including Hainan Island, based on an extensive geographical sampling. Responding to several critical opinions on the original version, the manuscript has been revised and at least partly improved.

However, again, I should say that the manuscript is still immature in many points including problems of structure, descriptions, and a huge number of minor points.

I hope that the authors will carefully consider the below comments and revise the manuscript thoughtfully. Also, I hope that you will forgive me for adding many new comments that I failed to point out last time. Also, my comments at this time will not be complete ones. There are too many points... At the very least, please check the English before resubmitting.

Major points:

(1) Structure of the “Materials and methods” part

I strongly recommend that the authors reconsider the structure of “Materials and Methods”. Specifically:

(1-1) The paragraph of L229–242 is for the interspecific phylogenetic analysis, so it should be placed in the “Phylogenetic analysis” section. Move to L190.

(1-2) The “Divergence times among major clades” section (L244–260) should be placed before the “Population structure” section (L191–). This is because the divergence time estimation is the analysis next to “Phylogenetic analysis”.

(2) Structure of the “Results” part

I strongly recommend that the authors reconsider the structure of “Results”, too.

After the “Phylogenetic relationships” section (L256–), the order of contents will be better to be [1] “Phylogenetic relationships” (including that among the related species), [2] “Haplotype network”, [3] “Divergence times among major clades” (but also see below), and [4] “Population structure” (L338–404, AMOVA, SAMOVA section).

Specifically, L338–404 should be moved to the new (last) section of [4] “Population structure”, as in Methods, because this part is not about “Phylogenetic relationships”.

The results of phylogenetic relationships among the related species (L407–412) should be moved to L338. However, if I were an author, [3] Divergence time estimation would be placed before the part of phylogenetic relationships among the related species, without establishing an independent section. This is because of the tight connection between the phylogeny and divergence time.

(3) Structure of the Discussion part

The structure of the Discussion part should be also reconsidered. Specifically,

(3-1) The newly added part to examine Huang et al.’s [24] dating (L489–496) should be moved to the place after the discussion based on the divergence time estimated in the present analysis (L585?). The place in the present manuscript is quite inadequate.

(3-2) The section starting from L550 (Comments on distribution of...), at least its first half is one of the main topics of this study. I strongly suggest changing the heading, e.g., “Distribution processes of Aphyocypris normalis in Hainan and southern China”. And, if the authors like, add “, with a comment on Aphyocypris arcus in Hainan”. Again, the former is one of the main topics of this paper, not just a comment.

(4) L501–, and Results: It is not well explained how the final phylogeographic pattern was derived. It should be made clear how the climate zones were modified. Does it differ from the pattern solely based on the cytb data?

L502, “we have constructed...”: Where? Probably here the authors “propose” the modified divisions.

Medium points:

(5) L35–: The sentence from L35 (The divergence events of...) is better to be moved to L27, before the sentence “Additionally...” with small modifications as:

“The divergence events of these clades were estimated to have occurred between 1.05–0.16 Ma.”

This is because the explanation of the divergence times is not well linked with the explanation of the population divisions.

(6) L36–: The two sentences from L36 (Furthermore..., The strait between them...) should be removed, because these two sentences are a repetition of the above part and hence are redundant.

(7) L82–83: The distribution of Aphyocypris normalis should include Vietnam and some Indochina regions, according to the Results and Discussion parts.

(8) L115–: If the (most of) present samples were collected by the authors, they should be declared in this paragraph. At present, it reads as if it were simply a study using museum specimens without any fieldwork.

(9) L144: “which...” “Specimens that have been treated as Aphyocypris amnis.”

(10) L184, Issue of genetic distances and “PAUP*”:

I believe that the evolutionary distance is better to be calculated under the best evolutionary model that is shown in the same paper. However, I do not dare to claim this again because the authors have already denied it twice.

Now the only thing I like to say is that the author’s explanation does not make sense. The authors indeed used PAUP* in this paper. Also, PAUP* has been continually updated until now (see https://paup.phylosolutions.com/), although the authors write that the software has stopped updating in the reply letter.

Again, I do not dare to request the change because the models will not largely change the results (distances) and the usage of K2P is not incorrect (only it is not the best). However, only the thing I want to say is the authors’ explanation did not convince me.

(11) L247, L462, Issue of the tree prior:

In the previous comments, I recommend that the authors should try to use also the “Birth-Death” model as a tree prior (see Ritchie et al. 2017: Syst Biol, DOI:10.1093/sysbio/syw095) because the constant population size model is the coalescent model for a practically single population. The use of the latter simple model seems to be inconsistent with the population differentiation claimed in this paper. The authors explain something in the reply letter, which did not convince me. Because this is the authors’ paper, I do not dare to repeatedly comment on the same thing. However, I only want to say that the use of the constant population model alone may not be the best choice.

(12) L256, “The parameters of the substitution model GTR were estimated independently for each haplotype data.”: Although the authors note that they estimate the model parameters independently for each haplotype, I have completely no idea what they mean by that. Since this study uses only one gene (cytb), there must be only one set of parameters for GTR. Or, since it is a protein-coding region, it is possible to give three different models for each codon site. The authors should provide a clear explanation.

(13) L320–330, Figure S1: The branch length should be reflected in the ML tree. Why did the authors remove the length information only from the ML tree? Strange.

(14) L361–, “, suggesting... AMOVA results”: Should be removed because this is a discussion, not a result from the analysis. Should more carefully discuss this in the Discussion part.

(15) L391–396: This part is a discussion, which should be moved to the “Discussion” part. Also, readers are surprised when Indochina is suddenly mentioned. It should be properly explained in the “Introduction” and the “Materials and methods” parts.

(16) L561, “In summary”: It’s not “in summary”. Should start summarizing the results and depicting the history of this fish.  For example, “According to our divergence time estimation of the major clades of A. normalis, the order of phylogenetic isolation events began with....”

(17) L605–, “since only four haplotypes were identified from 22 sequences that were 1,091 bp long and only one haplotype was found in the Dajia and Wu Rivers’ population”:

The number of haplotypes neither supports nor denies a different species. Should be rewritten.

(18) L627–629, “We suggest that both....in the future.”: What’s happened? Is this “suggestion” for A. amnis issue? How related? Should be rewritten.

Minor/Trivial points:

(19) L38: “We suggest that...”  “The present results indicate that...”

(20) L44, “closely to....closely and”: Remove the latter “closely”.

(21) L73: “animal diversification”  “intraspecific diversification”. Should be more specific.

(22) L86: Remove “the major river systems of”, because the same phrase appears at the end of the sentence again and it is not necessarily needed in the first part.

(23) L88: “major river systems.”  “major river systems in these regions.”

(24) L93, “had”: Remove.

(25) L94: “same fish”  “the same fish”

(26) L94, “longer”: Remove. No comparison can be done, because the present data have not yet appeared.

(27) L96, “specific”: Remove? Obscure.

(28) L99, “Same as the study of Huang et al. [24],”: Remove. No need to refer to here. It gives an only negative effect.  “The objective of....”

(29) L100, “mitochondrial”: Remove. No need.

(30) L101, “more”: Remove.

(31) L102: “comprehensive sampling efforts across the species’ entire range”  “comprehensive geographic sampling”. Redundant and not very precise (how about Vietnam?)

(32) L102–104, “is the first...sampling localities,”: Remove. Repetitive and redundant.

(33) L105: “with mitochondrial cytochrome b gene”  “with the mitochondrial cytochrome b gene”

(34) L105, “combining the molecular data and the evolutionary time to estimate the evolution or dispersal...”: Should be rephrased, because the sentence is strange in some respects.  For example, “estimating evolutionary time to infer the distribution history of A. normalis with respect to....”

(35) L107: “We also propose...”  “On the basis of our results, we propose...”

(36) L110: “the nomenclature status”  “its taxonomic status” (Any nomenclatural statement should not be given in this paper.)

(37) L111: “(comments...) proposed”  “(comments...) provided”

(38) L167–168, “Preliminary phylogenetic... DnaSP v5 [33].”: Removed. No relation to the Results.

(39) L193: “were”  “was”, or (L192) “contribution”  “contributions”

(40) L251, “a fixed clock pairwise evolutionary rate”: Remove “clock” here.

(41) L264–, “A total...specimens were ... deposited in the GenBank database”: Inaccurate statement.

(42) L285–, “from Changhua River”: Add “the”. Check it throughout the text again.

(43) L290: “that stream. Copious haplotype”  “that stream; copious haplotype”. The second sentence is the specific description of the first sentence. Some connection (logical words or style) is necessary.

(44) L319, “sequence.”: Of what? The caption should be understandable on its own.

(45) L321, and Figure 2, “>90%”: As the authors may know that the Bayesian posterior probability is usually expressed as 0.00–1.00 as probability, not in %, although I do not dare to request to change it.

(46) L325–, “Abbreviations beside clades or sub-clades correspond to the sampling sites listed in Table 1.”: I cannot find such abbreviations.

(47) L338, “the two a priori phylogenetic hypotheses that we examined”: There are four hypotheses. Also, the part “, isolated by the strait (Scenario 1 in Table 3) and drainage by river system (Scenario 2 in Table 3),” seems not clear. Please rephrase this sentence.

(48) L339–363: Replace “scenario” by “hypothesis”, as is in “Methods”. Confused.

(49) L339–341: Two “in Table 3” should be removed and put “(Table 3)” at the end of the sentence (i.e., “gene flow (Table 3).”). Redundant.

(50) L341–343, “we found in both hypothesis... of the Hainan Island vs. mainland hypotheses”: Contradicted. The last part should be for only Hypothesis 1, not both.

(51) L343: Remove “see”.

(52) L348–, “Our second set of experiments...”: In fact, the part below describes scenarios (hypotheses) 2 and 3. What’s happened?

(53) L352: “are”  “were”. These are the results of a specific analysis. Also should correct the tense in L354 (are) and L356 (do).

(54) L354, “low”: Are the 0.32773 and 0.51140 really low? Strange.

(55) L358, “(Scenario 4 in Figure 3)”: In Figure 3, no word of “Scenario 4” appears. Strange citation.

(56) L360: “We found...”  “That is, we found...” This sentence is a specific explanation of the former sentence. Some connection is needed.

(57) L365– (Table 2): K2P values are shown as two orders of magnitude larger.

(58) L372–379: This part is a footnote of Table 3, isn’t it?

(59) L378, “ FST values correspond to among individuals within populations.”: This is quite wrong.

FST in AMOVA is the variation component assigned to populations (without group division). So, 1–FST is the value that corresponds to the variation component among individuals within populations, isn’t it?

(60) L381: “can be”  “could be” or simply “was”

(61) L382, “population Lou River”: Strange.  “the Lou River population”?

(62) L385: Cite “(Figure S2)” after “Hainan Island”

(63) L385: “the AMOVA”  “the results of the AMOVA”

(64) L398: “climate”  “Climate”

(65) L415: “its sister species”  “its closely related species”. Check the meaning of “sister species”.

(66) L416, “of the MrBayes program”: I do not think that the software name is needed to be described in the legend of the figure.

(67) L417: “Bayesian inferences”  This should be “Bayesian posterior probabilities”

(68) L422, “A haplotype network is helpful in understanding the structure of phylogenetic relationships between haplotypes of Aphyocypris normalis among populations”: Should be rephrased. Strange. What is “the structure of phylogenetic relationships”? Phylogenetic relationships of what? “Between haplotypes”, or “among populations”?

(69) L424, “two groups were erected”: What are these? Specify. First, these two groups should be defined (explained) based on the network.

(70) L425, “only the small group of haplotypes h7, h8, h10, h11 is agreed with the Bayesian haplotype tree and ML as the Clade III”: Should be rephrased. See the above comment.

(71) L428, “network. . ”: Delete the second colon.

(72) L437, “In the small group of network result”: Reader cannot understand what this is. Rephrase, e.g., as “In the smaller group of the whole network (Clade III), the variation among the haplotypes was low...”

(73) L440, “occurred”: Should be removed.

(74) L441: “haplotype”  “specimen”

(75) L444, “Parsimony haplotype network”: More precisely, a “Statistical parsimony haplotype network” is obtained by TCS. Not a “parsimony” network.

(76) L445: “haplotypes”  “haplotype codes”?

(77) L453–456: Reconsider the number of significant digits to be shown. Also, make the number of significant digits the same among places

(78) L452: “MRCA divergence times for the three major clades were estimated at...”  “MRCA divergence time for each of the three major clades was estimated at...” I think that the authors want to explain NOT “the time of MRCA of the three clades” but “that of each of the three clades”. Also, remove “, respectively” at the end of the sentence.

(79) L461, “significant”: Unnecessary?

(80) L462 (Table 5): What are the numbers in parentheses? No explanation.

(81) L467: “reveals”  “revealed”

(82) L475: “the strait”  “the Qiongzhou Strait”. This is the first place about this strait in the Discussion part.

(83) L477: “the major clade, Clade I, ”  “one of the major clades, Clade I, ” or simply “Clade I”

(84) L482: “according to the phylogenetic analyses”  “according to the present comprehensive phylogenetic analyses”. Why don’t you emphasize this here?

(85) L486: “were”  “was”

(86) L489: “the corresponding estimates” of WHAT???

(87) L492: “which previously postulated”  “which was previously postulated”

(88) L493, “Nevertheless”: Rephrase. This word does not make sense in this context.

(89) L493–495, “Nevertheless, Huang... value.”: This is a guess (inference), not a fact. Should be rephrased so that it can be understood to be a guess. Also, the sentence from L495 (The inconsistency...) should be placed before the previous one.

(90) L529: “valid”  “distinct” or other adequate words?

(91) L540: “with the Pearl River System”  “with that of the Pearl River system”

(92) L543, “(the major)”: Should be Removed. No information.

(93) L557, “77].With”: Insert a space.

(94) L564: “by vicariance and by distance” -> “by vicariance or by distance”?

(95) L565: “three subclades:”  “three subclades of Clade I, i.e., ”

(96) L562–570: “separating” or “separated”?

(97) L568–: “in which show no efficient geographic barrier between them”  “in which no efficient geographic barrier is shown between them” (no subject)

(98) L571: “that run northwestward.”  “that run northwestward in Hainan.”

(99) L574: “land bridge existed...”  “land bridge that existed...”

(100) L577: “vicariance”  “isolated”

(101) L579–, “In general, this research demonstrates that vicariant events have not yet generated interregional population differentiation in A. normalis.”: Strange. Should be rephrased, as, e.g., “This research demonstrates that apparent vicariance events do not necessarily explain interregional population differentiation in A. normalis.”

(102) L582, “original”: What?

(103) L585: The author and description year should be shown for A. arcus anywhere, if any taxonomic discussion is done.

(104) L592, “We suggest that A. arcus...”  “We suggest a high possibility that A. arcus...”

(105) L594–, “More comprehensive comparative studies of A. arcus on Hainan Island both on DNA and morphology are necessary in the future.”: Strange. In the sentence immediately above, the authors state the possible absence of this species in Hainan, so a comprehensive comparative study may not be possible.

(106) L605, “however”: Should be removed.

(107) L609: “0.0035”  “0.0035 on average”

(108) L608, “Aphyocypris moltrechti and A. amnis”  “Aphyocypris moltrechti (only the specimens from the Dajia R. and Wu R.) and A. amnis (the specimens from the Zhoushui R.)”. The authors do not support the validity of the latter species, don’t they? Also, in L610, “these two species” will be better to be “these two groups”.

(109) L615, “Their”: Whose? Describe.

(110) L618: “Zhoushui River population”  “the Zhoushui River population”

(111) L620: “population”  “populations”

(112) L620: “Compare”  “Comparing”/“Compared”

(113) L622: “remarkable”  “remarkably”

(114) L625: “the conclusion comes with...”  the conclusion that comes with...”

(115) L626: “mitochondrial data as the only marker for inferring population histories may...”  “only mitochondrial data may...”

--- end of comments

7. PLOS authors have the option to publish the peer review history of their article (what does this mean?). If published, this will include your full peer review and any attached files.

Reviewer #1: No

---

## [Author Response · Author response to Decision Letter 2]

12 Sep 2021

Again, we deeply appreciate the reviewer’s great comments and suggestions on this work. The manuscript has been major revised accordingly as possible we can, and it had been modified completely almost. We accept almost all suggestions the reviewers offered, and some we need more time to reconsider them. 

The following are our answers to the comments which we did not follow completely. Others were all accept by the authors. 

Major points:

OUR REPLY—We have modified the manuscript following all your major comments.

Medium points:

(10) L184, Issue of genetic distances and “PAUP*”:

I believe that the evolutionary distance is better to be calculated under the best evolutionary model that is shown in the same paper. However, I do not dare to claim this again because the authors have already denied it twice.

Now the only thing I like to say is that the author’s explanation does not make sense. The authors indeed used PAUP* in this paper. Also, PAUP* has been continually updated until now (see https://paup.phylosolutions.com/), although the authors write that the software has stopped updating in the reply letter.

Again, I do not dare to request the change because the models will not largely change the results (distances) and the usage of K2P is not incorrect (only it is not the best). However, only the thing I want to say is the authors’ explanation did not convince me.

OUR REPLY—I understand your point that the K2P is not the best choice for distance analysis. As you see, we had applied PAUP to construct NJ and ML trees for this study. However, it was almost eight years ago that we started to prepare this manuscript. After that time, we had no update on this software for the compatibility with PC system was not so good then. We will consider to re-set up this software in our PC and try to work on the distance analyses in next revised version if you insist.

(11) L247, L462, Issue of the tree prior:

In the previous comments, I recommend that the authors should try to use also the “Birth-Death” model as a tree prior (see Ritchie et al. 2017: Syst Biol, DOI:10.1093/sysbio/syw095) because the constant population size model is the coalescent model for a practically single population. The use of the latter simple model seems to be inconsistent with the population differentiation claimed in this paper. The authors explain something in the reply letter, which did not convince me. Because this is the authors’ paper, I do not dare to repeatedly comment on the same thing. However, I only want to say that the use of the constant population model alone may not be the best choice.

(12) L256, “The parameters of the substitution model GTR were estimated independently for each haplotype data.”: Although the authors note that they estimate the model parameters independently for each haplotype, I have completely no idea what they mean by that. Since this study uses only one gene (cytb), there must be only one set of parameters for GTR. Or, since it is a protein-coding region, it is possible to give three different models for each codon site. The authors should provide a clear explanation.

OUR REPLY—For the comments 11 and 12, we need more time to discuss the detailed of such technical suggestion. Please allowed us to reply formally in the next revised version.

(13) L320–330, Figure S1: The branch length should be reflected in the ML tree. Why did the authors remove the length information only from the ML tree? Strange.?

OUR REPLY—We had scaled the length of the ML tree in Fig. S1. Please recheck it again.

(15) L391–396: This part is a discussion, which should be moved to the “Discussion” part. Also, readers are surprised when Indochina is suddenly mentioned. It should be properly explained in the “Introduction” and the “Materials and methods” parts.

OUR REPLY—We moved L391-396 to “Discussion” part (L588-593 in the revised manuscript) following your suggestion. In the “Introduction” we introduced the distribution of A. normalis in Vietnam in L82, 88-90 in the revised manuscript.

(17) L605–, “since only four haplotypes were identified from 22 sequences that were 1,091 bp long and only one haplotype was found in the Dajia and Wu Rivers’ population”:

The number of haplotypes neither supports nor denies a different species. Should be rewritten.

OUR REPLY—We deleted the words “However, our data do not support the specimens collected from the Zhoushui River being a different species, however, since” to make the paragraph more readable.

(18) L627–629, “We suggest that both....in the future.”: What’s happened? Is this “suggestion” for A. amnis issue? How related? Should be rewritten.

OUR REPLY—In the manuscript we ended it with the A. amnis issue. We modified this sentence as “We suggest that more appropriate nuclear markers should be used to evaluate the validity of A. amnis in the future.”

Minor/Trivial points:

(31) L102: “comprehensive sampling efforts across the species’ entire range”  “comprehensive geographic sampling”. Redundant and not very precise (how about Vietnam?)

OUR REPLY—We had modified this sentence to make it more readable. (L101-104)

(45) L321, and Figure 2, “>90%”: As the authors may know that the Bayesian posterior probability is usually expressed as 0.00–1.00 as probability, not in %, although I do not dare to request to change it.

OUR REPLY—We had modified the Fig. 2 following your suggestion.

(46) L325–, “Abbreviations beside clades or sub-clades correspond to the sampling sites listed in Table 1.”: I cannot find such abbreviations.

OUR REPLY—They are I, II, III and Ia, Ib, Ic, and the description should be “Abbreviations beside clades or sub-clades correspond to the clades listed in Table 1.”

(47) L338, “the two a priori phylogenetic hypotheses that we examined”: There are four hypotheses. Also, the part “, isolated by the strait (Scenario 1 in Table 3) and drainage by river system (Scenario 2 in Table 3),” seems not clear. Please rephrase this sentence.

(48) L339–363: Replace “scenario” by “hypothesis”, as is in “Methods”. Confused.

(50) L341–343, “we found in both hypothesis... of the Hainan Island vs. mainland hypotheses”: Contradicted. The last part should be for only Hypothesis 1, not both.

(52) L348–, “Our second set of experiments...”: In fact, the part below describes scenarios (hypotheses) 2 and 3. What’s happened?

OUR REPLY—These hypotheses were really confusing. Very sorry. We had re-written this section and re-named the hypotheses (1a, 1b, 2a, 2b) in L340-346. Please give us your suggestion on it.

(54) L354, “low”: Are the 0.32773 and 0.51140 really low? Strange.

OUR REPLY—It was compared to another hypothesis (2b). We had modified the sentence. (L439)

(55) L358, “(Scenario 4 in Figure 3)”: In Figure 3, no word of “Scenario 4” appears. Strange citation.

OUR REPLY—It should be “Table 3” and we had corrected it.

(59) L378, “ FST values correspond to among individuals within populations.”: This is quite wrong.

FST in AMOVA is the variation component assigned to populations (without group division). So, 1–FST is the value that corresponds to the variation component among individuals within populations, isn’t it?

OUR REPLY—We had modified this sentence to make it more readable. (L359-361)

(68) L422, “A haplotype network is helpful in understanding the structure of phylogenetic relationships between haplotypes of Aphyocypris normalis among populations”: Should be rephrased. Strange. What is “the structure of phylogenetic relationships”? Phylogenetic relationships of what? “Between haplotypes”, or “among populations”?

OUR REPLY—We had simplified this sentence to make it more readable. (L373-374)

(69) L424, “two groups were erected”: What are these? Specify. First, these two groups should be defined (explained) based on the network. 

OUR REPLY—We separate these two group with red broken line in Fin. 3.

(105) L594–, “More comprehensive comparative studies of A. arcus on Hainan Island both on DNA and morphology are necessary in the future.”: Strange. In the sentence immediately above, the authors state the possible absence of this species in Hainan, so a comprehensive comparative study may not be possible.

OUR REPLY—Agree. We removed the words “on Hainan Island”.

--- end of comments

---

## [Decision Letter · Decision Letter 3]

19 Nov 2021

PONE-D-20-37420R3Phylogeography of Aphyocypris normalis Nichols and Pope, 1927 at Hainan Island and adjacent areas based on mitochondrial DNA dataPLOS ONE

Dear Dr. Jang-Liaw,

Thank you for submitting your manuscript to PLOS ONE. After careful consideration, we feel that it has merit but does not fully meet PLOS ONE’s publication criteria as it currently stands. Therefore, we invite you to submit a revised version of the manuscript that addresses the points raised during the review process.

We look forward to receiving your revised manuscript.

Kind regards,

Tzen-Yuh Chiang

Academic Editor

PLOS ONE

Reviewers' comments:

Reviewer's Responses to Questions

**Comments to the Author**

1. If the authors have adequately addressed your comments raised in a previous round of review and you feel that this manuscript is now acceptable for publication, you may indicate that here to bypass the “Comments to the Author” section, enter your conflict of interest statement in the “Confidential to Editor” section, and submit your "Accept" recommendation.

Reviewer #2: (No Response)

Reviewer #3: (No Response)

2. Is the manuscript technically sound, and do the data support the conclusions?

Reviewer #2: No

Reviewer #3: Yes

3. Has the statistical analysis been performed appropriately and rigorously? 

Reviewer #2: N/A

Reviewer #3: No

4. Have the authors made all data underlying the findings in their manuscript fully available?

Reviewer #2: (No Response)

Reviewer #3: Yes

5. Is the manuscript presented in an intelligible fashion and written in standard English?

Reviewer #2: Yes

Reviewer #3: Yes

6. Review Comments to the Author

Reviewer #2: The manuscript titled “Phylogeography of Aphyocypris normalis Nichols and Pope, 1927 at Hainan Island and adjacent areas based on mitochondrial DNA data” investigated the phylogeographic structure of A. normalis, in Hainan Island and mainland china. Unfortunately, I found the paper is poorly organized with some mistakes as well as concept misunderstanding, not suitable for publication at current form. The aim and the abstract are not well clarified. The Introduction did not focus on clear background and miss some important references, in need of further revision on previous studies of freshwater fishes in Hainan island. Discussion and Results must be improved by removing general statements and explaining the results obtained and comparing these results with genetic studies for the structure of mitochondrial DNA. In reading the manuscript, I encountered several problems that required the author's most careful attention. Some of the issues are semantic/conceptual, others technical. I try to make my comment as constructive as possible and hope the author finds it useful.

L108-L111, The title of manuscript is “Phylogeography of Aphyocypris normalis Nichols and Pope, 1927 at Hainan Island and adjacent areas based on mitochondrial DNA data”. The author mentions the phylogeographic structure of A. normalis in Hainan Island and mainland China. However, the authors gave an interesting conclusion as “We suggest that more appropriate nuclear markers should be used to evaluate the validity of A. amnis in the future（L644-L646）”. As the specimen of Aphyocypris normalis Nichols and Pope, 1927 were from Hainan Province, and “A. amnis” were sampled in Taiwan Province, what is the evidences? And is this relevant to the topic of this MS.

L129, Why does the author combine populations into 17 groups as AMOVA analysis? L236-L238 “we combined populations into 17 groups based on drainage basins or close geographic proximity (i.e., the following populations were combined: sites 1–2, 4–5, 7–8, 12–14, 15–17, 19–25, and A2–A5; see Fig. 1).”

L285, “15 (1.283%) on the Zhubi River, which was represented by only two specimens”, why not use all the 7 specimens in the Zhubi River, and the author use 7 individuals in the AMOVA analysis.

L348-L349, Table 2, Clade Ⅰ is an unresolved tree. Clade Ⅰa, Ⅰb and Ⅰc are unresolved trees and inappropriately defined as monophyletic group. This is quite doubtful.

L418-L423, The author shows the phylogenetic relationships between A. normalis and closely related cyprinid species. But the author does not mention the phylogenetic relationships in discussion.

L485-L488, The reference (71) mention the phylogeography of Rana sauteri (Ranidae). R. sauteri is an amphibian, not freshwater fish.

L488-L491, The freshwater fish and the amphibian are the same in ecology?

L496-L512 According the phylogeny and network, the results do not support the eight phylogeographic ranges of A. normalis. The author must present enough evidence to prove it and explain in detail.

L517-L519, The author must explain why the same river system can be divided into two groups. What is the reason for the distinction?

L527, Subclade Ib is unresolved tree.

L534-L542, There are many studies of freshwater fish focus on the phylogeographic role of Changhua River. Needs to added the latest refs.

L571, Subclade Ia is unresolved tree and not a distinct monophyletic Clade. How about the h46?

L572-L573, Subclade Ic is a distinct monophyletic Clade. How about the h1 and h3? Why didn't the author combine the h1 and h3 in Subclade Ic?

L616-L646, Why does the author discuss the other A. species in Taiwan province? not concerning to this MS.

Reviewer #3: I had access to the previous MS, to the reviewer points, and to the answers given to the reviewer.

Generally, it was not easy to follow what has happened during the review process. Nevertheless I tried to keep up.

I see the effort to adequately answer the points, but I also feel, that sometimes the answers are quite elusive. For example the authors are asking for 'more time to reconsider' suggestions.

The reviewer was asking to use up-to-date softwares, or models. As I see, the authors are willing to use updated software, still they did not use it. The journal is PlosOne having high standards. So, if I were the lead author of this manuscript I would immediately recompute the results using better model even if the data is 8 years old. I would also try the suggested models to see what is the difference of the outcomes of the reported and the suggested models.

Additionally, rewriting of the sentences sometimes remained as confusing as previously were. I suggest to consult with more colleagues, go through the entire manuscript. Involve new eyes into the quest for such sentences.

I suggest to do the polishing work.

7. PLOS authors have the option to publish the peer review history of their article (what does this mean?). If published, this will include your full peer review and any attached files.

Reviewer #2: No

Reviewer #3: No

---

## [Author Response · Author response to Decision Letter 3]

3 Mar 2022

The following are our answers to the comments from reviewers.

Reviewer #2:

L108-L111, The title of manuscript is “Phylogeography of Aphyocypris normalis Nichols and Pope, 1927 at Hainan Island and adjacent areas based on mitochondrial DNA data”. The author mentions the phylogeographic structure of A. normalis in Hainan Island and mainland China. However, the authors gave an interesting conclusion as“We suggest that more appropriate nuclear markers should be used to evaluate the validity of A. amnis in the future（L644-L646）”. As the specimen of Aphyocypris normalis Nichols and Pope, 1927 were from Hainan Province, and “A. amnis” were sampled in Taiwan Province, what is the evidences? And is this relevant to the topic of this MS.

OUR REPLY— In this work we selected A. moltrechti, a species close to A. normalis both in morphology and ecology, as an outgroup to discuss the phylogeny of A. normalis. There might be another better choice rather than A. moltrechti occurred somewhere around Hainan Island, however, the most suitable samples we have in hand was the A. moltrechti. The A. amnis is a sibling species of the A. moltrechti and we included it in the outgroup as well. We found that there was no obviously difference between A. moltrechti and A. amnis based on the DNA data of this work. We mentioned this opinion in the last end of this manuscript, to point out that more researches are necessary in the taxonomy of this group, even in the Eastern Asian ichthyological fauna. This is not a main goal of this research; however, it is a valuable issue to discuss for local ichthyology society, which the authors of this manuscript belong to. We are not dare to overthrow the validity of A. amnis in such short comments. We described what we found in this work. 

L129, Why does the author combine populations into 17 groups as AMOVA analysis? 

OUR REPLY— We combined populations into 17 groups based on drainage basins or close geographic proximity (L226-229).

L236-L238 “we combined populations into 17 groups based on drainage basins or close geographic proximity (i.e., the following populations were combined: sites 1–2, 4–5, 7–8, 12–14, 15–17, 19–25, and A2–A5; see Fig. 1).”

L285, “15 (1.283%) on the Zhubi River, which was represented by only two specimens”, why not use all the 7 specimens in the Zhubi River, and the author use 7 individuals in the AMOVA analysis.

OUR REPLY— These are two different scales for discussion. One is by “population (AMOVA)”, the other is by “site location (Hd)”

L348-L349, Table 2, Clade Ⅰ is an unresolved tree. Clade Ⅰa, Ⅰb and Ⅰc are unresolved trees and inappropriately defined as monophyletic group. This is quite doubtful.

OUR REPLY— We agreed with you that Clade Ia, Ib and Ic are not resolved well and inappropriately defined as monophyletic groups. So, we assigned them as sub-groups. The Clade I is supported by Bayesian and NJ analyses as a monophyletic group.

L418-L423, The author shows the phylogenetic relationships between A. normalis and closely related cyprinid species. But the author does not mention the phylogenetic relationships in discussion.

OUR REPLY—This section provided a rough overview of A. normalis and its siblings. Its most phylogeny close species is A. moltrechti, which is distributed in Taiwan Island, and we chose it as outgroup. 

L485-L488, The reference (71) mention the phylogeography of Rana sauteri (Ranidae). R. sauteri is an amphibian, not freshwater fish.

OUR REPLY— The reference provides a case that “different climate and rainfall causing diverse breeding strategies”, no matter what animal it is. We cited several works focusing on animals other than fish in this manuscript, such as earthworms [20], a freshwater crab [21], freshwater fishes [22-29], a babbler [30], the peacock pheasant [31], deer [32], and ferns [33, 34].

L488-L491, The freshwater fish and the amphibian are the same in ecology?

OUR REPLY— The larval stage of amphibians is limited in freshwater habitats and isolated by lands between rivers, just like the freshwater fish. Yes, they can move on lands in their adult stage, anyway, their phylogeographic patterns are affected by freshwater habitats deeply. The freshwater fish and the amphibian are not the same in ecology totally but they are the same for being a good model species to discuss that how the river systems shape the fauna. 

L496-L512 According the phylogeny and network, the results do not support the eight phylogeographic ranges of A. normalis. The author must present enough evidence to prove it and explain in detail.

OUR REPLY— In this discussion, we did add subjective opinions to the discussion of zoogeographic zoning of this fish based on the climate zoning of Hainan proposed by He and Zhang [66]. This model is more reasonable than the well-accept one “shaping by the isolation of rivers” in this case. We had discussed this opinion in this manuscript. 

L517-L519, The author must explain why the same river system can be divided into two groups. What is the reason for the distinction?

OUR REPLY— Some river systems in Hainan are large and complex. The isolation mechanism of freshwater animals is not clear and well understood. For example, the main stream of river systems itself could be a barrier for some small freshwater fish species which hide in small tributaries. And, climate might be a force to shape the characters of genetic structures of freshwater animals. In the latter case we provide a case on an anura Rana sauteri [75] for refernce.

L527, Subclade Ib is unresolved tree.

OUR REPLY— The grouping of this clade is supported by BI/NJ/ML. We considered this grouping in subclade level.

L534-L542, There are many studies of freshwater fish focus on the phylogeographic role of Changhua River. Needs to added the latest refs.

OUR REPLY— We cited some latest works focus on phylogeography of Hainan freshwater fish in the revised manuscript following your suggestion. 

L571, Subclade Ia is unresolved tree and not a distinct monophyletic Clade. How about the h46?

OUR REPLY— Subclade Ia is supported by Bayesian analysis as a monophyletic clade among Clade I. H46 is outside from Subclade Ia. 

L572-L573, Subclade Ic is a distinct monophyletic Clade. How about the h1 and h3? Why didn't the author combine the h1 and h3 in Subclade Ic?

OUR REPLY— The supporting of grouping between h1, h3 and other Subclade Ic (by all three analyses) is relative weaker than that among Subclade Ic (only by Bayesian analysis). Besides, all the Subclade Ic are distributed in Dong R. (Pearl R.), mainland China; the h1 and h3 are in Nanyang R., Hainan Is. The grouping of Ic are supported both by DNA data and their distribution. 

L616-L646, Why does the author discuss the other A. species in Taiwan province? not concerning to this MS.

OUR REPLY— Please refer to our answer reply to the first question. (L108-L111, The title of manuscript……)

Reviewer #3: 

The reviewer was asking to use up-to-date softwares, or models. As I see, the authors are willing to use updated software, still they did not use it. The journal is PlosOne having high standards. So, if I were the lead author of this manuscript I would immediately recompute the results using better model even if the data is 8 years old. I would also try the suggested models to see what is the difference of the outcomes of the reported and the suggested models.

Additionally, rewriting of the sentences sometimes remained as confusing as previously were. I suggest to consult with more colleagues, go through the entire manuscript. Involve new eyes into the quest for such sentences.

OUR REPLY—In this version we had asked Dr. Graham L. Banes, associate scientist in the Wisconsin National Primate Research Center at the University of Wisconsin–Madison for proof-reading the revised manuscript. We sincerely thank you for your expectation and assistance in this work.

--- end of comments

---

## [Decision Letter · Decision Letter 4]

12 Apr 2022

PONE-D-20-37420R4Phylogeography of Aphyocypris normalis Nichols and Pope, 1927 at Hainan Island and adjacent areas based on mitochondrial DNA dataPLOS ONE

Dear Dr. Jang-Liaw,

Thank you for submitting your manuscript to PLOS ONE. After careful consideration, we feel that it has merit but does not fully meet PLOS ONE’s publication criteria as it currently stands. Therefore, we invite you to submit a revised version of the manuscript that addresses the points raised during the review process.

We look forward to receiving your revised manuscript.

Kind regards,

Tzen-Yuh Chiang

Academic Editor

PLOS ONE

Reviewers' comments:

Reviewer's Responses to Questions

**Comments to the Author**

1. If the authors have adequately addressed your comments raised in a previous round of review and you feel that this manuscript is now acceptable for publication, you may indicate that here to bypass the “Comments to the Author” section, enter your conflict of interest statement in the “Confidential to Editor” section, and submit your "Accept" recommendation.

Reviewer #2: (No Response)

Reviewer #3: All comments have been addressed

2. Is the manuscript technically sound, and do the data support the conclusions?

Reviewer #2: Partly

Reviewer #3: Yes

3. Has the statistical analysis been performed appropriately and rigorously? 

Reviewer #2: N/A

Reviewer #3: Yes

4. Have the authors made all data underlying the findings in their manuscript fully available?

Reviewer #2: No

Reviewer #3: Yes

5. Is the manuscript presented in an intelligible fashion and written in standard English?

Reviewer #2: Yes

Reviewer #3: Yes

6. Review Comments to the Author

Reviewer #2: The authors made an interesting topic in this MS, and I especailly apprecaite their detailed samling work and analysis on Wuzhi & Yinggeling, beyond the formers' study. However, the authors did not give more discussion on their differences and possible explanation. Here is another work focus on the same topic, pleased read it and discuss. Wei et al. 2021. Population genetic structure and phylogeography of Aphyocypris normalis on Hainan island and its neighboring regions. Chin J Appl Environ Biol, 2021, 27 (1): 191-199.DOI: 10.19675/j.cnki.1006-687x.2020.01025

The authors gave interesting workd by proposing hypothesis in explaining the phylogenetic pattern of A. normalis. As the resecrch focus in this MS,the details of the hypothesis as well as why giving such a hypothese are need, e.g., in hypothesis 2, the ciatations on why draniage or climate could be used as the controlled factor; the chracters of 8 climatic zone and its relation to freshwater fauna; what are the 12 drainages and their geological developmental traits; either in introduction or in discussion.

Line 487. what does the logic of this sentence "Nonetheless, .. fit...". Thus, did the results by Cytb fit/unfit the division of climate? This was a really problem need the authors to clarify seriously.

The result of 8 pholographic ranges was one of the two importants conclusion of this MS, seen from Abstract. Unluckily, I am puzzled by the authors work and explanation.

First, the details of samling sites needed illustration: climatic zone, river, corresponding to hypothesis 2b as sites (Just seen from figure 1 and figure 5a, I am not sure whether sample sites cover all the 8 climatic zones). Maybe still one more info, why the significant differences in site number in different climatic zones (only 1 site in a climate zone but many in another).

Second, seen from the figure 5a,b, the climate zone in Hainan Island was not concistent with the result of eight phylogeographic divisions. For example, climate 1 was divided into E and SE, climate zone 6 covered W and west part of SW; that means,the climatic division and the phylographic divisions intersected. As the authors suggested, the results in eight pholograhic ranges were moderated from climatic zone, so how to mederated, and the criteria? For example, the authors cancaelled the Subcalde (but in subgroup) in Clade I, then would it be suitabale to used subgroup to determine phylographical division?

The anthors took the results granted, too far-fetched without a equal criteria,and seemed eliberated to give such a result only aiming to match the 8 climatic zones for a possible explanation.

Third, the as an important conclusion, the author needed to give direct and clear result on the eight ranges by cytb, possibly directly marked in figure 2 or figure 3.

Moreover, adjacent areas were also inportant in this paper (the word also in the title), so the spatial pattern of A. normalis (clade II) also needed clarified and explanation, a 9th range or more? also controlled by climate or other factors? Should the factors controlling the phylographic range of A. nomalis be same or different in Hainan Island or adjacent areas?

In clade I, according to answers to the comments, the authors cancelled the validity of subclade and used subgroup, please giving personal difination on subgroup; otherwise, it is still cause confusion, as "group is monophyletic".

Moreover, in discussion the author still kept Subclade(since Line 562). I do think the author need to revised the paper in a throughout instead of only anwere the reviewers' comments and just simple corrections.

Why add a part of comment on A. moltrechti, and discussed a validity of a new species A. amins? It is non busisness of the thesis of this paper. It needs another complicated work in another paper, using works following by taxonomic standarss, including morphology, molecular,evolotion,as well as possible geological and ecological evidences, but not only in discussion.

Reviewer #3: The authors answered the fellow researchers' questions as well, they rewritten the manuscript.

I suggest the manuscript for publication.

7. PLOS authors have the option to publish the peer review history of their article (what does this mean?). If published, this will include your full peer review and any attached files.

Reviewer #2: No

Reviewer #3: No

---

## [Author Response · Author response to Decision Letter 4]

31 May 2022

We deeply appreciate the reviewer’s great comments and suggestions on this work. The manuscript has been major revised accordingly as possible we can, and it had been modified completely almost. We accept almost all suggestions the reviewers offered. The following are our answers to the comments. 

Reviewer #2: 

>The authors made an interesting topic in this MS, and I especailly apprecaite their detailed samling work and analysis on Wuzhi & Yinggeling, beyond the formers' study. However, the authors did not give more discussion on their differences and possible explanation. Here is another work focus on the same topic, pleased read it and discuss. Wei et al. 2021. Population genetic structure and phylogeography of Aphyocypris normalis on Hainan island and its neighboring regions. Chin J Appl Environ Biol, 2021, 27 (1): 191-199.DOI: 10.19675/j.cnki.1006-687x.2020.01025

OUR REPLY—Thank you for your useful information. We update our manuscript with the recent reference you provided (it should be Li et al. 2021). We reconsidered the possible isolation mechanisms occurred by the Yinggeling and Wuzhishan Mountains in accord with Li et al. s’ (2021) work in LN 476 -481.

>The authors gave interesting workd by proposing hypothesis in explaining the phylogenetic pattern of A. normalis. As the resecrch focus in this MS,the details of the hypothesis as well as why giving such a hypothese are need, e.g., in hypothesis 2, the ciatations on why draniage or climate could be used as the controlled factor; the chracters of 8 climatic zone and its relation to freshwater fauna; what are the 12 drainages and their geological developmental traits; either in introduction or in discussion.

OUR REPLY—We had modified the manuscript following your suggestion. We introduced the climate division in Introduction by adding “Moreover, we introduced a climate division hypothesis [36] to discuss the possible isolation mechanism on freshwater fauna in Hainan Island” (LN103-104) in introduction. In LN54-60, we had a short introduction on Hainan’ s rivers already. We also marked the characters of the climate divisions following He and Zhang [36] in LN 492-498 following your suggestion.

>Line 487. what does the logic of this sentence "Nonetheless, .. fit...". Thus, did the results by Cytb fit/unfit the division of climate? This was a really problem need the authors to clarify seriously. 

OUR REPLY—Our cytb data fit their division of climate with high similarity. We had removed the “Nonetheless,” from the sentence.

>The result of 8 pholographic ranges was one of the two importants conclusion of this MS, seen from Abstract. Unluckily, I am puzzled by the authors work and explanation.

First, the details of samling sites needed illustration: climatic zone, river, corresponding to hypothesis 2b as sites (Just seen from figure 1 and figure 5a, I am not sure whether sample sites cover all the 8 climatic zones). Maybe still one more info, why the significant differences in site number in different climatic zones (only 1 site in a climate zone but many in another).

OUR REPLY—Please see the Figure 5. In 5a, the original climate division proposed by He and Zhang (1985) was presented; In 5b, we marked our sampling sites on a modified phylogeography-climate map. The sampling sites were uneven on these modified phylogeography-climate zones, but not “significant differences” as you mentioned; the numbers of sampling sites are 1 (1 zone), 2 (1 zone), 3 (3 zones), 4 (2 zones) and 5 (1 zone). We collected all the used specimens by ourselves and we visit all the habitats (sampling sites) displayed in this manuscript. In general, it is reasonable to be “uneven on sampling sites” for a work used materials collected from wild, for we cannot control the nature distribution pattern of the species we’re working on. We can say we did our best on sampling in the phylogeographic research of this fish.

>Second, seen from the figure 5a,b, the climate zone in Hainan Island was not concistent with the result of eight phylogeographic divisions. For example, climate 1 was divided into E and SE, climate zone 6 covered W and west part of SW; that means,the climatic division and the phylographic divisions intersected. As the authors suggested, the results in eight pholograhic ranges were moderated from climatic zone, so how to mederated, and the criteria? For example, the authors cancaelled the Subcalde (but in subgroup) in Clade I, then would it be suitabale to used subgroup to determine phylographical division?

The anthors took the results granted, too far-fetched without a equal criteria,and seemed eliberated to give such a result only aiming to match the 8 climatic zones for a possible explanation.

OUR REPLY—We agree with you point. Yes, in this work we proposed an incomplete, premature hypothesis based on the result of only one species. We do need more evidence to solid the phylogeography-climate zones hypothesis of Hainan island. However, it is difficult to be “complete” for a novel idea being proposed in the first beginning. In this case, we do not have any references to support our phylographical-climate division. This work is the first one to link ecological climate information and DNA phylogeny data in the species on Hainan Island. We are looking for more discussion, both agree and dis-agree ones, on this issue in the future. 

>Third, the as an important conclusion, the author needed to give direct and clear result on the eight ranges by cytb, possibly directly marked in figure 2 or figure 3.

Moreover, adjacent areas were also inportant in this paper (the word also in the title), so the spatial pattern of A. normalis (clade II) also needed clarified and explanation, a 9th range or more? also controlled by climate or other factors? Should the factors controlling the phylographic range of A. nomalis be same or different in Hainan Island or adjacent areas?

OUR REPLY—The readers can connect the result on the eight ranges and cytb data by the linkage of clade/subclade identification. Please check the Fig 2 for the phylogeny position of each clade/subclade and check the Fig 5b for the location of each clade/subclade in the phylographical-climate ranges. Beside, we made a new Figure S1 to show the four division hypotheses. 

For the phylographic range of A. nomalis in adjacent areas out of Hainan Island, it is a huge issue. Our data/sampling effort is far from “enough” to have a clear view on the phylogeography of A. nomalis in this area. For example, the Pearl river system is very complex with long ranges and abundant tributaries, and we only have 24 samples collected from 4 sites in this river system. It could be (and should be) a complete research topic to discuss the phylogeography of A. nomalis or any other species in Pearl River system. According to our limited sampling effort in this area, we only can discuss the relationship between southern China and Hainan’s populations (a.k.a., the isolation mechanism of the Qiongzhou Strait), but detailed phylogeographic comparison of populations of each river system. If you insist, we can modify the title of this manuscript by removing the “and adjacent areas”. 

>In clade I, according to answers to the comments, the authors cancelled the validity of subclade and used subgroup, please giving personal difination on subgroup; otherwise, it is still cause confusion, as "group is monophyletic".

Moreover, in discussion the author still kept Subclade(since Line 562). I do think the author need to revised the paper in a throughout instead of only anwere the reviewers' comments and just simple corrections.

OUR REPLY—We did not use “subgroup” in our manuscript. 

>Why add a part of comment on A. moltrechti, and discussed a validity of a new species A. amins? It is non busisness of the thesis of this paper. It needs another complicated work in another paper, using works following by taxonomic standarss, including morphology, molecular,evolotion,as well as possible geological and ecological evidences, but not only in discussion.

OUR REPLY—This work reports the phylogeography of a cyprinid fish distributed mainly in the southern part of China and Hainan Island. As you mentioned, the comment on outgroup species A. moltrechti complex was not priority of this work. However, the argument of the validity of A. amins indeed exists and no researchers pay efforts on it before for the “new species” can be found only in a very small stream, which means it is difficult to write a good paper with limited sample size of this species. The argument becomes not a priority for the researchers who need SCI papers to keep their academic job (or you can say, to survive in the reality world... ).

We believe that it is the responsibility of local ichthyologists to establish/correct ichthyological taxonomy in their land. We feel embarrassed that we have not been able to make a complete statement on this issue in a single paper currently. Herein we integrate some previously known but ignored information [85] with few new data we provide in this work to emphasis to this view and put it forward again. We do not expect to have a complete statement to reply to this issue, but we hope to remind Taiwanese ichthyologists that this issue is controversial, and we hope that there will be more complete reports on this issue in the future.

Reviewer #3: The authors answered the fellow researchers' questions as well, they rewritten the manuscript.

I suggest the manuscript for publication.

OUR REPLY—Thank you very much!

5. Review Comments to the Author (minimum 200 characters)

This manuscript by IS Chen and NH Jang-Liaw is the fourth revised version of PONE-D-20-37420, which reports the mtDNA phylogeography of a cyprinid fish, Aphyocypris normalis, commonly distributed mainly in the southern part of China including Hainan Island, based on an extensive geographical sampling. I have read this manuscript four times (except for the last one before the current manuscript).

Frankly, I am surprised at the number of problems that still remain in the current manuscript (5th round!!) and especially at the lack of care taken by the authors in the structure of the paper. Since this is probably my last comment, I would very much hope that the author will properly understand the meaning of my comments and make adequate revisions for the acceptance of this laborious work. I look forward to reading this paper after it is properly revised, completed and published.

>(1) L218–: The explanation for AMOVA analysis (models compared) is difficult to understand and should be improved.

The major problem is that the division of the regions according to Hypothesis 2b is not indicated in the Methods nor in the Results, but appears only in the Discussion. It is understandable that Fig. 5a cannot be repeated in the text, so perhaps the most reasonable way to do would be to illustrate all four models (1a, 1b, 2a, 2b) as the Supplementary figure. This would not only improve the fact that the division under Hypothesis 2b is not clear to the reader until the end of the Results, but it would also allow the reader to visually understand the differences between the models, which is complicated by the words alone.

OUR REPLY—We made a new figure set (Figure S1) to explain the four hypothesis models (1a, 1b, 2a, 2b) following your suggestion. 

>Also, the below minor problems should be improved:

(1-2) L223, “mentioned”  “explained” 

(1-3) L223–, “To evaluate movement...”: I cannot understand what is beginning to be explained here. I guess that the sentence explains the hypotheses 1a and 1b first. If yes, the sentence should be like “For the hypothesis 1a, all sampled populations were combined into two groups representing the localities of their origin to evaluate movement across the strait. Alternatively, to evaluate movement between river systems (1b), we...” 

(1-4) L230, “Moreover...”: This is not understandable. It should be, e.g., “For the hypotheses 2a and 2b, we evaluated movement...”. Also, remove “also” in L232. 

(2) L116: “across the entire geographic range of the species”: Conflict to the statement in L85–88 (i.e., not entire). For example, add “almost”? 

OUR REPLY—We had modified the manuscript following your comments.

>(3) L198–201: “We assumed the constant population size setting as a coalescent tree prior, which is...”:

This setting should be too simple to apply to subdivided populations with complex history even in a species. If I were the author, I would compare the results under different models (e.g., Birth–Death model, as suggested by Ritchie et al. 2017: https://doi.org/10.1093/sysbio/syw095) and show the robustness of the results. However, I would like to leave this point to the authors’ decision. (Oh, I noticed that I have made the same comment at least three times... Every time I seem to find this problematic...) 

OUR REPLY—In our opinion, the phylogeography research is based on a number of assumptions that can only be known by inference and not repeatable (eg; past geological chronological changes, the happened evolution procedures…). The analysis and statistics of this kind of research are also constructed in many hypotheses, and various model hypotheses are constantly being proposed. However, there may be only one true (nature) evolution history. We understand that the goals of all “survived” models are trying to revise their inferences to the closest version of reality/nature, and no single model can fit all assumptions. As you suggest, comparing different models to get the most likely hypothesis from them is the general recommended drafting thinking. But we don't think the purpose of this work is to discuss the modeling/methods, and such comparisons would make this manuscript be out of focus. In addition, in our experience, we found that the simplest model always can lead to a similar inference as other new models. Therefore, we tend to choose simpler models or parameters instead of new models companied with many parameters that we really do not understand completely. In other fields of research, such as microbiology, where the model is not the key to the discussion of their results; simple analysis as MEGA without modeling can be acceptable (eg, https://link.springer.com/content/pdf/10.1186%2Fs42483-021-00103-z ). The authors can focus the manuscripts on their findings rather than on differences in methodological models. 

>(4) L253–259: Move to the “Materials and methods”. These are not results. Especially, L255–257 are just repetition of materials. The result can start from L259, e.g., as “All sequences obtained from 127 Aphyocypris normalis and 22 A. moltrechti specimens were 1,091 bp in length” 

OUR REPLY—We had modified the manuscript following your suggestion.

>(5) L295, Figure S1: The branch length should be reflected in the ML tree. Why did the authors remove the length information only from the ML tree? Very strange. I may have made this point before, too.

OUR REPLY— The ML tree we provided before showed a better shape. However, we replaced the ML tree following your suggestion.

>Also, in the legend of Figure S1, the support values should be explained. 

OUR REPLY—We had modified the legend of Figure S2 following your suggestion.

>(6) L327–333: What is this part at all? This is not the results of “Phylogenetic relationships” and just what is explained in the method. Completely unnecessary. Should be deleted. 

OUR REPLY—We had modified the manuscript following your suggestion.

>(7) Table 3: The place is quite inadequate. This table of AMOVA should be put in the “Population structure” part (L401–), so the table number should be renumbered (it will become Table 4). 

(8) L402–407: Why are these results of phylogenetic relationships placed in the “Population structure” part? Completely incomprehensible. Move to the last or adequate part of “Phylogenetic relationships”. 

OUR REPLY—We had modified the manuscript following your comments.

>(9) L408–410: Confused. Why is the matter of Fst explained here? No meaning. It DID NOT “imply few gene flow between tested groups”. Fct is the statistic to be focused. However, for Fct, it is not significant in all models (1a).

OUR REPLY—We modified the sentences following your comment; and we had more statements on FCT in SAMOVA. 

>(10) L418, “but also not be true”: Too strong expression. This kind of models is not completely true or wrong. Rephrase it. 

OUR REPLY—We modified it to “but also might not be true.”

>(11) L423–432, “The FCT value of Hypothesis 2a....within groups (Table 3): This section is confusing and inappropriate for context and explanation.

For example, these 10 lines can be... “(Delete the 1st sentence) The results of Hypothesis 2a, as well as Hypothesis 1b, did not strongly support genetic isolation between river systems. The last hypothesis (2b), dividing the region into eight biogeographical areas based on climate revealed that most of the variation was between groups (70.15%; Table 3[4]), which indicates that the hypothetical biogeographic region division is a strong barrier to gene flow. (Delete the last sentence because it is just an unnecessary description shown in Table.)”

OUR REPLY—We had modified the manuscript following your comment.

>(12) L477–479 (such as the elevation of different habitats after climatic events....strategies): Sorry, but I don't understand this part at all.

OUR REPLY—In a former study of the corresponding author, the phylogeographic pattern of a frog Rana sauteri is possibly formed in a large part by reproduction isolation formed by difference of climate characters between habitats, not by geographic barriers only [76]. It reminded us to check possible mechanisms which formed the phylogeography of animals other than geographic barriers. In this study, we checked the climate pattern of Hainan Island and tried to introduce it to our discussion. 

>(13) L532, “they are different from other populations”:

The authors seem confused between the groups and phylogenetic relationships of mitochondrial DNA haplotypes and the cohesion and relationships of populations. Although the results of mtDNA can show evidence of river capture, it is still unclear that the fish in this region compose a historical population group and it is different from other populations. At a minimum, the expression should be softened as, e.g., “They are possibly different from...”.

Furthermore, it should be clearly stated somewhere (e.g., here), “However, because our inferences were based on mtDNA only, nuclear genome data should be examined for further information on population relationships and history.", or like this.

This kind of confusion is found also in L579 (“a unique phylogenetic group (Clade III) composed of populations of...”)

OUR REPLY—Good suggestions. We had modified the manuscript following your comments.

>(14) L577–, “in A, normalis”: Too narrow. How about replacing by “even in freshwater fish”?

(15) L586–594: “Huang et al. [25]...”: It is better to remove this paragraph.

In my previous comment to indicate this “inconsistency”, I made a mistake at the end (Sorry!), and this manuscript seems to inherit that mistake. If Huang et al. used the 2%/Myr clock (twice faster clock), the estimated time should become short (not long!), shouldn’t it? So, although the explanation in the Huang et al. is quite unclear and the result is inconsistent with the present one, I think those should be left alone without mention.

OUR REPLY—We deleted the section following your suggestion.

Minor/Trivial points:

>(16) L21: Delete commas before and after “Aphyocypris normalis”. Here “the freshwater fish” is not equal to “Aphyocypris normalis” but used adjectivally.

(17) L36, “LGM”: Spell out (i.e., “Last Glacial Maximum”)

(18) L157, “MEGAX”  “MEGA X” (insert a space)

(19) L183, “and is shown herein (see Fig. 4)”: Should be removed. Unnecessary.

(20) L236: Change paragraph before “In addition”. The topic is changed here from AMOVA to SANOVA.

(21) L238, “or not”: Should be removed. Unnecessary.

(22) Several sentences in the results are sporadically written in the present tense, but should be written in the past tense (e.g., L259, 288, 291, 360, 362, 363, 542...). Check throughout.

OUR REPLY—We had modified the manuscript following your comments.

>(23) L323, “haplotype”  “sequence”

OUR REPLY—It should be “haplotype”. The number of “sequences” (all samples’ sequences) is different from “haplotype”. We used haplotype data in that section, not all samples’ sequences. 

>(24) L323, “0.75...”: Make the text so that it is clear which is the value of which clade.

(25) L324, “inferred from haplotypes”: Remove. Unnecessary.

(26) L325, L423: Insert a space between words (figures).

(27) L340–, The second column for Hypothesis 2b: “Hypothesis”  “Hypothetical”?

(28) L357–358, “A haplotype network is...relationships.”: Completely unnecessary. Should be removed.

(29) L370, “In our haplotype network”: Unnecessary and Should be deleted. “The haplotype h1 was...”

(30) L376, “five”  “five or six”

(31) L379, “each clade”  “haplotypes”. Not “network for each clade”.

(32) L390–391: Indicate which values are for which clades.

(33) L411–, “hypothesis” (Methods) or “Hypothesis” (Results)? Either one should be used.

(34) L474, “concludes”  “conclude”

(35) L475, “by the geographical”  “solely by the geographical”: It is wrong to deny it completely.

(36) L487, “Nonetheless, their divisions”  “Their divisions”. The context is not connected at all with the word "Nonetheless."

(37) L579, “Moreover”: Delete. The context is not connected at all with the word "Moreover" in the changed paragraph.

(38) L582, Cite a reference [19].

OUR REPLY—We had modified the manuscript following your suggestions.

>(39) L586, “MRCAs”  “tMRCAs”

(40) L592, “may”  “should”

OUR REPLY—The section had been removed following your another suggestion.

>(41) L598, “between A. normalis and A. arcus ... collected from southern mainland China and Hainan Island.”: Which was caught where. Or in both? Clearly explain it.

OUR REPLY—From both. We had modified the manuscript following your comment.

--- end of comments and replies

---

## [Decision Letter · Decision Letter 5]

1 Sep 2022

PONE-D-20-37420R5Phylogeography of Aphyocypris normalis Nichols and Pope, 1927 at Hainan Island and adjacent areas based on mitochondrial DNA dataPLOS ONE

Dear Dr. Jang-Liaw,

Thank you for submitting your manuscript to PLOS ONE. After careful consideration, we feel that it has merit but does not fully meet PLOS ONE’s publication criteria as it currently stands. Therefore, we invite you to submit a revised version of the manuscript that addresses the points raised during the review process.

We look forward to receiving your revised manuscript.

Kind regards,

Tzen-Yuh Chiang

Academic Editor

PLOS ONE

Reviewers' comments:

Reviewer's Responses to Questions

**Comments to the Author**

1. If the authors have adequately addressed your comments raised in a previous round of review and you feel that this manuscript is now acceptable for publication, you may indicate that here to bypass the “Comments to the Author” section, enter your conflict of interest statement in the “Confidential to Editor” section, and submit your "Accept" recommendation.

Reviewer #4: (No Response)

Reviewer #5: (No Response)

2. Is the manuscript technically sound, and do the data support the conclusions?

Reviewer #4: Partly

Reviewer #5: Partly

3. Has the statistical analysis been performed appropriately and rigorously? 

Reviewer #4: Yes

Reviewer #5: Yes

4. Have the authors made all data underlying the findings in their manuscript fully available?

Reviewer #4: Yes

Reviewer #5: Yes

5. Is the manuscript presented in an intelligible fashion and written in standard English?

Reviewer #4: No

Reviewer #5: Yes

6. Review Comments to the Author

Reviewer #4: I think this is a very interesting system with excellent regional sampling. The broad questions are compelling, and I think the methods and sampling are appropriate to answer some of these broad questions of population structure across the region. I think the results that the authors found are interesting and a valuable contribution. However, I do have some serious concerns regarding the clarity of this paper and the over-interpretation of the findings.

General comments

1: The writing needs a thorough editing for brevity and clarity.

2: Additional natural history of the species is needed, what's the life-cycle, what's the global distribution, habitat preferences, etc.

2: Overall I think the paper needs to be simplified, it will have a cleaner and more compelling result and conclusion that way and the readership will be able to follow the flow of the paper better. After a number of reads through the paper I see the following flow of information:

- We provide thorough geographic sampling of a freshwater fish to address two questions

1: what, if any, genetic structuring is there among populations/drainages?

2: does genetic structuring follow biogeographic expectations? (and outline these clearly)

- We used the following methods to identify genetic structure among populations

1: phylogenetic analysis

2: haplotype network analysis

3: clustering analysis

- We conclude that one drainage on Hainan Island forms a distinct genetic group (very interesting result!), but little differentiation/structure was identified across the other samples and between Hainan and the mainland.

I think keeping a clear and clean flow of information in mind would significantly increase the strength of this paper. In doing so, I think the authors need to be exceedingly careful to not read into the genetic patterns too much and to not over interpret their results.

3: Following on the end of the above comment, I think the genetic data is over interpreted. The authors do no explicit tests of gene flow but talk extensively about evidence for gene flow. Note that there are other processes that result in low divergence (recent colonization, large population sizes and slow/low genetic drift, conserved genetic regions under stabilizing selection, and yes, ongoing or recent gene flow). The data are analyzed solely to assess population level genetic structure, the interpretation of results should be restricted to this. It is an interesting question and the authors have good sampling to ask the question of what is the genetic structure of the species across the sampling, it will be a much stronger paper and argument if the results and discussion remain focused on describing the patterns of genetic structure of the sampling. Interpretation of gene flow and historical biogeography should be limited, if not completely removed. The interpretation of these data, based on the data and the analyses, should be restricted to commenting on population-level patterns of genetic divergence/similarity. The results, in my opinion, have two compelling conclusions, there is structure within Hainan Island with samples from Clade III (SW region, Hainan) showing a distinct genetic background suggestion historical isolation, and there is little structure between populations on the remaining drainages of Hainan and the nearby mainland of China. These are very interesting and well-supported results, and it's the strength of the authors' sampling to identify these patterns. In my opinion, over-interpretation of these patterns does nothing but weaken the argument and conclusions.

4: There needs to be more historical geological information in the introduction. In order to formulate the questions that the authors are asking, the reader needs to have a description of the geological history of the island and of patterns found in other organisms, both terrestrial and freshwater. This needs to be included in the introduction in order to support the questions being asked. Moving the information from lines 492 to 499 into the introduction would make a huge difference, expanding on this geologic history and providing evidence shown in other organisms would be extremely helpful.

5: I found it very difficult to interpret the haplotype-based tree figure. I understand what was done, and that it is essentially a polarized network with phylogenetic-focused analyses of node support. However, in my opinion, it provides very little information additional to the haplotype network. I would suggest combining the haplotype tree and network into one figure with two panels as these two visuals are very complementary. I found it extremely difficult to interpret the results in a geographic context without a phylogenetic analysis where individual samples were the tree tips, then some geography (maybe the group identities) were overlaid onto this tree. I got there eventually through reading the haplotype distribution portion and interpreting the clade distribution from the maps. However, this should be a result that is clearly presented visually. Running the same phylogenetic analyses that you did, but where individual samples represented tips would be extremely helpful. One could code tip colors to correspond with the map colors etc., for ease of interpretation. I think this addition would provide much needed clarity and strength to the conclusions.

Specific Comments

Throughout: be consistent on how you refer to Cytb, I suggest using "Cytb" throughout

Throughout: need to clearly outline early on how you define group/site/population, and ensure consistent use

Title: add some higher-level classification to the title, e.g., "Teleostei: Cyprinidae"

Here on, line numbers refer to lines in version 1

Line 21: replaces populations with sites

Lines 40–46: I'd remove this paragraph, I think starting at line 48 is perfectly fine

Line 97: remove "thoroughly" but add "with dense geographic sampling across the island" following "Hainan Island"

Methods: need to add exactly how the F-statistics were calculated

Table 1. Move to supplemental

Lines 245 & 246: remove, end paragraph with "Zhubi River."

Line 232: Sentence starting with "Eighty-three..." belongs in methods, not results

Line 253: Sentence starting with "A high number..." seems incomplete

Figure 2 caption, line 281: Be clear what the numbers at the nodes refer to, bootstrap support values is good, but the numbers presented by the Bayesian inference should be posterior probabilities (I believe)

Line 291: Sentence starting with "Standard error" should be moved to methods

Line 393: move this section to the end of Discussion, as such it breaks the flow up notably

Line 492: as noted above, this section (492 to 499) would fit much better in introduction

Reviewer #5: The authors of the manuscript entitled Phylogeography of Aphyocypris normalis Nichols and Pope, 1927 at Hainan Island and adjacent areas based on mitochondrial DNA data sequenced mtDNA from 127 specimens and created a phylogenetic tree. The results indicated 3 major clades- one with wide geographic distribution and 2 that were more limited, however no obvious geographic barriers exist between them. There also appeared to be 8 biogeographic regions of A. normalis that correspond to climate. Based on results of the phylogenetic tree, The authors conclude that A. normalis diverged between 1.05-0.16 Ma, before the temporary land bridge existed across the Qiongzhou Strait.

Line 184- can you please clarify- were the separate runs combined? Did they converge?

Line 207- can you provide more information about parameters of the model were selected- tree prior, clock model, etc. Was any model selection performed? Are you using a model that was reported previously.

Line 212- please check the references throughout. This reference was not correct.

Line 544 subclade Ib AND Subclade III

Line 554 BY geographic barriers

Line 557-577- how do your phylogeographic regions differ by the ones described by He and Zhang.

Line 605- can you suggest any alternative explanations?

Line 637- and BECOME isolated

7. PLOS authors have the option to publish the peer review history of their article (what does this mean?). If published, this will include your full peer review and any attached files.

Reviewer #4: No

Reviewer #5: No

---

## [Author Response · Author response to Decision Letter 5]

19 Oct 2022

Again, we deeply appreciate the reviewer’s great comments and suggestions on this work. However, we were surprised to find that the comments of the first reviewer (#4 reviewer) were based on the original version of our manuscript, which was submitted to PLOS ONE at the end of 2020, and has gone through 5 rounds of review so far. We appreciate the valuable comments provided by the #4 reviewer, nevertheless, we are unable to reply his comments for which are not suitable to discuss our latest submitted version.

The response to another reviewer’s comments are listed below:

Reviewer #5: The authors of the manuscript entitled Phylogeography of Aphyocypris normalis Nichols and Pope, 1927 at Hainan Island and adjacent areas based on mitochondrial DNA data sequenced mtDNA from 127 specimens and created a phylogenetic tree. The results indicated 3 major clades- one with wide geographic distribution and 2 that were more limited, however no obvious geographic barriers exist between them. There also appeared to be 8 biogeographic regions of A. normalis that correspond to climate. Based on results of the phylogenetic tree, The authors conclude that A. normalis diverged between 1.05-0.16 Ma, before the temporary land bridge existed across the Qiongzhou Strait.

Line 184 - can you please clarify- were the separate runs combined? Did they converge?

OUR REPLY— In this study MrBayes ran two completely independent and simultaneous analyses from different random trees. We set Nruns=2 to run the program. In the beginning of analysis, the two runs sampled different trees, but then they went convergence, the samples of the two trees were very similar. The trees of two runs converge at last. 

Line 207- can you provide more information about parameters of the model were selected- tree prior, clock model, etc. Was any model selection performed? Are you using a model that was reported previously.

OUR REPLY— We had provided such information in Line 207-214 already. 

Line 212- please check the references throughout. This reference was not correct.

OUR REPLY— We had checked the reference ([58] in the last version of manuscript) and corrected the references following your suggestion. 

Line 544 subclade Ib AND Subclade III

OUR REPLY—We had corrected it following your suggestion.

Line 554 BY geographic barriers

OUR REPLY—We had corrected it following your suggestion.

Line 557-577- how do your phylogeographic regions differ by the ones described by He and Zhang.

OUR REPLY—We added a description in lines 514-517 to response your suggestion.

Line 605 - can you suggest any alternative explanations?

OUR REPLY—We had thought about that Clade III might be a lineage close to population in Indochina Peninsula, across South China Sea. However, we need additional specimen from Indochina Peninsula (Vietnam) to prove it. Currently, we only can discuss it within Hainan Island based our limited data. 

Line 637 - and BECOME isolated

OUR REPLY—We had corrected it following your suggestion.

--- end of comments

---

## [Decision Letter · Decision Letter 6]

14 Nov 2022

PONE-D-20-37420R6Phylogeography of Aphyocypris normalis Nichols and Pope, 1927 at Hainan Island and adjacent areas based on mitochondrial DNA dataPLOS ONE

Dear Dr. Jang-Liaw,

Thank you for submitting your manuscript to PLOS ONE. After careful consideration, we feel that it has merit but does not fully meet PLOS ONE’s publication criteria as it currently stands. Therefore, we invite you to submit a revised version of the manuscript that addresses the points raised during the review process.

We look forward to receiving your revised manuscript.

Kind regards,

Tzen-Yuh Chiang

Academic Editor

PLOS ONE

Reviewers' comments:

Reviewer's Responses to Questions

**Comments to the Author**

1. If the authors have adequately addressed your comments raised in a previous round of review and you feel that this manuscript is now acceptable for publication, you may indicate that here to bypass the “Comments to the Author” section, enter your conflict of interest statement in the “Confidential to Editor” section, and submit your "Accept" recommendation.

Reviewer #4: All comments have been addressed

Reviewer #6: All comments have been addressed

2. Is the manuscript technically sound, and do the data support the conclusions?

Reviewer #4: Partly

Reviewer #6: Yes

3. Has the statistical analysis been performed appropriately and rigorously? 

Reviewer #4: Yes

Reviewer #6: Yes

4. Have the authors made all data underlying the findings in their manuscript fully available?

Reviewer #4: Yes

Reviewer #6: Yes

5. Is the manuscript presented in an intelligible fashion and written in standard English?

Reviewer #4: Yes

Reviewer #6: Yes

6. Review Comments to the Author

Reviewer #4: In general I think the authors present a well-sampled dataset that provides additional knowledge of population structuring of this freshwater species. Below I provide two sets of comments. I do have two general concerns both revolve around the fact that this is a single-marker mitochondrial dataset. With that, I'd like to see more discussion on the limitations of such a dataset (i.e., it's not appropriate to look for gene flow using this type of data), and a bit more caution with how "deep" into the methods and analyses the authors go. I think restricting analysis to patterns of differentiation, haplotype patterns, dated phylogeny, etc., is appropriate. Inferring gene flow, or lack thereof with a single marker mitochondrial dataset is difficult if not inappropriate. Further, some discussion on how mitochondrial capture or sweeps could have affected the conclusions would be useful.

This group of comments is primarily focused on wording and sentence flow, not content.

line 39: First sentence is too broad and nebulous in my opinion, I think you can combine the first two sentences to be a bit clearer: "Biogeography is tied closely to both ecology and phylogenetics and is a key topic..." The rest of this paragraph is very nice.

General: throughout, and this is likely a type-setter comment, the dash between a span of reference numbers should probably be an n dash, unless there is something special about a span of reference numbers.

Line 50: it'd be great to provide the maximum depth of Hainan Strait right off the bat here... "Strait (maximum depth ####; Fig. 1)"

Line 52: change to "se level (ASL) near its center, with Wuzhizhan Mountain being the highest point on the island...."

Line 70: I'm having a tough time linking the first and second sentences... maybe to increase flow you could add the following to the first sentence "among species and by general ecology (e.g., aquatic vs. terrestrial; [12]). In general, freshwater aquatic animals on islands, such as freshwater fishes, should share a similar evolutionary history to terrestrial species, and provide excellent opportunities for...."

Line 82: remove "commonly"

Line 86: change to "fish on the Indochina Peninsula is very limited, for example, the first formal record of A. normalis on this peninsula was in 2011 from northern and central Vietnam..."

Line 91: change to "conducted on terrestrial species on Hainan Island, including: earthworms....."

Line 94: change to "single sampling source"

Line 97: i'd remove the information in parentheses, the statement and reference are sufficient in my opinion

Line 99: replace comprehensive with "densely sampled"

Line 100: replace "natural history" with "historical biogeography"

Line 105: replace "at" with "based on data from"

Line 106: item ii reads like two items...i'd change this list to include three items with current item ii split into two

Line 116: Does "tissue" mean frozen in this example? Or were these also ethanol-fixed, if so, you could just say "utilized ethanol-fixed tissue samples from...." if not, i'd change to "frozen and ethanol-fixed tissue samples from...."

Line 118: Split this up...(44 specimens). This sampling encompasses almost...."

Line 122: collected at...not in

Line 128: I think this would read smoother as follows "All of the samples used in this study are specimen-vouchered, coming from collections established in 2009–2012 as part......"

Line 129: replace dash with n dash

Line 150: replace crude with "genomic"

Line 154: replace "an" with "the" and remove "segment of the"

Line 155: OK, I believe that abbreviations in the abstract don't follow through to the main text...so...right here I'd say you need to define the mtDNA abbreviation and cytb (as you did)....

Line 164: replace "with the aid of" with "using"

Line 165: "calibrated" seems like an odd word here? Maybe replace with "edited" or "trimmed"?

Line 271: add period after length

Line 403: Fst...need to clearly define in methods how this was calculated as there are multiple different ways to calculate Fst

Line 407: I'm having a tough time with this interpretation, there are major issues with using a single mitochondrial marker to infer anything about geneflow...the mitochondrion doesn't recombine... Further, low genetic divergence between regions in general can be explained by just low divergence (neutral or selective) or gene flow. And when talking about a mitochondrial marker....gene flow would either produce no change or entirely erase an evolutionary history due to the fact that these regions don't recombine...they either don't interact or replace.

Line 434: this is a poor definition of Fst, need to include at least some mention of relative...

Line 473: insufficient isolation barrier or just recent isolation...without sufficient time to build up differential mutations even if completely isolated.

Line 574: I think adding some clarity and discussion here would be good. First, a landbridge only provide connectivity for terrestrial animals...it may be a stronger barrier for freshwater species if there are not connecting drainages than saltwater may be. I think some commentary on this should be made...given unknown connectivity of freshwater drainage systems between mainland china and hainan island during terrestrial continuity, we are uncertain if there was freshwater connection or not, and if not, this terrestrial barrier may have been stronger than a saltwater aquatic barrier....

Line 642: replace variation with divergence...however, as eluded to above...I'd restrain from making too strong claims on taxonomy and species status based on a single mitochondrial marker. Mitochondrial capture/sweeps could have occurred, erasing the mitochondrial history even with very limited gene flow, and the nuclear genome could retain very large differences, not detectable with cytb sequencing along

Line 644: this sentence doesn't make sense...yes mtDNA is maternal, but comparing maternal origin with phylogenies at the species level or higher seems to not be a valid comparison. Comparing mitochondrial data with nuclear genomic data may provide different conclusions, using mtDNA data at population or species levels should change much...

Line 646: I'd suggest using reduced representation genome-wide markers

General comments:

Gene-flow: As mentioned above, I have strong reservations with inference of gene-flow, or lack thereof, from a single mitochondrial marker (cytb), see above for reasons. For this reason, I think it's important to either reduce the amount of discussion on gene flow or include a cautionary section on why a single mitochondrial marker and mitochondria in general are not good for detecting gene flow. Bottomline, I think there needs to be much more circumspection with inferring population structure and gene-flow from a single mitochondrial marker.

Landbridges: I'd appreciate more discussion on land bridges and how they affect freshwater aquatic organisms...as mentioned above, a landbridge between Hainan and China may have been a stronger barrier to gene flow than the current saltwater barrier, and without known patterns of freshwater drainages on this landbirdge (I assume), it's difficult to accurately infer the expectation for degree of this barrier to a freshwater species.

Reviewer #6: The manuscipt of phylogeography of Aphyocypris normalis at Hainan Island present a rather interesting study, that after these 5 rounds of revision has been considerably improved. There are only a few minor points that, in my opinion need revision, abd thus I suggest the manuscript to be accepted right after the authors proceed with these corrections, without the need for a further revision round.

Detailed points:

Line 105: the possible isolation mechanism of freshwater...

Lines 358 and forth: the terms used to describe the network (group, clade, subclade) are at points relatively confusing as they mix with the terms of the phylogenetic structure. For that I suggest the authors to name the compoenents of this network with a precise name (f.e. Cluster, Super-clusters), and follow that terminology with accuracy.

Line 439: SAMOVA indeed showed that K=3 has the highest value in terms of grouping options ; however, as K=4 was also very very close to this value, I would like the authors to say a few things more about this clustering option as well. What was this fourth cluster that exhbited such close possiblity?

Line 482: which might separate Subclade Ib...

Line 645-648: this sentence is indeed very important, as the authors rightfully acknowledge the fact that additional markers should be employed to understand the identity and phylogeny, as using only mtDNA marker might exhbiti only the single-organelle point of view.

7. PLOS authors have the option to publish the peer review history of their article (what does this mean?). If published, this will include your full peer review and any attached files.

Reviewer #4: No

Reviewer #6: **Yes: **Dimitrios Avtzis

---

## [Author Response · Author response to Decision Letter 6]

29 Dec 2022

Again, we deeply appreciate the reviewer’s great comments and suggestions on this work. The manuscript has been major revised accordingly as possible we can, and it had been modified completely almost. We accept almost all suggestions the reviewers offered. The following are our answers to the comments which we did not follow completely. Others were all accept by the authors.

Reviewer #4: In general I think the authors present a well-sampled dataset that provides additional knowledge of population structuring of this freshwater species. Below I provide two sets of comments. I do have two general concerns both revolve around the fact that this is a single-marker mitochondrial dataset. With that, I'd like to see more discussion on the limitations of such a dataset (i.e., it's not appropriate to look for gene flow using this type of data), and a bit more caution with how "deep" into the methods and analyses the authors go. I think restricting analysis to patterns of differentiation, haplotype patterns, dated phylogeny, etc., is appropriate. Inferring gene flow, or lack thereof with a single marker mitochondrial dataset is difficult if not inappropriate. Further, some discussion on how mitochondrial capture or sweeps could have affected the conclusions would be useful.

OUR REPLY—Thank you for your comments. Indeed multiple-markers datasets are more recommended on phylogeography researches currently, especially with nuclear DNA datasets. However, phylogenetic research using multigene datasets requires more funding/resource to perform as well. With limited financial support in this study, we must allocated resources carefully. Since one of our main goals is to analyze as many A. normalis as possible from different populations, our funding can only choose to use one marker for analysis. Based on our previous experience, nuclear genes always show fewer information on genetic variation rather than mtDNA markers. Under the condition of limited resources, we decided to use mtDNA cytb gene, a widely used marker on phylogenetic researches, in this study.

>Line 116: Does "tissue" mean frozen in this example? Or were these also ethanol-fixed, if so, you could just say "utilized ethanol-fixed tissue samples from...." if not, i'd change to "frozen and ethanol-fixed tissue samples from...."

OUR REPLY—The specimens we used should be "utilized ethanol-fixed tissue samples”. We had modified the manuscript following your comment.

>Line 403: Fst...need to clearly define in methods how this was calculated as there are multiple different ways to calculate Fst

OUR REPLY—Both AMOVA and Fst were computed using ARLEQUIN in this work. We had modified the manuscript following your comment.

>Line 407: I'm having a tough time with this interpretation, there are major issues with using a single mitochondrial marker to infer anything about geneflow...the mitochondrion doesn't recombine... Further, low genetic divergence between regions in general can be explained by just low divergence (neutral or selective) or gene flow. And when talking about a mitochondrial marker....gene flow would either produce no change or entirely erase an evolutionary history due to the fact that these regions don't recombine...they either don't interact or replace.

OUR REPLY—In theory you are right. However, mtDNA has been widely used in many phylogenetic studies and constructed many evolutionary hypotheses. We cannot fully understand all the details of the mtDNA mutation process, but the phylogenetic relationship trees constructed by mtDNA still has a certain reference value in systematic biology.

>Line 434: this is a poor definition of Fst, need to include at least some mention of relative...

OUR REPLY—We had modified the manuscript following your comment in Line 434—436.

>Line 473: insufficient isolation barrier or just recent isolation...without sufficient time to build up differential mutations even if completely isolated.

OUR REPLY—We think that the Qiongzhou strait plays an insufficient isolation barrier due to the short distance between Hainan and mainland; and the sea is shallow there. It is much possible to form a landbridge during the ice age in Qiongzhou strait. 

>Line 574: I think adding some clarity and discussion here would be good. First, a landbridge only provide connectivity for terrestrial animals...it may be a stronger barrier for freshwater species if there are not connecting drainages than saltwater may be. I think some commentary on this should be made...given unknown connectivity of freshwater drainage systems between mainland china and hainan island during terrestrial continuity, we are uncertain if there was freshwater connection or not, and if not, this terrestrial barrier may have been stronger than a saltwater aquatic barrier....

OUR REPLY—We partially agree your point. However, the landbridge hypo-theses are in fact widely accepted in distribution researches on freshwater animals, such in those researches from Japan and Taiwan. 

>Line 642: replace variation with divergence...however, as eluded to above...I'd restrain from making too strong claims on taxonomy and species status based on a single mitochondrial marker. Mitochondrial capture/sweeps could have occurred, erasing the mitochondrial history even with very limited gene flow, and the nuclear genome could retain very large differences, not detectable with cytb sequencing along

OUR REPLY—In this work, we displayed the genetic divergence in mtDNA aspect. This is our main goals in this project. It must be different from researches focused on other (nuclear) markers or NGS data. They must be good stories, but currently we like to tell the mtDNA one. 

>Line 644: this sentence doesn't make sense...yes mtDNA is maternal, but comparing maternal origin with phylogenies at the species level or higher seems to not be a valid comparison. Comparing mitochondrial data with nuclear genomic data may provide different conclusions, using mtDNA data at population or species levels should change much...

OUR REPLY—Sometimes some mito-nuclear discordance on phylogeny comparisons in species level can be found in animals where males have larger daily movement distances and home-range sizes than females. The limitation of female movement may initiate phylogenetic structure of the maternally inherited mtDNA, while increased male movement drives broader gene flow of the nuclear genome. MtDNA data shows its unique value different from nuclear data in such case. It can contribute in phylogenies at the species level indeed.

General comments:

>Gene-flow: As mentioned above, I have strong reservations with inference of gene-flow, or lack thereof, from a single mitochondrial marker (cytb), see above for reasons. For this reason, I think it's important to either reduce the amount of discussion on gene flow or include a cautionary section on why a single mitochondrial marker and mitochondria in general are not good for detecting gene flow. Bottomline, I think there needs to be much more circumspection with inferring population structure and gene-flow from a single mitochondrial marker.

OUR REPLY—We agree your point. We also had mentioned in Line 650—654 to remind readers again that this is a single-marker study. 

>Landbridges: I'd appreciate more discussion on land bridges and how they affect freshwater aquatic organisms...as mentioned above, a landbridge between Hainan and China may have been a stronger barrier to gene flow than the current saltwater barrier, and without known patterns of freshwater drainages on this landbirdge (I assume), it's difficult to accurately infer the expectation for degree of this barrier to a freshwater species.

OUR REPLY—It must be interesting to discuss the patterns of the possible freshwater drainages on landbirdge between Hainan and mainland. However, we found the references are very limited on the geology/oceanography in Qiongzhou strait, even from Chinese journals. We are sorry that such discussion is somewhat beyond the core objectives of this work (and our ability)… 

Reviewer #6: The manuscipt of phylogeography of Aphyocypris normalis at Hainan Island present a rather interesting study, that after these 5 rounds of revision has been considerably improved. There are only a few minor points that, in my opinion need revision, abd thus I suggest the manuscript to be accepted right after the authors proceed with these corrections, without the need for a further revision round.

OUR REPLY—Thank you very much for your helpful comments and long-term contribution on this work. Thank you. 

>Line 439: SAMOVA indeed showed that K=3 has the highest value in terms of grouping options; however, as K=4 was also very very close to this value, I would like the authors to say a few things more about this clustering option as well. What was this fourth cluster that exhbited such close possiblity?

OUR REPLY—Indeed, the Fct values of K=3 and K=4 are very close. The groupings of both hypothese are very close as well; only the Ninyuang R’s specimen was separated from others (populations of Changhua and Wanlou Rivers) in K=4 hypothesis. In our earlier version of the SAMOVA results we described both grouping hypotheses, but it seemed too details and we remove the K=4 hypothesis in our original manuscript. We remind the K=4 result briefly following your suggestion in this revised manuscript following your comment.

--- end of comments

---

## [Decision Letter · Decision Letter 7]

1 Feb 2023

PONE-D-20-37420R7Phylogeography of Aphyocypris normalis Nichols and Pope, 1927 at Hainan Island and adjacent areas based on mitochondrial DNA dataPLOS ONE

Dear Dr. Jang-Liaw,

Thank you for submitting your manuscript to PLOS ONE. After careful consideration, we feel that it has merit but does not fully meet PLOS ONE’s publication criteria as it currently stands. Therefore, we invite you to submit a revised version of the manuscript that addresses the points raised during the review process.

We look forward to receiving your revised manuscript.

Kind regards,

Tzen-Yuh Chiang

Academic Editor

PLOS ONE

Journal Requirements:

Reviewers' comments:

Reviewer's Responses to Questions

**Comments to the Author**

1. If the authors have adequately addressed your comments raised in a previous round of review and you feel that this manuscript is now acceptable for publication, you may indicate that here to bypass the “Comments to the Author” section, enter your conflict of interest statement in the “Confidential to Editor” section, and submit your "Accept" recommendation.

Reviewer #7: All comments have been addressed

2. Is the manuscript technically sound, and do the data support the conclusions?

Reviewer #7: Yes

3. Has the statistical analysis been performed appropriately and rigorously? 

Reviewer #7: Yes

4. Have the authors made all data underlying the findings in their manuscript fully available?

Reviewer #7: Yes

5. Is the manuscript presented in an intelligible fashion and written in standard English?

Reviewer #7: Yes

6. Review Comments to the Author

Reviewer #7: The article describes the values of genetic diversity, as well as its spatial distribution and ancestry-descent relationships based on a single gene. This is a limitation, but the information is undoubtedly relevant and provides a coherent explanation for the observed patterns. This limitation of using a single gene is identified by the authors, who suggest interesting lines of research proposals for the future.

212 I suggest the use of mode of previously reported molecular substitution rate instead mean

234 Combining the 19-25 populations that have intermediate to the Songtao Reservoir seems like a good idea, but can't a lentic system be a barrier for organisms that inhabit lotic systems? Explain

241-242 Define what values of Fst are considered High and low and references if corresponds ej Freeland 2005

243 Mention if the Fst values were computed using the information of sequences or just the haplotype frequency as the reviewer 4 suggested previously

272 Include the transition transversion ratio and mention the number of private haplotypes

278-283 It would be a good idea to explain in the discussion that the pattern of lower diversity in islands is common with respect to the continent and can be used to approximate the historical Ne of the populations. It can be suggested that in Zhubi river there is a higher Ne, i.e. how can it be explained that with an N of 2 it is the highest value? Considering that there are sites such as Wanquan river with n =10 with zero h and pi

285. Explain the average number of differences between these 12 haplotypes to get an idea of whether they are single nucleotide substitutions or involve several substitutions.

286-287 Indicate the number of differences of the shared haplotype between the Wanquan and Nandu, with respect to their closest haplotype. I.e. both systems are close but it is possible that it is the same haplotype by convergence?

292 haplotype 4 the ancestral one? Explain

366-369 h4 and h1 are the ancestral ones? Explain

419-420 Are there data on the characteristics of the rivers? Oxygen, temperature, which may be more relevant than what happens out of the water?

475-479 Mention that in addition to isolation by distance, there are other alternatives (e.g. isolation by environment).

491-496 Since the strait is not so relevant is it worth considering the depth of the strait, that space was land at glacial maximum?

497-498 Water characteristics of lotic systems this may be more relevant than just precipitation or temperature.

S2 please change the order of the two below figures d and c

7. PLOS authors have the option to publish the peer review history of their article (what does this mean?). If published, this will include your full peer review and any attached files.

Reviewer #7: No

---

## [Author Response · Author response to Decision Letter 7]

12 Feb 2023

Again, we deeply appreciate the reviewer’s great comments and suggestions on this work. The manuscript has been major revised accordingly as possible we can, and it had been modified completely almost. We accept almost all suggestions the reviewers offered. The following are our answers to the comments.

Reviewer #7: The article describes the values of genetic diversity, as well as its spatial distribution and ancestry-descent relationships based on a single gene. This is a limitation, but the information is undoubtedly relevant and provides a coherent explanation for the observed patterns. This limitation of using a single gene is identified by the authors, who suggest interesting lines of research proposals for the future.

>>212 I suggest the use of mode of previously reported molecular substitution rate instead mean

OUR REPLY—Thank you for your suggestion. The previously reported molecular substitution rates for cyprinid cytb ranged from roughly 0.6% to 1.3% Myr. We had adopted a frequently-used cytb clock rate covered in above range, following Ketmaier et al. [63] and Zhang et al. [64]. Please allow us to keep the former selection in the manuscript.

>>234 Combining the 19-25 populations that have intermediate to the Songtao Reservoir seems like a good idea, but can't a lentic system be a barrier for organisms that inhabit lotic systems? Explain

OUR REPLY—It is a good idea to discuss the barrier mechanism of Songtao Reservoir on freshwater animals in Hainan. A huge lentic system such as Songtao Reservoir can be a barrier for small-sized non- pelagic freshwater fish like Aphyocypris normalis. It is too difficult and risky for adult A. normalis to move through a deep, wide water body, for they will be easy to be hunted by predators during their movement without hidden substrates they prefer. Pelagic phase eggs or larvae are possible, but with huge risk for being food of others as well. Populations of sites 19—25 were widely separated by Songtao Reservoir in Nandu River as two parts (sites 19—22 in the upstream of Songtao Reservoir, and sites 23—25 in the downstream of it), and the genetic data were I accordance with this distribution pattern. 

We had modified the manuscript following your comment in lines 547—559.

>>241-242 Define what values of Fst are considered High and low and references if corresponds ej Freeland 2005

OUR REPLY—We had modified the manuscript following your comment in lines 440—442 and added the reference Freeland 2018 as [72].

>>243 Mention if the Fst values were computed using the information of sequences or just the haplotype frequency as the reviewer 4 suggested previously

OUR REPLY—The Fst values were computed using the information of haplotype instead of all sequences. We had modified the manuscript following your comment in lines 243—244.

>>272 Include the transition transversion ratio and mention the number of private haplotypes

OUR REPLY—The estimated Transition/Transversion ratio is 11.50, which was conducted by MEGA, and 41 are private haplotypes. We had modified the manuscript following your comment in lines 272—275.

>>278-283 It would be a good idea to explain in the discussion that the pattern of lower diversity in islands is common with respect to the continent and can be used to approximate the historical Ne of the populations. It can be suggested that in Zhubi river there is a higher Ne, i.e. how can it be explained that with an N of 2 it is the highest value? Considering that there are sites such as Wanquan river with n =10 with zero h and pi

OUR REPLY—Thank you for your nice suggestion. We had modified the manuscript following your comment in lines 493—502. 

>>285. Explain the average number of differences between these 12 haplotypes to get an idea of whether they are single nucleotide substitutions or involve several substitutions.

OUR REPLY—These haplotypes could be separated into two groups. One belonged to Clade I (h4, h6, h26, h28, and h29), in which they were single or two nucleotide substitutions to each other. Another belonged to Subclade Ib (h20, h21, h22, h23, h24, h25, and h27), in which they are single or several substitutions (up to 11) to each other. We had modified the manuscript following your comment in lines 286-290.

>>286-287 Indicate the number of differences of the shared haplotype between the Wanquan and Nandu, with respect to their closest haplotype. I.e. both systems are close but it is possible that it is the same haplotype by convergence?

OUR REPLY—The haplotypes found in Nandu R. were h4, h6, and h20—h29 (please check the Table 1), and the only haplotype found in Wanquan R. were h4. According to h4 could be found in other river system (Xi R. of Pearl R.in mainland, in our opinion, the h4 is possibly the most ancestral haplotype in this study, for its wide distribution and central position in the statistical parsimony haplotype network (Fig.4), rather than being a convergence haplotype in different river systems. 

>>292 haplotype 4 the ancestral one? Explain

>>366-369 h4 and h1 are the ancestral ones? Explain

OUR REPLY—Both h4 and h1 are the possible ancestral ones. They were present in more than three populations, and h4 is the most frequent by far. Besides, h4 is also the only haplotype shared between Hainan Island and mainland China (sites 4, 5, 23, A5; Fig. 1). These are the characteristics of possible ancestral haplotypes, and h4 is more possible to be ancestral haplotype than h1 for a wider distribution across the strait. In Fig. 4 the h4 occupies the center of the statistical parsimony haplotype network. It is also a characteristic of possible ancestral haplotypes. 

>>419-420 Are there data on the characteristics of the rivers? Oxygen, temperature, which may be more relevant than what happens out of the water?

OUR REPLY—We have no data on the characteristics of the rivers. It is somewhat pity to absent from it in discussion. However, please understand that it will be too complex to have such discussion in this paper. 

>>475-479 Mention that in addition to isolation by distance, there are other alternatives (e.g. isolation by environment).

OUR REPLY—We had modified the manuscript following your comment in lines 592-593.

>>491-496 Since the strait is not so relevant is it worth considering the depth of the strait, that space was land at glacial maximum?

OUR REPLY—Based on our data and other’s works, the geographic isolation of the Qiongzhou strait between Hainan Island and mainland China did not create efficient isolation barriers. However, it is necessary to point it out clearly for it is generally accepted that geographical isolation is a very efficient isolation mechanism for freshwater animals. Furthermore, a former reviewer suggested us to describe the depth information of Qiongzhou Strait. For some readers it seems being necessary to describe the depth of the strait anymore. 

>>497-498 Water characteristics of lotic systems this may be more relevant than just precipitation or temperature.

OUR REPLY—We had modified the manuscript following your comment in Line 508.

>>S2 please change the order of the two below figures d and c

OUR REPLY—We think that you meant S1. We had modified the order of the two below figures d and c following your suggestion.

---

## [Decision Letter · Decision Letter 8]

16 Feb 2023

Phylogeography of Aphyocypris normalis Nichols and Pope, 1927 at Hainan Island and adjacent areas based on mitochondrial DNA data

PONE-D-20-37420R8

Dear Dr. Jang-Liaw,

We’re pleased to inform you that your manuscript has been judged scientifically suitable for publication and will be formally accepted for publication once it meets all outstanding technical requirements.

Kind regards,

Tzen-Yuh Chiang

Academic Editor

PLOS ONE

Additional Editor Comments (optional):

Reviewers' comments:

Reviewer's Responses to Questions

**Comments to the Author**

1. If the authors have adequately addressed your comments raised in a previous round of review and you feel that this manuscript is now acceptable for publication, you may indicate that here to bypass the “Comments to the Author” section, enter your conflict of interest statement in the “Confidential to Editor” section, and submit your "Accept" recommendation.

Reviewer #7: All comments have been addressed

2. Is the manuscript technically sound, and do the data support the conclusions?

Reviewer #7: Yes

3. Has the statistical analysis been performed appropriately and rigorously? 

Reviewer #7: Yes

4. Have the authors made all data underlying the findings in their manuscript fully available?

Reviewer #7: Yes

5. Is the manuscript presented in an intelligible fashion and written in standard English?

Reviewer #7: Yes

6. Review Comments to the Author

Reviewer #7: All my concerns about the manuscript was attended. The response of authors was according the theory as far as I know

7. PLOS authors have the option to publish the peer review history of their article (what does this mean?). If published, this will include your full peer review and any attached files.

Reviewer #7: No

---

## [Editor Report · Acceptance letter]

20 Feb 2023

PONE-D-20-37420R8 

Phylogeography of *Aphyocypris normalis* Nichols and Pope, 1927 at Hainan Island and adjacent areas based on mitochondrial DNA data 

Dear Dr. Jang-Liaw:

I'm pleased to inform you that your manuscript has been deemed suitable for publication in PLOS ONE. Congratulations! Your manuscript is now with our production department. 

Kind regards, 

on behalf of

Dr. Tzen-Yuh Chiang 

Academic Editor

PLOS ONE